

# The UKC2 regional coupled environmental prediction system

Huw W. Lewis[1], Juan Manuel Castillo Sanchez[1], Jennifer Graham[1], Andrew Saulter[1], Jorge Bornemann[1], Alex Arnold[1], Joachim Fallmann[1], Chris Harris[1], David Pearson[1], Steven Ramsdale[1], Alberto Martínez de la Torre[2], Lucy Bricheno[3], Eleanor Blyth[2], Vicky Bell[2], Helen Davies[2], Toby R. Marthews[2], Clare O'Neill[1], Heather Rumbold[1], Enda O'Dea[1], Ashley Brereton[3], Karen Guihou[3], Adrian Hines[1], Momme Butenschon[4], Simon J. Dadson[5], Tamzin Palmer[1], Jason Holt[3], Nick Reynard[2], Martin Best[1], John Edwards[1], John Siddorn[1]

[1]Met Office, Exeter, EX1 3PB, UK
[2]Centre for Ecology & Hydrology, Wallingford, OX10 8BB, UK
[3]National Oceanography Centre, Liverpool, L3 5DA, UK
[4]Plymouth Marine Laboratory, Plymouth, PL1 2LP, UK
[5]School of Geography and the Environment, University of Oxford, South Parks Road, Oxford, OX1 3QY, UK

*Correspondence to*: Huw W. Lewis (huw.lewis@metoffice.gov.uk)

**Abstract.** It is hypothesised that more accurate prediction and warning of natural hazards, such as of the impacts of severe weather mediated through various components of the environment, requires a more integrated Earth System approach to forecasting. This hypothesis can be explored using regional coupled prediction systems, in which the known interactions and feedbacks between different physical and biogeochemical components of the environment across sky, sea and land can be simulated. Such systems are becoming increasingly common research tools. This paper describes the development of the UKC2 regional coupled research system, which has been delivered under the UK Environmental Prediction Prototype project. This provides the first implementation of an atmosphere-land-ocean-wave modelling system focussed on the United Kingdom and surrounding seas at km-scale resolution. The UKC2 coupled system incorporates models of the atmosphere (Met Office Unified Model), land surface with river routing (JULES), shelf-sea ocean (NEMO) and ocean waves (WAVEWATCH III). These components are coupled, via OASIS3-MCT libraries, at unprecedentedly high resolution across the UK within a north-west European regional domain. A research framework has been established to explore the representation of feedback processes in coupled and uncoupled modes, providing a new research tool for UK environmental science. This paper documents the technical design and implementation of UKC2, along with the associated evaluation framework. An analysis of new results comparing the output of the coupled UKC2 system with relevant forced control simulations for 6 contrasting case studies of 5-day duration is presented. Results demonstrate that at least comparable performance can be achieved with the UKC2 system to its component control simulations. For some cases, improvements in air temperature, sea surface temperature, wind speed, significant wave height and peak wave period highlight the potential benefits of coupling between environmental model components. Results also illustrate that the coupling itself is not sufficient to address all known model issues. Priorities for future development of the UK Environmental Prediction framework and component systems are discussed.



## 1 Introduction

Development from single-component models towards more fully integrated regional coupled environmental prediction systems across atmosphere, land and sea is becoming an increasingly viable approach for research (e.g. Pullen et al., 2006; Pullen et al., 2007; Chen et al., 2010; Sandery et al., 2010; Warner et al., 2010; Renault et al., 2012; Carniel et al., 2016;

Licer et al., 2016; Bruneau and Toumi, 2016) and operational applications (e.g. Pellerin et al., 2004; Smith et al., 2013; Durnford et al., 2017) to improve the representation of feedbacks in simulations and predictions of the Earth System. This is consistent with the vision to accelerate progress in Earth System prediction from weather to climate and global to local scales (e.g. Shapiro et al., 2010); Brunet et al.,2015).

It is hypothesised that more accurate prediction and warning of the impacts of severe weather requires a more integrated

approach to forecasting. While mature national-scale operational forecasting capability on timescales of hours to days, such as delivered by the Met Office in the UK (e.g. Lewis et al., 2015), typically includes simulations of the atmosphere and land surface, hydrology, ocean, waves and other environmental components, these forecast systems tend to be developed and run in relative isolation, with limited information sometimes exchanged between systems by file input through initial conditions or forcing data. Recent advances in the skill, resolution and information content (e.g. on uncertainty) of environmental

models along with increases in operational computational resources now make it more relevant to attempt to directly integrate or couple forecast models of these distinct systems at high resolution.

Greatest sensitivity to coupling on timescales of hours to days and at high resolution is more likely for phenomena with sensitivity to wind and precipitation, where local geographic or meteorological details can significantly affect model skill. For example, guidance on the evolution and impact of severe storm surges (e.g. Staneva et al., 2016) requires detailed

prediction and synthesis of both the atmospheric (pressure, wind) and ocean state (tides, waves). In addition, numerous high impact flood events illustrate that floods result from a combination of weather, land and river conditions and an evolving inland flood situation (e.g. Stephens and Cloke, 2014) and requires integration  of the atmospheric (rainfall), land surface (soil moisture, runoff) and hydrological state (aquifer state, river flow, level).

Coupling should be particularly relevant for regional predictions around the UK because of its maritime location with

prevailing weather approaching from the south-west over a long ocean fetch with large potential for air-sea exchange. There are also significant populations and critically important national infrastructure located on or near coastlines, often vulnerable to multiple hazards originating from the atmosphere, oceans or land.

Better understanding the potential benefits of more integrated approaches requires investment in technical and scientific development of coupled prediction systems, as described in this paper. The UK Environmental Prediction Prototype project

was initiated in April 2014 in order to begin exploring the potential benefits and limitations of coupling relative to current uncoupled systems and tools in the UK context. Particular drivers include the need to provide evidence on whether more integrated systems are capable of:

i)          improving the accuracy and skill of predictions over current operational approaches,



ii)      providing either new or more relevant and consistent hazard advice to users, particularly concerning multi-hazards,

iii)      improving the analysis, understanding and process representation of the known feedbacks between components.

The UKC2 represents a first implementation of a coupled atmosphere-land-ocean-wave modelling system focussed on the UK at km-scale resolution. An interim atmosphere-land-ocean coupled prototype configuration, termed UKC1, was

previously developed based on a slightly different domain and earlier code revisions. This configuration was not formally released for research application, but formed the foundation for development of the UKC2 configuration described here,

This paper is organised as follows. Section 2 introduces the UKC2 regional coupled prediction system, including the coupling framework and interactions between components represented. Section 3 provides details of the UKA2 atmosphere, UKL2 land surface, UKO2 ocean and UKW2 wave model configurations, and which are coupled together within UKC2. The

evaluation framework and case study configurations are introduced in Sect. 4. A summary of initial evaluation results based on 6 contrasting case study simulations over 5-day periods is presented in Sect. 5. Finally, priorities for ongoing system development towards a UKC3 coupled configuration and beyond are outlined in Sect. 6.

## 2 The UKC2 regional coupled prediction system

The second research-mode regional coupled prediction system UKC2 consists of configurations of the Met Office Unified

Model (MetUM) atmosphere (version 10.1; Cullen 1993), and JULES (Joint UK Land Environment Simulator) land surface model (version 4.2; Best et al., 2011; Clark et al., 2011), coupled to the NEMO (Nucleus for European Models of the Ocean) model (version3.6, revision 5518; Madec et al., 2016) and WAVEWATCH III wave model (version 4.18; Tolman et al., 2014). Coupling is achieved through use of the OASIS3-MCT (Ocean-Atmosphere-Sea Ice-Soil) coupling libraries (version 2.0; Valcke et al., 2015). While UKC2 refers to the coupled system, the equivalent uncoupled atmosphere, ocean and wave

configurations of these components will be referred to as UKA2, UKO2 and UKW2 respectively.

The skill of UKC2 critically depends on the long-term development of each of component model codes and systems. This new system development adopts a seamless approach to weather and climate prediction (e.g. Brown et al., 2012) and benefits from the efficiencies provided through use of common model codes, system design and coupling frameworks in UKC2 as initially developed for more established coupled systems applied in global numerical weather prediction research (e.g.

Williams et al., 2015), monthly to decadal forecasting (e.g. MacLachlan et al., 2015), climate prediction (e.g. Roberts et al., 2016) and Earth System modelling (e.g. Jones et al., 2011).

In contrast to these systems, UKC2 represents the first coupled application in a high resolution regional context at the Met Office. It is also the first system in which coupling between a wave model and the MetUM atmosphere and NEMO ocean components have been fully tested. Further, UKC2 marks an important step in the development towards a more integrated

land surface and terrestrial hydrology system capable of consistently simulating the flow of water through the hydrological cycle from the sky to the sea at high resolution.



Namelists describing the configuration for all components discussed in this paper are defined as suites under the rose vn6.0 framework for managing and running model systems (http://metomi.github.io/rose/doc/rose.html). The suite framework is described further in Sect. 4, with all configurations described made available to registered users under a repository at https://code.metoffice.gov.uk/trac/roses-u. A more detailed description of the namelists used is also included in the

Supplementary Material to this paper.

All model components described are set up in free-running simulation mode, with no data assimilation applied during case study runs. As described in Sect. 3, initialisation and model boundary forcing drawn from operational archives where relevant act to provide some updating to observed conditions.

Exactly the same codes were compiled and run for both coupled and uncoupled configurations. This ensures that all runs

performed within a case study experiment use identically built code, even though many of the adaptations described here are only relevant and triggered when coupling is enabled in the configuration namelists.

## 2.1 Establishing a common domain for UK environmental prediction

The UKC2 system and its component models are defined on a common domain, though on different grids. The domain selected is shown in Fig. 1. This domain aims to deliver a computationally affordable system, but which covers sufficient

spatial extent to provide robust atmosphere, land surface and ocean simulations over the region of interest. This domain now also represents the common domain selected and in use in uncoupled mode configurations at the Met Office as the operational weather forecast UK atmosphere (UKV; e.g. Tang et al., 2013) from November 2016 and in research development for future operational implementation as an ocean forecast system (AMM15; Graham et al., 2017).

To reduce the impact of smaller longitudinal grid cell sizes at higher latitudes, all model grids are defined as rectilinear grids

on a rotated latitude and longitude coordinate system. The computational North Pole origin is set at an actual position of 37.5° N, 177.5° E. Further detail on the model grids selected for each UKC2 component is provided in Sect. 3.

## 2.2 Coupling framework

Hewitt et al. (2011) provided a comprehensive description of MetUM and NEMO coupling (with sea ice) in the context of global coupled climate simulations. As with any coupled prediction system, the primary aim for development of UKC2 is to

better represent the feedbacks between components of the Earth system. The initial focus is on shorter-timescales, from hours to days, such that interaction with longer timescale processes such as atmospheric chemistry and composition is not considered, and no sea ice is assumed to form in the domain. Rather, a representation of the feedbacks at high resolution through exchange of momentum, heat and freshwater is attempted. Figure 2 illustrates the coupling exchanges considered within UKC2. The lack of sea ice or coupling to 3-dimensional fields such as required for atmospheric composition makes

UKC2 a relatively simple coupled system with 18 surface fields exchanged via the OASIS3-MCT library (Valcke et. al., 2015), in addition to the coupling of the atmosphere to the land surface through the JULES implicit and explicit coupling schemes (Best et al, 2004).



Table 1 lists the variables exchanged between model components within UKC2. All fields are exchanged at a coupling frequency of 1 hour. The technical flexibility exists to exchange different fields with different coupling frequencies, and future research will investigate the sensitivity of results to the coupling frequency chosen. The need for all model components to reach a particular common simulation time before coupling exchange can place some constraints on the

5 model time step used. For simplicity, all model components are run in coupled and uncoupled mode with a 60 s time step.

The addition of a wave model within UKC2 represents an extension of the coupling fields presented by Hewitt et al. (2011), although the processing and exchange of fields via OASIS3-MCT closely mirrors that used between atmosphere and ocean components. While the technical capability has been implemented in WAVEWATCH III and NEMO code branches to exchange variables from the surface waves to ocean (e.g. Breivik et al., 2015), the relevant physics required in NEMO were

10 not implemented or tested within UKC2. This is a priority for future development within the next UKC3 system.

### 2.2.1 Generation of interpolation weights

Interpolation weights are required in order to translate fields between the different component model grids. Although OASIS3-MCT has the capability of generating interpolation weights at run-time, the interpolation between atmosphere and ocean grids is achieved by calculating weights offline using ESMF tools (Jones, 2015) to specify the mapping between

15 source and destination grids. This is more efficient, traceable, and allows checking and potentially minor adjustments to be made to the calculated weights prior to their use. Remapping of all scalar fields is achieved using first-order conservative interpolation. Remapping of vector fields (i.e. wind, wind stress and ocean currents) is achieved by bilinear interpolation. Following testing and assessment of exchanged variables, it was decided to generate remap weights between grids without taking into account the land/sea masks defined for each component. This avoided some issues which resulted from

20 calculating remap weights across grid boxes where, due to either variable grid resolution in the atmosphere or use of non-identical masks even where grids matched exactly, at least part of the grid in one component was defined as land (invalid points for remapping) while in the other grid it was defined as ocean (valid). Instead, all points in both grids were considered as potentially valid source or destination points for remapping, and the interpolation weights computed everywhere. Modifications to the relevant model codes were then implemented in code branches to ensure that exchanged variables were

25 only used where there was valid source information available or a valid destination point.

While the UKO2 ocean and UKW2 wave configurations are defined using identical bathymetry and land/sea masks, ocean variables at the surface in NEMO are defined on the full 2-d grid while wave variables in WAVEWATCH III are defined on a 1-d vector of sea points only. A simple remapping was therefore also defined for 2-d ocean to 1-d wave field exchanges. Given prior calculation of ocean to atmosphere remap weights using ESMF, it was also possible to directly apply this

translation to infer the required remapping and interpolation weights between wave and atmosphere grids.



## 2.3 A physical basis for coupled feedbacks

The coupling within UKC2 is focussed on the exchange of momentum, heat and fresh water at the surface (Fig. 2). This is mediated through the surface exchange and boundary layer schemes within JULES and the MetUM (Lock et al., 2000), and the treatment of surface boundary condition forcing within NEMO (Madec et al., 2016) and WAVEWATCH III (Tolman et al., 2014).

It is worth emphasising that some representation of the feedback processes of interest in UKC2 are already included within the uncoupled component model parameterizations. However in uncoupled simulations information about the state of other environmental components is often represented either through external files, with no feedbacks during the simulation on those forcing data. Alternatively key parameters defining the role of other components are assumed to be set as constant rather than dynamically changing through time (with or without feedbacks). Further details on the external forcing is discussed in Sect. 3. A brief summary is provided here to describe the key UKC2 coupled interactions highlighted in Fig. 2.

In general, surface exchange to the atmosphere makes the assumption that Monin-Obukhov similarity theory for the surface layer is valid and that vertical gradients of model variables for velocity $\mathbf{v}$, temperature $T$ and moisture $q$ in this region are related to surface fluxes by the following equations.

$$\frac{\partial \mathbf{v}}{\partial z} = \frac{\tau_0}{\rho_0 u_*} \frac{\phi_m(z/L)}{\kappa z} \tag{1}$$

$$\frac{\partial T}{\partial z} + \frac{g}{c_p} = -\frac{H_0}{c_p \rho_0 u_*} \frac{\phi_h(z/L)}{\kappa z} \tag{2}$$

$$\frac{\partial q}{\partial z} = -\frac{E_0}{\rho_0 u_*} \frac{\phi_h(z/L)}{\kappa z} \tag{3}$$

Note that the definition for all variables used in equations is provided in Appendix A, along with their units.

### 2.3.1 Momentum exchange

The key terms and exchanges describing momentum-related processes in UKC2 are illustrated in Fig. 2(a). The wind speed $U$ at a reference height $z$ is related to the surface friction velocity, $u_*$, and roughness length, $z_{0m}$, according to

$$U = \frac{u_*}{\kappa}\left[\log\left(\frac{z}{z_{0m}}\right) - \psi_m\right] \tag{4}$$

where the von Kármán constant, $\kappa$, has a value of 0.4. The stress exerted at the surface can then be expressed as

$$\tau_0 = \rho_a u_*^2 = \rho_a c_D |\Delta \mathbf{v}|^2 \tag{5}$$

In the MetUM boundary layer scheme, the near-surface wind speed profile is computed using surface similarity via Eq. (4), with the similarity functions, $\phi_m, \phi_h, \psi_m$, defined by Beljaars and Holtslag (1991) used for stable conditions, whilst in unstable conditions the functions of Dyer and Hicks (1970) are used.

In uncoupled atmosphere-only simulations, or over land grid cells, the surface wind speed is taken as zero, and the difference in wind speed between the surface and the first model level is simply $\Delta \mathbf{v} = \mathbf{v}_1$. When coupling to an ocean model in UKC2, the near surface wind speed is expressed relative to the ocean surface current speed, $\mathbf{v}_0$, such that $\Delta \mathbf{v} = \mathbf{v}_1 - \mathbf{v}_0$. With



$|\mathbf{v}_1| > |\mathbf{v}_0|$ in most conditions, the effective surface layer flow is relatively decelerated where the surface current aligns with the overlying wind and is accelerated where the current opposes the wind.

For model grid cells over the ocean, the roughness length for momentum depends on both the atmospheric surface layer flow and the underlying surface wave state. The momentum roughness length is related to the surface friction velocity, $u_*$, as:

$$z_{0m}(sea) = \frac{0.11\nu}{u_*} + \frac{\alpha}{g}u_*^2 \qquad (6)$$

This is a generalisation of Charnock's formula (Charnock, 1955) to include low-wind conditions (Smith, 1988) with the dynamic viscosity of air, $\nu$, having a constant value of $14 \times 10^{-6}$ ms$^{-1}$. In an uncoupled UKA2 atmosphere-only simulation, as in the operational UKV system, an empirically-based constant value for the Charnock coefficient, $\alpha$, of 0.011 is specified everywhere. When coupling to a wave model in UKC2, a spatially-varying wave-dependent Charnock parameter field calculated by WAVEWATCH III is updated and exchanged via OASIS3-MCT throughout the simulation. A parameterisation for computing the roughness length for scalars, $z_{0h}(sea)$, is then applied following Edwards (2007).

Surface stress and near surface wind speed variables provided by the atmosphere model within UKC2 act as forcing to the ocean and wave model components. The surface stress provides an upper friction boundary condition on the vertical diffusive flux in the NEMO ocean model. The effect of wind-wave interaction is described in the WAVEWATCH III model in terms of a source term, $S_{in}(k, \theta_w)$, for wave number $k$ and wave direction $\theta_w$. A variety of parameterisations that depend on the calculation of $u_*$ from input wind speed components are available in WAVEWATCH III (Tolman et al., 2014). By default in UKC2 the "ST3" wave parameterisation scheme is used, based on the growth theory of Miles (1957), modified by Janssen (1982) and Bidlot (2012). For a wave field defined with a wave action density spectrum, $N(k, \theta_w)$, and intrinsic frequency, $\sigma$, then:

$$S_{in}(k, \theta_w) = \frac{\rho_a}{\rho_w}\frac{\beta_{max}}{\kappa^2}e^Z Z^4 \left(\frac{u_*}{C} + z_\alpha\right)^2 \cos^{p_{in}}(\theta_w - \theta_u)\sigma N(k, \theta) + S_{out}(k, \theta_w) \qquad (7)$$

The $S_{out}(k, \theta_w)$ term provides a linear damping of swells (Janssen, 2004).

The input source term is then used in the calculation of the wave-supported stress, $\tau_w$, as:

$$\tau_w = \left| \int_0^{k_{max}} \int_0^{2\pi} \frac{S_{in}(k', \theta_w)}{C}(\cos \theta_w, \sin \theta_w)dk'd\theta_w + \tau_{hf}(u_*, \alpha)(\cos \theta_u, \sin \theta_u) \right| \qquad (8)$$

with $\tau_{hf}$ providing the stress supported by shorter waves (also dependent on the Charnock parameter $\alpha$). Given an input wind speed (assumed neutral) at 10 m above the surface, $U_{10m}$, and the calculated wave-supported stress $\tau_w$, the two-way feedback between the atmosphere and wave field is then described in terms of the friction velocity $u_*$. This is defined in WAVEWATCH III through a look-up table (Bidlot, 2012) describing the total surface stress $\tau = \rho u_*^2$ as a function of $U_{10}$ and $\tau_w$. An iterative calculation is performed to calculate roughness length $z_0$ and total stress $\tau$ from:

$$z_{0m} = \frac{z_{00}}{\sqrt{1 - {\tau_w}/{\tau_0}}} \qquad (9)$$

with an initial guess for $z_{00}$ on each of 10 iterations given by $\frac{\alpha_{00}\tau}{g}$, with $\alpha_{00}$ specifying a minimum possible Charnock coefficient (default value of 0.0095). Given the iterative solution for $u_*$ (from $\tau$), the roughness length $z_0$ is determined again





from the input 10 m wind speed $U_{10}$, according to Eq. (4) and the output Charnock parameter $\alpha$ diagnostic or coupling field back to the atmosphere derived using:

$$\alpha = g \frac{z_{0m}}{u_*^2} \tag{10}$$

Note again that no wave-to-ocean feedbacks which will also impact on momentum exchange across the atmosphere-ocean interface and its mixing within the ocean interior are represented in UKC2 (e.g. Breivik et al., 2015), but are planned for implementation in a future configuration.

### 2.3.2 Heat exchange

The key terms describing the exchange of heat across the land-atmosphere and ocean-atmosphere interface are illustrated in Fig. 2(b). The surface radiation budget is described by the partitioning of the total surface heating, $Q$, into solar $Q_s$ and non-solar $Q_{ns}$ components, such that:

$$Q = Q_{sr} + Q_{ns} = Q_{sr} + (Q_{LW} - Q_H - Q_E) \tag{11}$$

where $Q_{LW}$ is the longwave heating, $Q_E$ the latent heating due to evaporation and $Q_H$ the sensible heat flux.

The land-use dependent partitioning of energy over land grid cells to the vegetation and surface soil layers within the JULES land surface model is described in detail by Best et al. (2011). The surface energy balance over land can describe the rate of change of surface temperature, $T_0$:

$$C_s \frac{\partial T_0}{\partial t} = (1 - \alpha_s)Sw_\downarrow + \epsilon Lw_\downarrow - \sigma_{SB}\epsilon(T_0)^4 - H_0 - L_c E_0 - G \tag{12}$$

The turbulent surface sensible heat flux to the atmospheric boundary layer can be expressed, based on Eq. (2), as:

$$H_0 = -c_H U \left( \Delta T + \frac{g}{c_p}(z_1 + z_{0m} - z_{0h}) \right) \tag{13}$$

and the turbulent moisture flux, based on Eq. (3), as:

$$E_0 = -\rho_a c_H U \Delta q \tag{14}$$

Over the sea, if a constant sea surface temperature ($T_0 = SST$) is assumed, then the evolution of the surface sensible heat flux becomes only a function of the overlying air temperature, with $\Delta T(t) = T_1(t) - T_0$. When coupled to a dynamic ocean model however, the surface temperature $T_0$ is diagnosed directly, based on solving the primitive dynamical equations for the ocean and representing sub-grid physics due to turbulent motions and diffusion. In this case, the surface sensible heat flux to the atmosphere is then dictated by the evolution of the near-surface gradient of air and sea temperatures $\Delta T(t) = T_1(t) - T_0(t)$. The resulting surface buoyancy flux, which dictates the overlying boundary layer evolution and stability, is then:

$$\Delta B = g\beta_{T1} \left( \Delta T + \frac{g}{c_p}(z_1 + z_{0m} - z_{0h}) \right) + g\beta_{q1}\Delta q \tag{15}$$

The radiation penetrating beneath the ocean surface is treated by the NEMO model using an "RGB" scheme representing the different absorption characteristics of the ocean to different wavelength radiation (Lengaigne et al., 2007). Longwave radiation (wavelengths greater than about 700 nm) is absorbed in the upper few centimetres of the ocean, contributing to





surface heating. Shortwave radiation penetrates more deeply, causing sub-surface as well as surface heating. In UKC2, an empirically-based absorption parameter is specified to indicate that 66% of incoming radiation is non-penetrating (see Sect. 3.3). For penetrating wavelengths, the RGB scheme splits shortwave radiation into three wavebands, representing red, green and blue light. For each of the three wavebands, a chlorophyll-dependent attenuation coefficient can be specified at each

model grid point to define how the solar irradiance, $I$, penetrates with depth, $z$, into the ocean, according to:

$$I(z) = Q_{sr}\left(R_{abs}e^{-z/\xi_0} + \left(\frac{1-R_{abs}}{3}\right)\left(e^{-z/\xi_{rr}} + e^{-z/\xi_{gg}} + e^{-z/\xi_{bb}}\right)\right) \tag{16}$$

In UKC2, a constant and small chlorophyll concentration of 0.05 mg m$^{-3}$ is assumed everywhere. In a future evolution to the regional coupled system, the addition of a dynamic marine biogeochemical component (e.g. Butenschon et al., 2016) should enable biophysical feedbacks on the radiation attenuation to be considered. It would also be possible to use an estimate of the

near surface ocean chlorophyll and sediments to modify the sea surface albedo, and feedback directly on the surface radiation balance computed within the MetUM following Jin et al. (2011).

### 2.3.3 Freshwater exchange

The processes that describe the cycling of freshwater across atmosphere, land and ocean components within UKC2 are illustrated in Fig. 2(c). The partitioning of precipitation falling onto vegetation or the land surface into runoff and soil

moisture is determined by the JULES soil hydrology parameterisations (Best et al., 2011). The sub-grid scale heterogeneity of soil moisture is represented in the UKA2 and UKC2 configurations using the Probability Distributed Model (PDM; Moore, 2007). This calculates the fraction of each model grid cell that is saturated as precipitation falls into the soil stores, $F_{sat}$ according to Eq. (17):

$$F_{sat} = 1 - \left(1 - \frac{S-S_0}{S_{max}-S_0}\right)^{\frac{b}{b+1}} = 1 - \left(1 - \frac{S-S_0}{(\theta_{sat}z_{PDM})-S_0}\right)^{\frac{b}{b+1}} \tag{17}$$

where $S$ is the grid cell soil water storage, $S_0$ the minimum storage below which there is no surface saturation and $S_{max}$ is the maximum grid cell storage. Any saturation excess over the saturated area then generates surface runoff (Clark and Gedney, 2007). The sub-surface runoff (or grid cell baseflow) is obtained as the free drainage at the bottom of the soil column (at 3 m depth). The saturation fraction in the PDM scheme is controlled by the three parameters: the shape parameter, $b$, the minimum storage, $S_0$, and the depth of the surface soil column considered, $z_{PDM}$. The maximum

storage, $S_{max}$, is defined as $\theta_{sat}z_{PDM}$, where $\theta_{sat}$ is the volumetric water content at saturation. The saturation fraction is controlled by the $b$ shape parameter. Modifications to this parameterisation specifically implemented for UKC2 are described further in Sect. 3.2.

A kinematic wave equation scheme (Bell et al., 2007) has been introduced in JULES, termed the River Flow Model (RFM), to represent the routing of surface and sub-surface runoff from inland grid cells across the land surface and within the river

network out to sea. In general, a channel flow $q_d$ in either the surface or sub-surface is related to the lateral inflow into a grid cell per unit length, $r$, and a kinematic wave speed $c$, as:





$$\frac{\partial q_d}{\partial t} + c \frac{\partial q_d}{\partial x} = c(r + R) \tag{18}$$

where $R$ represents a positive or negative return flow which allows for transfer between the sub-surface and surface pathways. A derivation of the RFM routing algorithm is provided in Appendix B for clarity. The RFM incorporates a series of tuneable parameters such as surface and sub-surface wave speeds $c$ for both river and land grid cells (Table C3).

For uncoupled atmosphere-only simulations, the coastal freshwater discharge provides a useful and observable diagnostic characterising the land surface moisture state and an integrated characteristic of the model rainfall and land surface processes. In coupled simulations, the discharge provides a mass exchange boundary condition to the ocean model component. This modifies the sea surface salinity, especially in the vicinity of major river mouths. Options are available in NEMO to distribute this flux across the full depth of the water column or only the surface grid level. In future, further

development may be required to provide a more sophisticated representation of mixing processes within shallow estuarine environments. A coupled prediction system also provides a framework in which to simulate the development of inundated areas and wetlands through river overbank inundation along with inundation of the land surface at the coastline through overtopping during high sea level and stormy conditions.

Figure 2(c) also illustrates the direct input of freshwater from the atmosphere at the ocean surface. When coupled to an ocean

model, the kinematic surface boundary condition is modified by the difference in precipitation and evaporation, $P - E$, mass flux, such that for a sea surface height $\eta$:

$$w = \frac{\partial \eta}{\partial t} + U_n|_{z=\eta} \cdot \nabla(\eta) + P - E \tag{19}$$

This changes the ocean salinity due to the adjustment to ocean volume and the subsequent effect on dilution or concentration.

**3 System components - UKA2 atmosphere, UKL2 land, UKO2 ocean and UKW2 wave models**

Table 2 provides a summary of the components of the UKC2 coupled prediction system. The uncoupled single-component atmosphere, land surface, ocean and wave configurations are referred to as UKA2, UKL2, UKO2 and UKW2 respectively. These are defined with identical domain, grid and physics options as when coupled in UKC2, but are capable of running in uncoupled mode by use of appropriate forcing and initialisation inputs as described below.

A distinction is also made between configurations of UKC2 in which only atmosphere and ocean components are coupled, referred to as UKC2ao, in which only ocean and wave components are coupled, UKC2ow, and the fully coupled atmosphere-land-ocean-wave system referred to as UKC2aow.

**3.1 The UKA2 atmosphere component**

The atmospheric component within UKC2 uses the MetUM code at version 10.1 (e.g. Walters et al., 2016). This uses the

"ENDGame" dynamical core (Even Newer Dynamics for General Atmospheric Modelling of the Environment; Wood et al.,



2014). As described by Walters et al. (2016), the prognostic fields are three-dimensional wind components, virtual dry potential temperature, Exner pressure, dry density, mass mixing ratio of water vapour and cloud fields. These are discretised horizontally onto a regular grid with Arakawa C-grid staggering (Arakawa and Lamb, 1977) and a Charney-Phillips vertical staggering (Charney and Phillips, 1953) using terrain-following hybrid height coordinates. The discretised equations are

solved using a nested iterative approach centred about solving a linear Helmholtz equation. The boundary layer scheme is a first-order turbulence closure mixing adiabatically conserved heat and moisture variables, momentum and tracers as described by Lock et al. (2000) and Brown et al. (2008). The UKA2 atmosphere configuration mirrors an implementation of that used in the UKV variable resolution atmosphere-land weather forecast system (Tang et al., 2013), defined at Parallel Suite 38 (PS38). Required MetUM code changes for development of the UKC2 configuration, implemented as branches to

the vn10.1 trunk code, are described in Appendix C.1.

### 3.1.1 UKA2 model grid

The UKV PS38 configuration has been in operational use at the Met Office since November 2016 and represents a substantial increase in the extent of the domain over the previous operational configuration described by Tang et al. (2013). A variable resolution grid methodology is applied with square grid cells defined within an inner region over UK and Ireland

having horizontal resolution of 0.0135° (approximately 1.5 km at mid-latitudes) across 622 cells across west-east and 810 across north-south coordinates (Fig. 1). Beyond the inner region, the model grid expands over a thin transition zone of width 18 grid cells to an outer region with 0.0135° by 0.036° resolution (approximately 1.5 km by 4 km) and 0.036° by 0.036° square grid cells in the domain corners. This gives a variable resolution model grid with 950 cells across the west-east and 1025 cells in the north-south coordinate. An increase of order 95% in the geographical coverage of the UKA2 domain

relative to the previous UKV implementation described by Tang et al. (2013) is achieved relatively efficiently by only expanding the domain in the coarser resolution outer region, thereby requiring only 41% more grid cells.

To maintain consistency with the operational weather forecast system, a different land/sea mask definition to the ocean is used, even in the inner region where the two grids are identical. Grid cells are defined as either entirely land or sea, in contrast to configurations with coupling to a global MetUM atmosphere, in which "coastal tiling" allows grid cells around

the coast to have a fraction of sea and land defined (e.g. Williams et al., 2015),

The same set of 70 vertical coordinates as used in the operational UKV implementation is used, with a terrain-following coordinate near the surface evolving to a constant height at 40 km above sea-level at the model top. The vertical coordinate focuses resolution nearest to the surface, with 16 levels defined in the lowest 1 km. The lowest model level for density is set at 2.5 m above the surface. Details of the vertical level set are included in the Supplementary Material to this paper.

### 3.1.2 UKA2 initialisation and forcing

The atmosphere component is initialised by first re-running a forecast-only global configuration of the MetUM at N768 (approximately 17 km at mid-latitudes) resolution (Walters et al., 2016). This global run is initialised from archived analyses



of the operational global MetUM forecast run with data assimilation at the time of a given case study. Boundary conditions for the UKA2 regional domain are then extracted using the MetUM makeBC utility (Whitehouse, 2014) and initial conditions are specified by interpolation from the global model dump file using the MetUM reconfiguration utility (Mancell, 2014).

### 3.1.3 UKA2 sea surface boundary conditions

Of particular relevance to its application within an environmental prediction system with which to study air-sea interactions, is consideration of the initialisation and evolution of the sea surface temperature (SST). By default in the operational UKV weather forecast system, SST are initialised from the daily OSTIA (Operational Sea Surface Temperature and Sea Ice Analysis; Donlon et al., 2012) interpolated onto the atmosphere model grid. This field, defined globally at a resolution of 1/20° (approximately 6 km), is then persisted (i.e. kept constant) as the lower boundary condition temperature over sea points in the domain throughout a forecast simulation, currently of typically 48 hours in duration in operations.

To support research using the coupled UKC2 and uncoupled UKA2 configurations, two options for initialising SST were implemented. For simplicity, the default configuration follows the approach described in Sect. 3.1.2, whereby the SST used to initialise the global N768 run is interpolated onto the UKA2 grid via the MetUM reconfiguration package. This field is based on OSTIA, but has first been interpolated onto the relatively coarse 17 km resolution global atmosphere model grid (e.g. Fig. 3(a)). Case study simulations run in this "global persisted SST" mode, with SST rooted in a global-scale observational analysis, are referred to as UKA2g.

For a more direct comparison with the coupled simulation, the SST field used in the initialisation of a corresponding uncoupled and free-running UKO2 ocean model simulation (see Sect. 3.4) can be used as a more directly relevant persisted SST control simulation for comparison with coupled simulations (e.g. Fig. 3(b)). Case study simulations run in this "high-resolution persisted SST" mode are referred to as UKA2h.

Following this approach it is also possible to define sensitivity tests with user-modified initial SST ancillary fields in order to assess the impact of SST biases or variations on the atmospheric evolution. In the absence of any updated ancillary information, the initial SST defined for a case study persists throughout a simulation cycle and across any successive model run cycles given that they are initialised from the restart dump of a previous cycle.

As described in Sect. 2.3, the lower boundary in UKA2 is also typically assumed to be at rest (i.e. ocean surface currents initialised to zero). The effect of roughness from surface waves is specified in terms of a constant Charnock parameter everywhere, set to the default operational UKV value of 0.011. In contrast in the UKC2 coupled prediction system, SST and the zonal and meridional surface current components (e.g. Fig. 4(a)) are updated each coupling period by the latest simulated fields from the UKO2 ocean model and a spatially and temporally varying Charnock parameter is computed and exchanged from the UKW2 wave model (e.g. Fig. 4(b)).



### 3.2 The UKL2 land surface component

The JULES land surface model (Best et al., 2011; Clark et al., 2011) is implicitly coupled to the MetUM atmosphere in all configurations to provide exchanges of momentum, heat and water between the surface and atmospheric boundary layer. The JULES system can also be run in standalone mode without linking to the MetUM, given suitable external driving data. This

provides a powerful tool for efficient offline testing and evaluation. JULES version 4.2 was implemented in UKC2, running on the same variable resolution grid as defined for the atmosphere (Sect. 3.1.1). The JULES science parameters were also set according to the UKV PS38 physics definition (see Supplementary Material).

Heat and water exchange processes in the sub-surface are represented on 4 soil layers at fixed layer thicknesses from the top down of 0.1 m, 0.25 m, 0.65 m and 2.0 m. The soil hydraulic conductivity is calculated according to the method of Brooks-

10 Corey following Cosby et al. (1984).

To support environmental prediction development, the hydrological functionality of the JULES trunk code has been extended to include a river routing scheme (see Sect. 2.3) to compute the freshwater fluxes from the land into the ocean (e.g. Dadson et al., 2011). This provides a foundation towards more integrated and consistent treatment of both land surface feedbacks to the atmosphere and the terrestrial hydrology, in common with the evolution of other land surface prediction

systems (e.g. Senatore et al., 2015). The JULES framework provides the necessary flexibility for further research and improvement on this approach, for example to improve the definition of variable depth soil layers, representation of lateral flows, and introducing more robust representation of groundwater processes (e.g. Clark et al, 2015; Davison et al., 2016).

Further details of the code modifications implemented as branches to the JULES trunk code for UKC2 are provided in Appendix C2.

### 3.2.1 UKL2 surface exchange

Proportions of 9 surface tiles are defined for each grid cell to represent sub-grid heterogeneity, with the surface of each land point subdivided into five types of vegetation (broadleaf trees, needle-leaved trees, temperate $C_3$ grass, tropical $C_4$ grass and shrubs) and four non-vegetated surface types (urban areas, inland water, bare soil and land ice), using information from the Centre for Ecology & Hydrology Land Cover Map 2007 (CEH, 2007). The urban scheme described by Best (2005) was

25 implemented. Surface fluxes are calculated separately on each tile within JULES using surface similarity theory, as introduced in Sect. 2.3.

While the initial boundary layer and surface exchange configuration implemented in UKC2 is described here, which follows closely the approach used in the operational UKV weather forecast configuration, the UKC2 regional coupled prediction system now provides a testbed for further exploring the surface flux parameterisations and assumptions adopted within both

the atmosphere and ocean components, and of the impact of an evolving wave surface at their interface.



### 3.2.2 UKL2 runoff generation

The surface and sub-surface hydrology used within UKC2 is based on extensive testing of potential options through offline evaluation and improvement of calculated river discharges for 13 selected catchments across the UK from a 10-year long JULES simulation driven by the CHESS meteorological data (Climate, Hydrology and Ecology research Support System; Robinson et al., 2017). An optimised value for the soil depth considered in the PDM scheme, $z_{pdm}$ (Sect 2.3.3), of 1.0 m was chosen, in contrast to the value used in the operational UKV configuration of 0.5 m. A number of JULES tests were also conducted assessing the impact of runoff and river discharge to the PDM b-parameter, with a value of 2 selected for UKC2, in contrast to the operational UKV value of 0.4. This implies that a relatively larger saturated fraction is calculated for a given soil water store. The value of parameter $S_0$ was also explored, with regard to the fraction $S_0/S_{max}$. A value of 0.5 indicates that no surface runoff is produced until the soil is 50% saturated. When $S_0/S_{max}$ is set to 0.0, as used by default in JULES, every rainfall event will produce saturation excess runoff. In order to develop spatially varying parameter sets, a new dependency of $S_0/S_{max}$ with local terrain slope in each model grid cell was applied in UKC2. A variety of potential linear and discontinuous functions were tested to define this dependency, and it was concluded that the best representation was found using:

$$\frac{S_0}{S_{max}} = max\left[1 - \left(\frac{slope}{slope_{max}}\right), 0.0\right] \tag{20}$$

A value for the maximum slope, $S_{max}$, of 6° was chosen, such that for grid cells with mean slopes in excess of 6° all rainfall generates saturation excess runoff while for flat terrain $S_0/S_{max}$ tends to 1 and no saturation excess runoff is produced unless the soil column is 100% saturated. This adaptation to the PDM scheme tends to reduce saturation excess surface runoff generation, relative to the default configuration, particularly over flatter terrain.

### 3.2.3 UKL2 river routing

The RFM river routing scheme (Sect 2.3.3) requires ancillary information on the river pathways and their connectivity across the model grid. This has been defined for a regular 1.5 km resolution grid across the UKC2 domain using the GMTED2010 digital terrain model and expert human intervention to ensure that the flows are routed correctly (e.g. Davies and Bell, 2009). For simplicity in initial testing, the river routing was considered on the same grid as the land surface model (i.e. the same as the variable resolution atmosphere grid), and so flows were only computed within the regular resolution inner domain across UK and Ireland. Extensions to the grid remapping within JULES will enable more flexibility to interpolate runoff calculated on the variable resolution land surface grid to a regular high resolution routing grid across the whole domain in future implementations.

Based on the river routing parameter values for high resolution models recommended in Bell et al. (2007) and experience working with the operational implementation of RFM in the Grid-to-Grid national flood forecasting system (e.g. Bell et al., 2009), the parameters listed in Table C.3 were applied as a baseline configuration in UKL2. Sensitivity tests were conducted modifying the surface wave speed for river cells $c_r$. For numerical stability conditions with a model grid cell spacing $\Delta x$ and



time step $\Delta t$, the wave speed must comply with $c \le \Delta x / \Delta t$. It was concluded that the sensitivity of simulated river flows to routing parameters was low relative to the sensitivity of results to the PDM parameters chosen. Further work and more extensive calibration, for example to establish spatially varying or flow-dependent parameter sets, may be necessary as a future development, but in general the implementation of the first-guess parameters provided in Table C.3 everywhere was

considered to be adequate to generate initial river flows within UKC2.

### 3.2.4 UKL2 surface and sub-surface moisture initialisation

Soil moisture is initialised in UKL2 by interpolation of soil moisture fields provided by a global soil moisture analysis in the MetUM reconfiguration process (Sect. 3.1.2). The interpolation attempts to preserve the level of saturation by taking into account changes in soil properties as defined by the higher resolution ancillaries, but it cannot take account of changes in the

precipitation associated with higher resolution, for example due to more detailed representation of orography, relative to the global configuration. This is likely to cause imbalances between the high resolution soil moisture and the model's climate. Work is ongoing at the Met Office to develop an operational UKV surface analysis, which should lead to future improvements in the initialisation of soil moisture in both operational and research-mode systems.

The new requirement to simulate river flows in UKC2 has also led to further land surface initialisation challenges.

Information on four prognostic variables are required for initialising RFM, namely the surface and sub-surface inflows $r$ at each grid cell and the surface and sub-surface storages, $S$ (see Appendix B). These variables are not readily available within operational archives, and so in order to initialise RFM with a realistic surface and sub-surface state for a given case study starting point, it was necessary to first run a UKL2 configuration of JULES in offline mode from empty for several (nominally at least 3) months, driven by meteorological forcing from the archived operational UKV weather forecast model.

Required driving variables include air pressure, specific humidity, air temperature, precipitation, and radiation and wind components.

### 3.3 The UKO2 ocean component

UKO2 represents the research use of a new mesoscale eddy-resolving coastal ocean model configuration for the north-west shelf region. In contrast to the current operational AMM7 configuration (Fig. 1; O'Dea et al., 2017), which runs on a

latitude-longitude grid at a horizontal resolution of approximately 7 km, the UKO2 ocean component uses a uniform 1.5 km resolution grid. As illustrated in Fig. 4(a), this step-change improvement in horizontal resolution enables smaller-scale processes such as internal tides, which are known to play a key role in both shelf-break exchange and on-shelf circulation to be resolved. The 1.5 km horizontal resolution is sufficient for resolving the internal Rossby radius on the shelf of order 4 km (Holt and Proctor, 2008; Holt et al., 2017), and it is known that mesoscale eddies play a crucial role in transporting heat,

freshwater and nutrients in the region (Palmer et al., 2015).



The UKO2 ocean configuration mirrors the Atlantic Margin Model (AMM15) ocean-only shelf-seas forecasting system (Graham et al., 2017), which is being developed and further tested towards future operational implementation as part of the evolution of the EU Copernicus North West Shelf Marine Service ([marine.copernicus.eu](marine.copernicus.eu)).

The UKO2 ocean component is a regional implementation of NEMO (Madec et al., 2016) at version 3.6_stable (trunk
revision 5518). Model physics options used mirror those defined in the AMM15 configuration namelists (see Supplementary Material), and further details are provided by Graham et al. (2017). Given the km-scale resolution, only a minimal amount of eddy viscosity is applied in the lateral diffusion scheme, to ensure model stability. For momentum and tracers, bilaplacian viscosities are applied on model levels (using coefficients of $6{\times}10^7$ m$^4$s$^{-1}$ and $1{\times}10^5$ m$^4$s$^{-1}$ respectively). The Generic Length Scale scheme is used to calculate turbulent viscosities and diffusivities (Umlauf and Burchard, 2003) and surface wave
mixing is parameterised using the Craig and Banner (1994) scheme. Dissipation under stable stratification is limited using the Galperin limit of 0.267 (Galperin et al., 1988) and bottom friction is controlled through a log layer with a non-linear drag coefficient of 0.0025. A series of compilation keys, described in Table C.5, are applied on building the NEMO executable for UKO2 and UKC2.

As described in Sect. 2.3, the treatment of surface solar radiation is controlled by an RGB light penetration scheme, in which
the fraction of shortwave solar radiation that is absorbed in the upper few centimetres rather than penetrate to depth is controlled by namelist parameter *rn_abs*. After some testing, a fraction of 0.66 (i.e. 66% absorption) is selected and input radiation is partitioned into solar $Q_{sr}$ and non-solar $Q_{ns}$ component fluxes.

A summary of the NEMO code changes, merged and compiled as discrete branches, required as adaptations for UKC2 is described in more detail in Appendix C3.

**3.3.1 The UKO2 model grid**

The ocean component is defined on a regular 1.5 x 1.5 km grid in rotated coordinates across the entire domain (Fig. 1), with the central region exactly matching the inner domain of the UKA2 atmosphere grid. This requires 1458 grid cells in the west-east zonal direction and 1345 grid cells in the north-south meridional direction, with Arakawa C-grid staggering (Arakawa and Lamb, 1977). The model grid has 51 vertical levels and a non-linear free surface. The vertical grid uses a
stretched terrain following "S-coordinate" system as described by Siddorn and Furner (2013), which masks vertical cells over steep slopes where the gradient of the bathymetry exceeds a specified parameter (*rmax*) and ensures a minimum depth of the surface layer. A minimum ocean depth of 10 m is imposed, with no wetting and drying at the coastal grid points.

The extent of the UKC2 domain was carefully assessed with a view to the implementation of the ocean component. To the south (lower-left grid cell centre located at 17.617° W, 44.065° N), the extent was chosen so that the domain boundary was
sufficiently far north of the Spanish coast such that the shelf-break transport flows into the domain perpendicularly (Fig. 1). The northern boundary (upper-right grid cell centre located at 16.254° E, 62.206° N) is set sufficiently far north of the Faroe Isles to allow transport around the islands, but far enough south to avoid partially representing overflows or transport around Iceland.



The bathymetry defined is based on EMODnet (EMODnet Portal, Sep 2015 release). An adjustment had to be applied to the EMODNET bathymetry to convert the reference depth from lowest astronomical tide to mean sea level, as required for NEMO. This process used an estimate of the lowest astronomical tide from a 19-year simulation of the CS3X tidal model (Batstone et al., 2013).

The EMODnet bathymetry data includes a land sea mask based on Open Street Map, which has been interpolated onto the UKO2 grid. For grid cells of partial land/sea, they were originally set as land if the EMODnet land covered more than half of the target grid cell. The resulting mask was then also manually assessed to check the representation of narrow channels, estuaries and small islands.

### 3.3.2 UKO2 initialisation and forcing

For case study simulations based on 2014 dates presented in Sect 5, daily boundary data of sea surface height, 2-d currents and 3-d temperature and salinity are provided from the archived ¼° resolution ocean data from the GloSea5 operational global seasonal forecast system (MacLachlan et al., 2015), and initial conditions provided from a 1-year run of the AMM15 model initialised on 1 January 2014 from GloSea5 with meteorological forcing from the ERA-Interim reanalysis (Dee et al., 2011). For case study simulations based on 2015 dates in Sect 5, boundary data are provided from the archived 12 km

resolution NATL12 operational ocean model configuration (e.g. Siddorn et al., 2016), and the initial conditions are taken from a 1-year run of the UKO2 configuration initialised from the 2014 AMM15 hindcast on 1 January 2015. For initial development, a climatological river discharge data are applied as freshwater forcing (Graham et al., 2017). The impact of using the freshwater fluxes from UKL2 on the ocean component will be assessed in future work.

Simulations are conducted using the direct forcing approach, whereby the heat fluxes computed by an atmosphere model are

applied, rather than being computed by NEMO based on bulk input properties. The *key_shelf* compilation key is also used, which implies that wind forcing is provided in the form of the $U$ and $V$ wind components rather than the surface stress components directly, and a surface layer parameterisation applied to translate to the stress forcing at the surface.

By default, the UKO2 configuration is forced with 3-hourly radiation and hourly wind and mean sea level pressure forcing data taken from archived operational global MetUM forecast output, at a horizontal resolution of approximately 17 km. Case

study simulations run with this "global forcing" mode are referred to as UKO2g.

In order to provide more direct comparison to the coupled configuration, a configuration of UKO2 forced with the km-scale atmospheric data at hourly temporal resolution from the UKA2 simulation was also run, using the same interpolation weights generated for coupling to translate data between grids. Case study simulations run in this "high resolution forcing" mode are referred to as UKO2h. Comparison of ocean-only results between UKO2g and UKO2h enable assessment of any benefits of

the availability of higher resolution meteorological information for operational ocean prediction, although it should be noted that the UKO2g meteorological forcing is taken from an assimilative operational forecast system, including assimilation of global scatterometer winds at the ocean surface, whereas the UKA2 meteorology comes from a free-running case study



mode simulation. The operational implementation of the UKV weather forecast system on the UKA2 domain in the Met Office since November 2016 will provide the potential for more rigorous investigation of these issues in future studies.

### 3.4 The UKW2 surface wave component

The UKW2 surface wave component within UKC2 uses the NOAA/NCEP community third generation spectral wave model

WAVEWATCH III (Tolman, 2014) at version 4.18. The governing equations of WAVEWATCH III include refraction and straining of the wave field due to variations of the mean water depth and currents. Various wave parameterisation schemes for the source terms are available, including both "ST3" (WAM Cycle-4; Komen et al., 1994) and "ST4" (Ardhuin et al., 2010) packages. Source term physics in UKC2 use the ST3 approach as default, following the tuning described by Bidlot (2012) to establish some consistency with existing work on atmosphere-wave coupling (e.g. Janssen, 1991; Breivik et al.,

2015). Note this choice is in contrast to the ST4 approach more typically used by operational centres running WAVEWATCH III for global wave model simulations. Modelled source terms include wave growth and decay due to winds, nonlinear resonant interactions, dissipation, bottom friction, depth-induced breaking and scattering due to wave-bottom interactions. Nonlinear wave-wave 'quadruplet interactions', which shift wave energy toward lower frequencies, are parameterised using the Discrete Interaction Approximation (Hasselmann et al., 1985). Wave energy propagation uses a

second order upstream non-oscillatory scheme (Li, 2008) with 'Garden Sprinkler Effect' alleviation following the averaging scheme proposed by Tolman (2002). A series of compilation switches, described in Table C.7, are applied on building the WAVEWATCH III executable for UKW2 and UKC2.

A summary of the code changes applied in the UKW2 configuration is described in Appendix C4.

### 3.4.1 The UKW2 model grid

For simplicity in initial implementation and testing, the UKW2 grid was defined on an identical grid and with identical bathymetry to the UKO2 ocean configuration (see Sect. 3.3). Note that for operational use, a more efficient spherical multiple-cell (SMC; Li and Saulter, 2014) approach, using variable resolution wave grids with increased resolution nearer the coastlines is under trial. Investigation of generating remap weights between a variable resolution atmosphere grid, regular ocean grid and unstructured wave grid is planned for future implementation in a UKC3 system, to reduce the computational

cost of running a high resolution wave component.

### 3.4.2 UKW2 initialisation and forcing

As surface waves grow to maturity quickly, the wave state spins up from rest relatively quickly and typically within a 5 day simulation time. Case study simulations were initialised from a restart file generated by running the UKW2 configuration from rest for the 5 day period prior to the case study initial time. These spin-up simulations used hourly wind forcing

generated from the operational global MetUM archive at approximately 17 km resolution (as used in forcing UKO2g



simulations). Spectral boundary conditions were provided from archived operational global wave model output, for which the WAVEWATCH III model resolution in open waters of the Atlantic was set at approximately 25 km.

For wave-only case study simulations, UKW2 can continue to be run in this mode for the period of interest. This "global wind forced" approach is termed UKW2g. As with the hourly high-resolution meteorology forcing for UKO2, it is also possible to interpolate wind speed components from the UKA2 grid to the UKW2 grid to produce high resolution forcing, in uncoupled wave-only simulations termed UKW2h. Forced wave-only simulations additionally including ocean current information read from file are termed UKW2c, with surface currents taken from UKO2h case study output. Finally, forced wave-only simulations termed UKW2l have also been run with wind, current and water level forcing, with the water levels also taken from the same UKO2h case study NEMO output.

## 4 Developing an evaluation framework

In order to explore, understand and demonstrate the skill and limitations of more integrated systems for UK environmental prediction, and inform future development priorities, a robust and traceable evaluation framework for coupled predictions, relative to uncoupled approaches was designed and implemented. Figure 5 provides a summary of the UKC2 evaluation system, and the interdependencies between the various control simulations. All coupled and uncoupled configurations were defined and run for a given case study period as rose suites (http://metomi.github.io/rose/doc/rose.html) and version controlled under the Flexible Configuration Management (FCM) system (http://metomi.github.io/fcm/doc/). The suite framework provides the flexibility to run with different science and coupling options with a common approach, with relatively minor and traceable namelist changes invoked between different suites within the evaluation system. The suite design also provides common build libraries and configurations, despite components originating from different modelling systems with their own underpinning working practices.

Table 2 summarises the nomenclature introduced in Sect. 3 to define the evaluation framework. Required changes for initialisation or forcing of uncoupled components are implemented as branches of a suite configuration. A number of comparisons can then be explored to highlight sensitivities to initial conditions, changes in forcing or coupling as described in Table 3.



## 4.1 Case study evaluations

A selection of 5-day duration case study periods are discussed in the remainder of this paper, as an illustration of the performance of the UKC2 configurations and the potential benefit of coupling relative to the control simulations. These cover a range of seasons and environmental conditions, including a warm summer storm (7-11 August 2014) and more severe autumn and winter storm cases (2-6 October 2014, 7-11 December 2014). More stable and generally dry autumnal (6-10 September 2014), winter (7-11 February 2015) and hot summer (30 June – 4 July 2015) cases are also presented.

It is noted that the use of case studies over a few days is a more routine approach for assessing atmosphere model performance than ocean, wave or land models, given the relatively faster evolving processes in the atmosphere. This is however considered a suitable starting point for evaluating the impact of coupling on short time scales, with further work planned to assess the impact of coupling on the ocean over longer timescales through longer integrations.

The initial assessment of the UKC2 is conducted in terms of bulk properties of the atmosphere, ocean and wave state, namely the air temperature at 1.5 m, wind speed at 10 m, sea surface temperature, sea surface height, significant wave height and wave peak period. While consideration of these variables provides a somewhat crude headline indication of model performance within the scope of this paper, it should be noted that this does not represent a sufficient or definitive evaluation of the system. In order to more fully evaluate the extent to which UKC2 represents the coupling feedbacks, a more thorough analysis of the time-varying characteristics of these variables and in particular of the simulated surface fluxes and profile information in the near surface ocean and atmosphere is required, but is beyond the scope of this introductory paper. For example, Fallman et al. (2017) provide a more detailed evaluation of stratiform cloud development over the North Sea in the UKC2ao configuration, relative to UKA2h. The study highlights how diurnal sea surface warming in the coupled simulations lead to increased shallow convection, leading to modified boundary layer evolution and formation of low level clouds.

The following discussion focuses on analysis relative to the UKA2h, UKO2h and UKW2h configurations as the reference control, since these are initialised from identical initial conditions and have more directly comparable forcing to the coupled simulations. As highlighted in Sect. 2, comparisons between UKA2h with UKA2g (and between the equivalent ocean or wave configurations) highlight the combined impact of resolution and assimilation within the initial conditions and/or forcing on that reference system performance. Discussion of the impact of coupling on land surface variables, and of the performance of the UKL2 river flow computations is also omitted to focus the scope of evaluation presented here on the impact of atmosphere-ocean interactions.

## 4.2 Observations

Model outputs are compared with a variety of *in-situ* observations taken from Met Office operational archives and routinely exchanged over the World Meteorological Organization Global Telecommunication System (GTS). Observations of atmospheric variables over land are taken from the network of surface automatic weather stations operated by the Met Office and other national weather services across Europe. In open waters of the northwest European shelf seas and Atlantic ocean,



observations of atmospheric variables, sea surface temperature and wave conditions are provided from drifting buoys, moored buoys, ships and offshore oil installations. Closer to the UK coastline, key sources of ocean and wave state observations are provided by the WaveNet monitoring network for the UK, operated by the Centre for Environment, Fisheries and Aquaculture Science (Cefas; wavenet.cefas.co.uk) and the Channel Coast Observatory's coastal buoys

(www.channelcoast.org). It is therefore worth noting that the majority of observation sites over the ocean considered for evaluation here are located in coastal regions, where it is known that model skill may be poorer than over open waters due to more limited fetch, increased importance of local geography on wind and current flows, and the imposition of minimum depth limitations in ocean and wave models.

## 5 Performance of UKC2 and the impact of coupling

Tables 4-9 provide summary root-mean square error (RMSE), bias and linear regression correlation coefficient $r^2$ statistics for each case study for the atmosphere-land-ocean-wave UKC2aow and atmosphere-land-ocean UKC2ao coupled systems and for the relevant uncoupled UK[AOW]h and UK[AOW]g atmosphere, ocean or wave model simulations forced with high resolution and global resolution data respectively. The values given are an average of statistics computed separately at each observation site, weighted by the number of matched observation and model data points contributing to each statistic. Using

the UK[AOW]h statistics as the reference, average metrics for which a statistically significant difference between results is found using the Student's *t*-test at the 95% level are underlined, and also highlighted in bold where those differences indicate an improvement relative to the reference.

These results highlight that all configurations provide at least representative simulations of atmosphere, ocean and wave states. UKC2aow and UKC2ao therefore represent a successful initial development of regional coupled prediction systems at

20 high resolution, which is non-trivial given the technical and scientific complexities involved in bringing together disparate model systems initially developed within an uncoupled context.

## 5.1 Atmosphere model variables

An overview of the performance of the atmospheric component of the fully coupled atmosphere-land-ocean-wave UKC2aow system relative to the UKA2h control simulation for each of the case study periods considered is presented for surface

temperature (Fig. 6), 1.5 m air temperature (Fig. 7) and 10 m wind speed (Fig. 8). While UKC2aow includes dynamic SST, surface currents and Charnock parameter as surface boundary conditions over the ocean, in UKA2h the initial SST (identical to the initial SST in UKC2aow) remains constant, surface currents are assumed zero and a default value for the Charnock parameter of 0.011 is applied everywhere for the duration of a case study.

Figures 6-8 show the relative root-mean square error (RMSE) statistic computed between modelled and observed data over

30 the 5-day duration of each case study for UKC2aow and UKA2h at each observation site as:



$$RMSE_{rel} = \frac{RMSE_{UKC2} - RMSE_{UKA2}}{RMSE_{UKA2}} = \left( \sqrt{\frac{\sum(obs-UKC2)^2}{n}} \middle/ \sqrt{\frac{\sum(obs-UKA2)^2}{n}} \right) - 1 \qquad (21)$$

Observation points shown in green ($RMSE_{rel} < 0$) are at locations where the RMSE for UKC2aow shows smaller errors relative to observations than UKA2h, while those in purple ($RMSE_{rel} > 0$) indicate locations where in UKC2aow exceed those for UKA2h.

### 5.1.1 Surface temperature

The comparison of MetUM atmosphere model surface temperature over the ocean in Fig. 6 indicates the relative performance of the dynamic NEMO ocean component within UKC2aow (and UKC2ao, not shown) relative to the persisted SST case. It is therefore encouraging that, while not universal, for most sites during most case studies the RMSE in SST is reduced, by typically more than 25%, through coupling relative to UKA2h. There are clusters of sites for which the persisted SST compares better with observations during each of the case studies however. These merit further investigation, noting in particular sites in northern Scotland and along the east England coast during August 2014, and locations along the south-west England coast during September 2014 and February 2015 cases. The summary statistics for SST between the coupled and persisted atmosphere-only simulations shown in Tables 4a)-9a) highlight a significant improvement in overall RMSE and bias for all cases other than September 2014 and February 2015, when improvements were also found. This indicates model skill in the UKO2h configuration at simulating the diurnal and longer timescale variability in SST, and of the potential value of using a dynamically evolving SST for improving the surface boundary condition within atmospheric models relative to persistence.

The comparison between average statistics for the persisted SST between UKA2h (persisting the UKO2 initial SST based on NEMO model simulation) and UKA2g (persisting a coarse resolution OSTIA SST based on observations) in Tables 4a)-9a) do not show any statistically significant differences. These results provide some confidence that the UKO2 configuration within UKC2 is providing robust ocean predictions and case study initialisation.

### 5.1.2 Air temperature

It might be expected that improvement (or degradation) to the SST should lead to improvement (or degradation) in the simulation of near-surface air temperature, as diagnosed at 1.5 m above the surface. Figure 7 illustrates the overall impact of coupling on the comparison of air temperature with observations for each case study. Comparing Fig. 6 and Fig. 7 highlights that there is not a direct relationship between surface and air temperature, with air temperatures also being strongly driven by non-local factors such as advection and cloud cover. This is apparent in Tables 4a)-9a) with relatively similar $r^2$ values for air temperature across all model configurations despite $r^2$ for SST being zero for the persisted surface temperature in UKA2 configuration runs. Figure 7 however demonstrates evidence of improvement in 1.5 m air temperature through coupling, most strongly over ocean areas. Air temperatures over the Celtic Sea to the south-west of the UK and southern North Sea are particularly improved during August, September, October 2014 and June/July 2015 case studies. Similar results are seen for





the North Sea region east of Scotland apart from the June/July 2015 period, for which there is a degradation in air temperature through coupling relative to observation.

Despite an overall improvement in the simulation of the surface temperature during the severe winter storm December 2014 case, simulated air temperature is also degraded in UKC2aow relative to UKAh (or UKA2g) above the North Sea more generally, with increases in RMSE in excess of 25% at some locations. Similar results are also found for UKC2ao, indicating that the increased errors in this region can be attributed air-sea coupling rather than wave coupling. There is an observed decrease in SST in the region by about 0.5 K over the 5-day duration of the December 2014 case study, which is well captured in the UKC2 and UKO2h ocean simulations. The coupled sea surface temperature therefore tends to be of order 0.25-0.5 K cooler than the uncoupled simulations towards the end of the simulation. This appears to exacerbate a general cool bias in modelled air temperatures over the ocean for this case, indicating that while UKA2h and UKA2g may verify better in the region for this case, the model system is not well representing the physical processes which took place in reality. While of smaller magnitude to changes over the sea, there are some notable differences in air temperature results over land, likely linked to differences due to coupling in the evolution of weather systems and local features such as sea breeze development. For example, air temperatures over Scotland are shown to be particularly improved during the hot July 2015 case (Fig. 7f). According to the summary statistics in Tables 4b)-9b) RMSE statistics are significantly improved due to coupling when considering all observation sites and ocean-only sites for August, September and October 2014 and for the June/July 2015 case. Very similar conclusions can be drawn considering statistics for all or ocean-only sites. As anticipated, there is relatively little difference for surface or air temperature between results for UKC2aow and UKC2ao simulations with and without wave coupling.

### 5.1.3 Wind speed

Figure 8 shows summary differences between UKC2aow and UKA2h near-surface wind speed simulations relative to observations. The magnitude of changes between models is generally smaller than for temperature differences, and areas with on average slightly improved near surface wind speed can also be found within the domain during each case study. A notable exception is the region of degraded wind speed in UKC2aow over the North Sea (interestingly, coinciding with a region of improved surface and air temperatures) during the August 2014 case study. This region also shows degradation by a smaller magnitude in September 2014 and June/July 2015 cases. In this region, all configurations underestimate the wind speed, most significantly during a period of increased winds during 9 August 2014 when peak observed wind speeds are up to 20 ms$^{-1}$ while modelled speeds are typically only 10 to 15 ms$^{-1}$. In this case, the impact of wave coupling leads to a reduction in wind speed due to increased roughness and hence even poorer agreement compared with observations. Further analysis of this case would be of value.

Summary statistics shown in Tables 4-9 illustrate very similar wind speed results across all model configurations, suggesting that wind speed within the high resolution MetUM at least is not highly sensitive to the surface forcing. Consideration of longer-term climatologies would be helpful to begin to unpick whether this is representative of the strength of surface-wind





coupling in this region or represents a potential improvement required to the MetUM surface exchange scheme assumptions. While not strongly apparent from Fig. 8, Table 7 shows significant improvement on average in wind speed over the ocean due to the coupling with the WAVEWATCH III Charnock parameter during the strong winter storm December 2014 case, in general due to a reduction in simulated wind speed due to enhanced extraction of momentum at the surface by a growing

wave state. In this case, the MetUM wind speed across all configurations was biased high relative to observations such that wave coupling led to an overall improvement in system performance.

## 5.2 Ocean model variables

Figure 9 provides a comparison of the NEMO simulated sea surface temperature in UKC2aow and ocean-only UKO2h. With no wave-to-ocean coupling physics implemented in UKC2 (see Sect. 2.2), and the coupling and forcing information both

provided each hour, the key difference between these simulations is of the impact of the modified atmospheric forcing including ocean feedbacks in UKC2aow on the evolution of the ocean state. The impact of coupling is generally mixed, with areas of particularly strong improvement or degradation in simulation skill more limited in geographical extent than discussed in Sect 5.1. The summary statistics in Tables 4c)-9c) indicate statistically significant improvement in the SST RMSE during the September and October 2014 case studies, when particular improvement in SST along the English Channel

and Bay of Biscay coasts is apparent in Fig. 9. Relatively small differences are shown in Tables 4c)-9c) overall for other cases. This might be illustrated by the December 2014 results in Fig. 9 which show regions of particular improvement in SST along the eastern English Channel coast being offset by degradation of similar magnitude further west.

An alternative summary view of relative model performance is illustrated in Fig. 10, which for each case study plots the model configuration for which the lowest overall RMSE statistic at a site is achieved. This is also a somewhat crude

approach, given that differences need not be very large or statistically significant to register a configuration as having the lower RMSE value. It is however instructive for highlighting regions and case studies for which coupling may have a greater effect. The number of sites for which each statistic considered is best is also summarised for all variables in Tables 4c)-9c). This highlights improvement in ocean SST simulations due to coupling for all cases other than June/July 2015, and for coastal locations along the English Channel and southern North Sea in particular. Further north along the north-eastern coast

of England and in locations off the shelf edge to the west of Ireland, coupling to the atmosphere has a much smaller impact on results.

Tables 4c)-9c) also show a summary comparison between UKO2h and UKO2g SST results, where UKO2h is forced by meteorology from the free-running 1.5 km resolution UKA2h run and UKO2g forced by meteorology from the assimilative global 17 km resolution operational MetUM archive. Results are generally improved in UKO2h relative to UKO2g (i.e. an

improvement with increased resolution of forcing, despite the lack of meteorological assimilation) with a statistically significant difference in RMSE for August, September and October 2014 cases.

Figure 11 shows a similar presentation to Fig. 10 for sea surface height at a number of coastal buoy locations. This highlights a tendency for the global forced meteorology in UKO2g to provide best SSH estimates overall, likely due to the impact of



data assimilation in the ocean forcing. However, distinct regions where coupling between the waves and atmosphere in particular appears to provide benefit overall for all but the August 2014 case study. Differences in sea surface height between simulations are generally small (Tables 4c)-9c)), and so any improvements result from adjustments to the phasing of tides around the UK coastline. It is anticipated that more conclusive results will be drawn on running longer ocean simulations

with and without coupling, and after introducing wave-ocean coupling feedbacks within UKC3 (e.g. Staneva et al., 2016).

## 5.3 Wave model variables

A comparison of the wave only UKW2h (using UKA2h wind forcing) and UKW2g (using global resolution operational wind forcing) configurations is made with both the coupled UKC2aow system and a wave only simulation UKW2l which includes both surface current and water level forcing from UKO2h in addition to high resolution UKA2h wind forcing.

Figure 12 shows the distribution of coastal observation sites and the model configuration for which the lowest RMSE in signigicant wave height is achieved for each case study. These results, together with the summary statistics presented in Tables 4c)-9c) starkly show that the wave model performance is significantly improved in all cases when forced with global resolution rather than high resolution wind forcing.

The degradation in the quality of the wind forcing between UKW2h and UKW2g is in part attributed to the impact of data

assimilation in global operational systems, particularly of satellite-based scatterometer winds. Assessment of timeseries of the wind forcing also highlights much greater variability in the uncoupled or coupled high resolution winds than those in the global resolution forcing. In all cases, wind speed values are instantaneous hourly fields, implying that the coarser resolution implicitly smooths the input wind field at a point. Experiments to understand the impact of applying a temporal averaging or some other filtering to the high resolution input wind forcing to reduce this variability on WAVEWATCH III results would

be of value, but are beyond the scope of this paper. It will also be instructive to isolate the impact of data assimilation in the global forcing, and therefore potential improvements from data assimilation within the high resolution systems, by comparing results with a UKW2g configuration forced by a free-running global MetUM simulation. These considerations highlight the potential flexibility of the UKC2 evaluation framework for assessing the sensitivity of model predictions to the input boundary conditions as an approach to improving forecast skill, even before considering the impact of closer coupling

between components.

The smoother wind field derived from the global atmosphere model may not be generally detrimental to the skill of the wave model, since major signals in the wave time-series are governed by development of the waves over fetches of tens of kilometres or more. It should also be recognised that the standard sample window of 15-20 min used in the wave observations means that a stationary platform will measure wave energy travelling over distances of approximately 3-25 km

(for wave periods in range 3-20 sec), such that the observation will not only be influenced by local wind conditions.

Given the context that forcing by high resolution atmospheric winds tend to degrade the WAVEWATCH III model skill, that there are occasions and regions where coupling leads to improved performance is particularly noteworthy. Fig. 12 shows particular improvement during the strong winter storm case, in which observed significant wave heights exceeded 15 m. This



is reflected in Table 7c), where results for UKC2aow are comparable to UKW2g, while for all other cases results are overall poorer than for UKW2g.

In comparison to the reference UKW2h wave model simulation however, the impact of coupling to the ocean and representing the two-way feedback between waves and atmosphere results in significant improvements. Figure 13 summarises RMSE differences during each case. In accordance with Tables 4c) and 6c), summary results are generally degraded during the August and October 2014 cases. The August 2014 results can be attributed to the generally poor simulation of winds during the 9 August 2014 storm (Sect. 5.1.3).

Improvements in the remaining case studies are largely focussed along the English Channel coast, where strong currents are known to result in current-wave interaction. For the relatively calm February and June/July 2015 cases, for which significant wave heights of a few metres were observed, relative improvements are also apparent along the North Sea coast of eastern England and through the Irish Sea along the western coast of Wales and England. For the December 2014 winter storm case, improved significant wave heights are highlighted to the north of the Scotland. This coincides with the north-west approaches of the incoming storm, where increased surface roughness and decreased winds in UKC2aow led to a reduction in significant wave heights relative to all uncoupled wave-only UKW2 simulations.

Figure 14 presents a comparison of simulated peak wave period, which represents the period for the most energetic waves in the spectrum. While significant wave height results in UKC2aow are generally poorer than UKW2h for the August 2014 case, results for peak period show a number of sites with improved agreement with observations along the southern England coasts due to coupling in UKC2aow. Table 4c) indicates improved summary statistics relative to both UKW2h and UKW2g. The improvement is also strong, in excess of 25% in RMSE, during the September 2014 case. Table 5c) shows a significant improvement in RMSE and bias relative to either UKW2h or UKW2g for both UKC2aow and the forced UKW2l configurations. This highlights that the improvement in wave period is mostly driven by coupling to the ocean currents rather than by improvements to the meteorological forcing in this case. These results emphasise that assessment of the significant wave height provides an important but not complete representation of the performance of the wave model, with indications that coupling might also provide improved characterisation of the wave state (i.e. discrimination between wind and swell waves).

## 6 Discussion and ongoing development

This paper provides an introduction to the UKC2 high resolution atmosphere-land-ocean-wave coupled prediction system. Development and implementation of UKC2 and the associated uncoupled configurations (Table 2) within a traceable evaluation framework has set in place good foundations on which to develop improved understanding of coupled processes at high resolution for the UK and surrounding region.

Summary results presented from a number of case study simulations in contrasting atmospheric and sea state conditions at different times of year provide an initial indication of model performance. It can be concluded that the UKC2 system



provides robust and representative predictions across atmosphere, ocean and surface wave components. This assessment however only begins to scratch the surface of evaluating the UKC2 system and provides only a first order illustration of the potential improvements that might be gained from closer coupling between model components. More detailed evaluation of the case studies introduced here will be published as their analysis continues (e.g. Fallmann et al., 2017).

5 These results also provide an important check on the limitations of coupling for improving model skill. Coupling alone is not a panacea for correcting all environmental model errors. Rather, it provides new tools for understanding sensitivities to boundary layer processes in the atmosphere and ocean across the surface wave interface, and for improving their representation alongside other developments to component models. This may require revisiting a number of the assumptions and parameterisations embedded within the component models, which have typically been developed and tuned in an 10 uncoupled context.

This work highlights the shorter-term potential for improving operational predictions from atmosphere, ocean and wave models run in uncoupled mode, through making better use of the available information contained in the other components as more representative boundary conditions or forcing. More completely representing the various feedbacks between components within a fully coupled system remains an achievable but challenging goal, for which further detailed evaluation 15 and refinement is required.

The key focus for future work in the context of UK Environmental Prediction is on further applying the existing UKC2 system for new research, aiming to improve the assessment and understanding of coupled processes in the north-west European region. On a longer timescale the ambition, if proven, is to work towards developing a well tested, characterised and optimised coupled system for the UK. The vision is for a future UKCx capability which couples assimilating operational 20 model components, with sufficient flexibility to support a varied range of scientific research. Consideration of the suitability of such an environmental prediction capability for operational delivery of integrated natural hazard forecasts and warnings requires considerably more evidence on its potential skill and limitations. Demonstration of pathways towards operational implementation beyond 2020 will also require consideration of coupled and uncoupled approaches to assimilation and interfaces within the context of ensemble prediction systems representing uncertainties within components.

25 The potential range of research questions to be explored and the need to support a variety of users to apply the system emphasises a need for ongoing system development, with a focus on:

    i)    improving the functionality and flexibility of use of the coupled prediction system and related inputs, for example by adding wave-to-ocean coupling physics, and adding a marine biogeochemistry model component,

    ii)    continuing to support standard configurations of traceable model experiments with consistency between 30 uncoupled (e.g. UKA2), partially coupled (e.g. UKC2ow) and fully coupled modes (e.g. UKC2aow),

    iii)    supporting a growing community of researchers working across disciplines underpinned by shared tools and computing facilities.



Building on UKC2, there is a requirement for ongoing pull-through of related environmental prediction research developments and component model system improvements into a regular (e.g. annual) series of version-controlled UKCx system updates for community research use. A number of specific system developments are envisaged.

   i)      development of wave-to-ocean coupling physics, including shallow-water processes such as the effect of
bottom friction,

   ii)     transition from free-running to model components running (uncoupled) data assimilation,

   iii)    increased flexibility in coupling approach, including testing of coupling at sub-hourly frequency and greater
           independence in choice of frequency for different variables,

   iv)     scientific optimisation, improved configuration and parameterisation choices,

v)       more explicit representation of near-coastal and estuarine processes, including coupling with wetting and
           drying across the ocean and land interface,

   vi)     more routine initialisation strategies and ongoing development of related forcing and boundary input tools,

   vii)    technical optimisation, improved quantification and reduction of computational costs of running simulations.

The UKC2 development is focussed on short timescale processes and applications in the north-west European region. In line
with research systems such as COWAST (Warner et al., 2010), the potential for the MetUM-JULES-NEMO-WAVEWATCH III coupled systems to be applied at high resolution for other regions to support testing and development in different environments should be explored. Having tested their fidelity on short timescales, there is a strong potential for application of regional coupled predictions for developing more integrated scenarios of the environment under future climate and land/ocean use change.

By building on progress delivered through the development of UK environmental prediction and the establishment of the UKC2 system in particular, a new phase of developing the functionality and flexibility of regional coupled prediction tools is envisaged. This will enable a community to utilise these effectively and to further assess and improve the performance and value of a fully coupled prediction capability at high resolution for Earth System forecasting.

**Code availability** *Intellectual property*

Due to intellectual property right restrictions, neither the source code or documentation papers for the Met Office Unified Model or JULES can be provided directly. All model codes used within the UKC2 configuration are accessible to registered researchers, and links to the relevant code licences and registration pages are provided for each modelling system below. All code used can be made available to the Editor for review. Supplementary material to this paper does include a set of Fortran
namelists that define the atmosphere, land, ocean and wave configurations in UKC2 simulations.

*Obtaining the Met Office Unified Model*

The Met Office Unified Model is available for use under licence. A number of research organisations and national meteorological services use the Met Office Unified Model in collaboration with the Met Office to undertake basic atmospheric process research, produce forecasts, develop the Unified Model code and build and evaluate Earth system



models. For further information on how to apply for a licence see http://www.metoffice.gov.uk/research/collaboration/um-collaboration. The MetUM vn10.1 trunk code and associated modifications for UKC2 (Appendix C.1) are then available on the MetUM code repository.

*Obtaining JULES*

JULES is available under licence free of charge. For further information on how to gain permission to use JULES for research purposes see http://jules.jchmr.org. The JULES vn4.2 trunk code and associated modifications for UKC2 (Appendix C.2) are then freely available on the JULES code repository.

*Obtaining NEMO*

The model code for NEMO vn3.6 is freely available from the NEMO website (www.nemo-ocean.eu). After registration the

FORTRAN code is readily available using the open source subversion software (http://subervsion.apache.org). Additional modifications to the NEMO vn3.6 trunk (Appendix C.3) are also freely available code branches in the NEMO repository.

*Obtaining WAVEWATCH III*

WAVEWATCH III® is distributed under an open source style license to registered users through a password protected distribution site. The licence and link to request model code can be found at the NOAA National Weather Service

Environmental Modeling Center webpages at http://polar.ncep.noaa.gov/waves/wavewatch/. The model is subject to continuous development, with new releases generally becoming available after implementation of a new model version at NCEP. Research model versions may also be made available to those interested in and committed to basic model development, subject to agreement.

*Obtaining OASIS3-MCT*

OASIS3-MCT is disemminated to registered users as free software from https://verc.enes.org/oasis.

*Obtaining Rose*

Case study simulations and configuration control namelists were enabled using the rose suite control utilities. Further information is provided at http://metomi.github.io/rose/doc/rose.html, including documentation and installation instructions.

*Obtaining FCM*

All codes were built using the fcm make extract and build system provided within the Flexible Configuration Management (FCM) tools. Met Office Unified Model and JULES codes and rose suites were also configuration managed using this system. Further information is provided at http://metomi.github.io/fcm/doc/.

**Data availability**

The nature of the 4D data generated in running the various UKC2 case studies and a range of control simulations requires a

large tape storage facility. These data is of the order 10 Tb. However, the data can be made available upon contacting the authors. Each simulation namelist and input data are also archived under configuration management, and can be made available to researchers to promote collaboration upon contacting the authors.



Ocean bathymetry was obtained from the EMODnet Portal: EMODnet Bathymetry Consortium, EMODnet Digital Bathymetry (DTM), EMODnet Bathymetry (September 2015 release).





# Appendices

## Appendix A – List of symbols

| Symbol | Units | Description | Equation reference |
|---|---|---|---|
| $\Delta\mathbf{x}$ | | Finite difference of variable '$x$' | 5,13,14,15 |
| $\mathbf{x}_n$ | | Variable '$x$' specified at model level $n$ (surface at $n=0$) | 1,2,3,5,9,12,13,14,15 |
| | | | |
| $\Delta B$ | | Surface buoyancy flux | 15 |
| $b$ | - | PDM shape parameter | 17 |
| $C$ | - | WAVEWATCH III integration constant | 7,8 |
| $C_s$ | J m$^{-2}$ K$^{-1}$ | Areal heat capacity associated with surface material | 12 |
| $c$ | m s$^{-1}$ | Kinematic wave speed | 18 |
| $c_p$ | J kg$^{-1}$ K$^{-1}$ | Specific heat capacity of air | 2,13,15 |
| $c_D$ | - | Surface exchange coefficient for momentum | 5 |
| $c_H$ | - | Surface exchange coefficient for scalars | 13,14 |
| $E$ | kg m$^{-2}$ s$^{-1}$ | Evaporation | 19 |
| $E_0$ | kg m$^{-2}$ s$^{-1}$ | Turbulent moisture flux at surface | 3,12,13 |
| $F_{sat}$ | - | Saturated land surface grid cell fraction | 17 |
| $G$ | W m$^{-2}$ | Soil heat flux | 12 |
| $g$ | m s$^{-2}$ | Acceleration due to gravity | 2,6,13,15 |
| $H_0$ | W m$^{-2}$ | Turbulent sensible heat flux at surface | 2,12,13 |
| $I(z)$ | W m$^{-2}$ | Downward solar irradiance penetrating with ocean depth | 16 |
| $k$ | - | Wave number | 7,8 |
| $L$ | m | Monin-Obukhov length | 1 |
| $L_c$ | J kg$^{-1}$ | Latent heat of condensation of water at 0 °C | 12 |
| $Lw_\downarrow$ | W m$^{-2}$ | Downward component of longwave radiation | 12 |
| $N(k,\theta)$ | | Wave action density spectrum | 7 |
| $P$ | kg m$^{-2}$ s$^{-1}$ | Precipitation | 19 |
| $Q$ | W m$^{-2}$ | Net energy budget at surface | 11 |
| $Q_{ns}$ | W m$^{-2}$ | Net non-solar energy at surface | 11 |
| $Q_{sr}$ | W m$^{-2}$ | Solar radiation at surface | 11,16 |
| $Q_E$ | W m$^{-2}$ | Latent heat flux due to evaporation at surface | 11 |





| Symbol | Units | Description | Eq. |
|---|---|---|---|
| $Q_H$ | W m$^{-2}$ | Sensible heat flux at surface | 11 |
| $Q_{LW}$ | W m$^{-2}$ | Longwave radiation at surface | 11 |
| $q$ | kg kg$^{-1}$ | Specific humidity | 3,14,15 |
| $q_d$ | kg m$^{-2}$ s$^{-1}$ | Channel flow | 18 |
| $R$ | kg m$^{-2}$ s$^{-1}$ | Return flow between surface and sub-surface | 18 |
| $R_{abs}$ | - | Fraction of irradiance absorbed by ocean surface | 17 |
| $r$ | kg m$^{-2}$ s$^{-1}$ | Lateral water inflow | 18 |
| $S$ | kg m$^{-2}$ s$^{-1}$ | Soil water storage | 17 |
| $S_0$ | kg m$^{-2}$ s$^{-1}$ | Minimum soil water storage below which no saturation | 17 |
| $S_{in}(k,\theta_{wd})$ | - | Wind-wave interaction source term | 7,8 |
| $S_{out}(k,\theta_{wd})$ | - | Wind dissipation wave source term | 7 |
| $Sw_\downarrow$ | W m$^{-2}$ | Downward component of solar radiation | 12 |
| $T$ | K | Temperature | 2,12,15 |
| $t$ | s | Time | 12,18,19 |
| $U$ | m s$^{-1}$ | Atmospheric wind speed | 4,13,14 |
| $u_*$ | m s$^{-1}$ | Surface friction velocity | 1,2,3,4,5,6,7,10 |
| $\mathbf{v}$ | m s$^{-1}$ | Velocity vector | 1,5 |
| $V$ | m s$^{-1}$ | Ocean surface current speed | 19 |
| $w$ | m s$^{-1}$ | Vertical velocity component | 19 |
| $x$ | m | Horizontal zonal coordinate | 18 |
| $y$ | m | Horizontal meridional coordinate | 18 |
| $Z$ | - | Wave parameter (Tolman et al., 2014, Eq. 2.79) | 7 |
| $z$ | m | Vertical coordinate | 1,2,3,4,13,15,16 |
| $z_{00}$ | m | WAVEWATCH III initial guess for $z_{0m}$ [0.0095] | 9 |
| $z_{0m}$ | m | Surface roughness length for momentum | 4,6,9,10,13,15 |
| $z_{0h}$ | m | Surface roughness length for scalars (e.g. heat) | 13,15 |
| $z_\alpha$ | - | WAVEWATCH III zalpha constant [0.011] | 7,8 |
| $z_{PDM}$ | m | Depth of soil column considered in PDM scheme | 17 |
| | | | |
| $\alpha$ | - | Wave-dependent Charnock coefficient | 6,8,10 |
| $\alpha_s$ | - | Surface albedo | 12 |



| | | | |
|---|---|---|---|
| $\beta_{T1}, \beta_{q1}$ | - | Surface buoyancy coefficient | 15 |
| $\beta_{max}$ | - | Wave growth parameter | 7 |
| $\epsilon$ | | Surface emissivity | 12 |
| $\eta$ | m | Sea surface height | 19 |
| $\theta_{sat}$ | | Volumetric soil water content at saturation | 17 |
| $\theta_u$ | ° | Wind direction | 7,8 |
| $\theta_w$ | ° | Wave direction | 7,8 |
| $\kappa$ | - | von Karman constant [0.4] | 1,2,3,4,7 |
| $\nu$ | m s$^{-1}$ | Dynamic viscosity of air ($14 \times 10^{-6}$ m s$^{-1}$) | 6 |
| $\xi_0$ | - | e-folding depth scale for solar penetration in the ocean | 16 |
| $\xi_{rr}, \xi_{gg}, \xi_{bb}$ | - | Ocean extinction length scale for red, green and blue light | 16 |
| $\rho_0$ | kg m$^{-3}$ | Surface air density | 1,2,3,5,7,14 |
| $\rho_w$ | kg m$^{-3}$ | Surface water density | 7 |
| $\sigma$ | radian | Intrinsic wave frequency | 7 |
| $\sigma_{SB}$ | W m$^{-2}$ K$^{-4}$ | Stefan Boltzmann constant | 12 |
| $\tau_0$ | N m$^{-2}$ | Surface stress | 1,5,9 |
| $\tau_{hf}$ | N m$^{-2}$ | Stress supported by shorter waves | 8 |
| $\tau_w$ | N m$^{-2}$ | Wave supported stress | 8,9 |
| $\emptyset_m(z/L)$ | - | Monin-Obukhov stability function for momentum | 1 |
| $\emptyset_h(z/L)$ | - | Monin-Obukhov stability function for scalars | 2,3 |
| $\psi_m$ | - | Monin-Obukhov stability function for momentum | 4 |





## Appendix B – Derivation of the River Flow Model iteration scheme

The RFM river routing algorithm within JULES implemented as part of development towards UKC2 is based on a finite-difference iteration of the 1-d kinematic wave equation (Bell et al., 2007). A derivation of the algorithm is provided here for clarity to support the new code introduced to JULES and as an update to the original formulation introduced by Bell et al.

(2007).

Considering a 1-d flow down an inclined rectangular channel of uniform width, a continuity equation can be written as

$$\frac{\partial h}{\partial t} = -\frac{\partial q}{\partial x} + r \tag{B1}$$

where $x$ is the distance in the down-slope direction, $t$ is time (s), $h$ is the depth of the flow (m), $q$ is the discharge per unit width ($m^2$ $s^{-1}$) and $r$ is the rate of lateral inflow per unit width per unit length (m $s^{-1}$).

In practice, the routing algorithm is implemented in terms of a volume storage of water, $S = h.A$ in $m^3$ where $A$ is the grid cell area, in order to be applicable for applications with variable grid box areas (e.g. for latitude/longitude-based grids). Expressing Eq. (B1) in terms of $S$ and expressing the rate of change of $q$ with $S$ to be a constant, $c/A$, gives:

$$\frac{\partial S}{\partial t} = -\frac{\partial q}{\partial x}A + rA = -\left(\frac{\partial q}{\partial S}\frac{\partial S}{\partial x}\right)A + rA = -c\frac{\partial S}{\partial x} + rA \tag{B2}$$

A finite difference numerical solution to Eq. (B2) is computed if $t$ and $x$ are divided into discrete intervals $\Delta t$ and $\Delta x$ such

that $k$ and $n$ denote position in discrete space and time. Note this notation differs slightly from that used in Eq. (2) of Bell et al., (2007), but is more consistent with standard conventions. Making a forward difference approximation to the time derivative and a backward difference approximation to the spatial derivative, gives the first-order upwind scheme:

$$\frac{S_k^{n+1}-S_k^n}{\Delta t} = -c\frac{S_k^n-S_{k-1}^n}{\Delta x} + rA \tag{B3}$$

Normalising the wave speed $c$ by the characteristic velocity of the grid, so that $\theta = c\Delta t/\Delta x$ gives the update function:

$$S_k^{n+1} = (1-\theta)S_k^n + \theta S_{k-1}^n + rA\Delta t \tag{B4}$$

Equation B4 is solved separately for surface and sub-surface stores and distinguishing different flow speeds for surface and sub-surface flows in grid cells defined as land or river points. A return flow, $R$, is used to transfer a fraction of the water between surface and sub-surface stores within a grid cell. River flow, $q$, at time $k$ and location $n$ is then given by:

$$q_k^n = \frac{1}{A}\frac{c}{\Delta x}S_k^n \tag{B5}$$

Approximate values for the flow speed $c$ can be obtained by using the Darcy-Weisbach flow resistance formula for open channel flow to relate water flow velocity $v$ and depth $h$ as:

$$v = \left[\frac{2gR_hS_l}{f}\right]^{\frac{1}{2}} \tag{B6}$$

where $g$ is the gravitational acceleration, $R_h$ is the hydraulic radius (area / wetted perimeter), $S_l$ is the river bed or land slope and $f$ is the Darcy-Weisbach friction factor. Assuming shallow flow (i.e. $w \gg h$) then $R_h$ tends to $h$, and taking the

30 discharge per unit width $q = vh$ gives:





$$c = \frac{\partial q}{\partial h} = \frac{3}{2}\left[\frac{2ghS_l}{f}\right]^{\frac{1}{2}} = \frac{3}{2}v \qquad \text{(B7)}$$

With $S_l$ of order 0.001, $h$ typically 1 m and $f$ for a natural channel between 0.1 and 1.0 m s$^{-1}$, suggests that appropriate values for the surface wave speed $c$ tend to lie in the range 0.2-0.7 m s$^{-1}$ (Table C.3). The surface wave speed is also termed the kinematic wave celerity, giving the speed at which a disturbance propagates, rather than the water flow speed $v$.



## Appendix C – Code modifications applied for UKC2

A number of code adaptations were required in order to develop the UKC2 configuration, and associated UKA2, UKL2, UKO2 and UKW2 component and control configurations. These are summarised below. All code modifications are provided as distinct branches from the baseline code (e.g. trunk). These branches are merged to form a single code set as part of the
fcm_make configuration build process within each rose suite prior to running.

### C.1. Met Office Unified Model adaptations for UKC2

All MetUM codes used withinUKC2 are available to registered researchers via a shared MetUM code repository, which can be accessed via https://code.metoffice.gov.uk/trac/um/wiki. Table C1 summarises the code branches which were written and applied to the MetUM version 10.1 release code to enable running of the UKA2 component in uncoupled mode and as part
of the UKC2 coupled system. Merging with the baseline code, modifications for UKA2 and UKC2 are categorised as being required either to:

- enable the recommended land surface runoff generation and river routing science (see Sect. 3.2),
- enable dynamic coupling and exchange of information between the atmosphere and a wave model (see Sect. 2),
- couple effectively between ocean and atmosphere grids where valid data may not be available due to a mismatch in
coastlines, either due to grid interpolation or mis-matched land/sea masks (see Sect. 2).

### C.2 JULES adaptations for UKC2

All JULES codes used within UKC2 are available to registered researchers via a shared JULES code repository, which can be accessed via https://code.metoffice.gov.uk/trac/jules/wiki. Table C2 summarises the code branches which were written and applied to the release JULES version 4.2 baseline code to enable running of the UKA2/UKL2 atmosphere-land component in
uncoupled mode and as part of the UKC2 coupled system. These branches are merged to form a single code set as part of the fcm_make configuration build process within each suite. Merging with the root code, modifications for UKA2/UKL2 and UKC2 are required either to:

- enable the recommended slope-dependent formulation of PDM (see Sect. 3.2.2),
- enable use of a spatially-varying Charnock parameter in surface exchange rather than a constant (see Sect. 3.2.1).

### C.3 NEMO ocean code adaptations for UKC2

All NEMO codes used are available to registered users via a shared NEMO code repository, which can be accessed via http://www.nemo-ocean.eu.

Table C3 summarises all NEMO vn3.6 code adaptations implemented for inclusion within the UKC2 system. Merging with the baseline trunk code at revision 5518, modifications for UKO2 and UKC2 are required either to:

- update to known bug-fixes found in the initial NEMO vn3.6_STABLE release
- apply capability specific to running NEMO for a domain including a shelf-seas region



- represent the inverse barometer effect of surface pressure on the ocean
- enable technical exchange of information between NEMO and a wave model via a coupler
- enable NEMO to run within a coupled system without necessarily including the Unified Model and its coupling utilities as the 'master' system, for example to be able to run ocean-wave coupling only.

## 5  C.4 WAVEWATCH III wave code adaptations for UKC2

The WAVEWATCHIII code base is distributed by NOAA under an open source style licence via http://polar.ncep.noaa.gov/waves/wavewatch/wavewatch.shtml. Interested readers wishing to access the code are requested to register to obtain a license via http://polar.ncep.noaa.gov/waves/wavewatch/license.shtml. Model codes used in UK Environmental Prediction research are kept under configuration management via a mirror repository hosted at the Met 10 Office, and can be made available to researchers for collaboration on request, given prior approval to access WAVEWATCHIII from NOAA.

Table C4 summarises the key WAVEWATCH III version 4.18 code adaptations implemented for inclusion within the UKC2 system. A new code branch has been developed to handle coupling between WAVEWATCH III and atmosphere and ocean codes. In this implementation coupling is achieved by the use of the serial partition, in which only one processor 15 sends/receives the information to/from other models. Currently, the fields that can be exchanged via coupling are the Charnock parameter (send), significant wave height (send), wave to ocean energy flux (send), 10 m wind components (receive), total ice fraction (receive), superficial current components (receive), and the water depths (receive). The list of exchanged fields will be extended after new science changes are added to the ocean component. It should be noted that the more recent WAVEWATCH III trunk code at version 5.0 includes support for coupling with OASIS3-MCT, and this will be 20 included and further developed within a future UKC3 configuration.

**Supplement link (will be included by Copernicus)**

**Team list**

**Author contribution**

25  **Competing interests**

The authors declare that they have no conflict of interest





**Disclaimer**

**Acknowledgements**

This research has been carried out under national capability funding as part of a directed effort on UK Environmental Prediction, in collaboration between Centre for Ecology & Hydrology (CEH), the Met Office, National Oceanography

5 Centre (NOC) and Plymouth Marine Laboratory (PML).





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





| Order | Interface | Exchanged variable | Symbol | Units | Frequency | Time processing |
|---|---|---|---|---|---|---|
| | | | | | | |
| 1 | W – A | Wave-dependent Charnock parameter | $\alpha$ | - | 1 hour | Instantaneous |
| | | | | | | |
| 2 | O – A | Sea surface temperature | $SST$ | K | 1 hour | Instantaneous |
| 2 | O – A | Zonal surface current | $u_{curr}$ | m s$^{-1}$ | 1 hour | Instantaneous |
| 2 | O – A | Meridional surface current | $v_{curr}$ | m s$^{-1}$ | 1 hour | Instantaneous |
| | | | | | | |
| 3 | O – W | Water level relative to local bathymetry | $d$ | m | 1 hour | Instantaneous |
| 3 | O – W | Zonal surface current | $u_{curr}$ | m s$^{-1}$ | 1 hour | Instantaneous |
| 3 | O – W | Meridional surface current | $v_{curr}$ | m s$^{-1}$ | 1 hour | Instantaneous |
| | | | | | | |
| 4 | A – O | Zonal wind stress on ocean surface | $\tau_x$ | N m$^{-2}$ | 1 hour | Instantaneous |
| 4 | A – O | Meridional wind stress on ocean surface | $\tau_y$ | N m$^{-2}$ | 1 hour | Instantaneous |
| 4 | A – O | Solar surface heat flux (all wavelengths) | $Q_{sr}$ | W m$^{-2}$ | 1 hour | Instantaneous |
| 4 | A – O | Non-solar net surface heat flux | $Q_{ns}$ | W m$^{-2}$ | 1 hour | Instantaneous |
| 4 | A – O | Rainfall rate | $R$ | kg m$^{-2}$ s$^{-1}$ | 1 hour | Instantaneous |
| 4 | A – O | Snowfall rate | $S$ | kg m$^{-2}$ s$^{-1}$ | 1 hour | Instantaneous |
| 4 | A – O | Evaporation of fresh water from ocean | $E$ | kg m$^{-2}$ s$^{-1}$ | 1 hour | Instantaneous |
| 4 | A – O | Wind speed at 10 m above ocean surface | $w_{10}$ | m s$^{-1}$ | 1 hour | Instantaneous |
| 4 | A – O | Mean sea level pressure | $Pmsl$ | Pa | 1 hour | Instantaneous |
| | | | | | | |
| 5 | A – W | Zonal wind speed at 10 m above surface | $U_{10}$ | m s$^{-1}$ | 1 hour | Instantaneous |
| 5 | A – W | Meridional wind speed at 10 m height | $V_{10}$ | m s$^{-1}$ | 1 hour | Instantaneous |

**Table 1: Summary of coupling exchanges between atmosphere/land (A), ocean (O) and wave (W) components within the UKC2 coupled prediction system. Ensuring that exchanges occur between model components in the coupling order shown avoids system deadlocks within OASIS3-MCT. The coupling frequency highlights that all fields are currently exchanged every hour of the simulation time, and that all fields are instantaneous (i.e. no temporal or spatial averaging). See Sect. 2 for further details.**

| Suite configuration name | Status | Comment |
|---|---|---|
| UKA2g | Atmosphere only | Global OSTIA SST boundary condition persisted |
| UKA2h | Atmosphere only | High-resolution UKO2g SST boundary condition persisted |
| UKO2g | Ocean only | Global (17 km) Unified Model meteorology forcing |
| UKO2h | Ocean only | High resolution UKA2h UM meteorology forcing |
| UKW2g | Wave only | Global (17 km) Unified Model wind forcing |
| UKW2h | Wave only | High resolution UKA2h UM wind forcing |
| UKW2c | Wave only | As UKW2h, with UKO2h current forcing (wind + current) |
| UKW2l | Wave only | As UKW2c with UKO2l water level forcing (wind + current + level) |
| UKC2ao | Coupled A-O | Atmosphere-ocean coupled suite, no wave interactions |
| UKC2ow | Coupled O-W | Ocean-wave "partially coupled" suite, no atmosphere interactions |
| UKC2aow | Coupled A-O-W | Fully coupled atmosphere-ocean-wave suite |

**Table 2: Summary of UKC2 system coupled and uncoupled evaluation suites**



| | Suite differences | Research purpose |
|---|---|---|
| *Characterising atmosphere and land surface sensitivity* | | |
| | UKA2h – UKA2g | Sensitivity of modelled atmosphere/land state to initial SST (resolution and drift from analysis) |
| | UKC2ao – UKA2h | Sensitivity of atmosphere/land state to ocean coupling and feedbacks |
| | UKC2aow – UKA2h | Sensitivity of atmosphere/land state to fully coupled ocean and wave feedbacks |
| | UKC2aow – UKC2ao | Examine impact of wave coupling on modelled atmosphere feedbacks |
| *Characterising ocean sensitivity* | | |
| | UKO2h – UKO2g | Sensitivity of modelled ocean state to meteorological forcing resolution |
| | UKC2ow – UKO2h | (expect null test, given no feedbacks from wave to ocean implemented in UKC2) |
| | UKC2ao – UKO2h | Sensitivity of modelled ocean state to representing atmospheric feedbacks |
| | UKC2aow – UKO2h | Sensitivity of modelled ocean state to fully coupled atmosphere and wave feedbacks |
| | UKC2aow – UKC2ao | Examine impact of wave coupling on modelled ocean feedbacks |
| *Characterising wave sensitivity* | | |
| | UKW2h – UKW2g | Sensitivity of wave state to meteorological forcing resolution |
| | UKW2c – UKW2h | Sensitivity of wave state to current forcing |
| | UKW2l – UKW2c | Sensitivity of wave state to water level forcing |
| | UKC2ow – UKW2l | (expect null test, given no feedbacks from wave to ocean implemented in UKC2) |
| | UKC2aow – UKO2l | Sensitivity of wave simulations to fully coupled atmosphere and ocean feedbacks |

**Table 3: Summary of research approaches enabled by comparison of UKC2 system coupled and uncoupled evaluation suites**



**(a)**

| 4 – 11 Aug 2014 | UKC2aow | | | UKC2ao | | | Forced UKA2h | | | Forced UKA2g | | |
|---|---|---|---|---|---|---|---|---|---|---|---|---|
| | RMSE | \|Bias\| | r² | RMSE | \|Bias\| | r² | RMSE | \|Bias\| | r² | RMSE | \|Bias\| | r² |
| [Aₒ] Sfc temp | **0.70** | **0.50** | **0.48** | **0.69** | **0.50** | **0.47** | 0.85 | 0.62 | 0.00 | 0.79 | 0.54 | 0.00 |
| (° C) [65] | 15 | 16 | **31** | 16 | 10 | 32 | 9 | 10 | 0 | **25** | **29** | 1 |
| [Aₒ] Air temp | **1.14** | **0.44** | 0.31 | **1.11** | **0.43** | 0.34 | 1.22 | 0.53 | 0.29 | **1.18** | **0.48** | 0.29 |
| (° C) [74] | 10 | 17 | 14 | **38** | 22 | **38** | 7 | 10 | 11 | 19 | **25** | 11 |
| [Aₒ] Wind speed | 3.38 | **1.25** | 0.51 | 3.22 | 1.06 | 0.54 | 3.18 | 0.93 | 0.54 | 3.12 | 1.06 | 0.56 |
| (m/s) [42] | 7 | 7 | 6 | 8 | 9 | 8 | 10 | **19** | 10 | 17 | 7 | **18** |

**(b)**

| 4 – 11 Aug 2014 | UKC2aow | | | UKC2ao | | | Forced UKA2h | | | Forced UKA2g | | |
|---|---|---|---|---|---|---|---|---|---|---|---|---|
| | RM5E | \|Bias\| | r² | RMSE | \|Bias\| | r² | RMSE | \|Bias\| | r² | RMSE | \|Bias\| | r² |
| [A] Air temp | **1.40** | **0.43** | 0.64 | **1.41** | **0.43** | 0.65 | 1.43 | 0.45 | 0.64 | **1.41** | **0.42** | 0.64 |
| (° C) [534] | 131 | 120 | 137 | 129 | 113 | 136 | 93 | 125 | 112 | **181** | **176** | **149** |
| [A] Wind speed | 2.27 | **0.88** | 0.52 | 2.24 | 0.85 | 0.53 | 2.24 | 0.84 | 0.53 | **2.22** | 0.85 | 0.53 |
| (m/s) [412] | 87 | 101 | 88 | 91 | 88 | 97 | 99 | **120** | 95 | **135** | 103 | **132** |

**(c)**

| 4 – 11 Aug 2014 | UKC2aow | | | UKC2ao *[UKW2l]* | | | Forced UK[OW]2h | | | Forced UK[OW]2g | | |
|---|---|---|---|---|---|---|---|---|---|---|---|---|
| | RMSE | \|Bias\| | r² | RMSE | \|Bias\| | r² | RMSE | \|Bias\| | r² | RMSE | \|Bias\| | r² |
| [O] SST (° C) | 0.68 | 0.53 | 0.52 | 0.68 | 0.53 | 0.51 | 0.70 | 0.55 | 0.51 | 0.71 | 0.55 | **0.56** |
| [82] | **29** | 26 | 11 | 22 | 18 | 11 | 13 | 11 | 14 | 18 | 27 | **46** |
| [O] SSH (m) | 1.06 | 0.60 | 0.80 | 1.08 | 0.64 | 0.80 | 1.04 | 0.58 | 0.80 | **1.04** | 0.57 | **0.80** |
| [41] | 2 | 2 | 8 | 2 | 1 | 3 | 5 | 9 | 4 | **32** | **29** | 26 |
| [W] Sig. wave ht. | 0.31 | 0.12 | 0.69 | 0.29 | 0.10 | 0.72 | 0.29 | 0.10 | 0.73 | **0.19** | **0.07** | **0.88** |
| (m) [60] | 0 | 6 | 2 | 0 | 9 | 0 | 3 | 9 | 1 | **57** | 36 | 57 |
| [W] Peak period | 1.61 | 0.51 | 0.25 | 1.61 | 0.50 | 0.25 | 1.72 | 0.46 | 0.24 | 1.65 | 0.53 | **0.34** |
| (s) [44] | 9 | 12 | 8 | 9 | 6 | 8 | 6 | 13 | 4 | **20** | 13 | **24** |

5   **Table 4: Summary RMSE, bias and r2 correlation coefficient statistics for the 7-11 August 2014 case study. Average statistics are computed from a comparison of model output against in-situ observations of (a) atmosphere [Aₒ] model variables over sea grid cells only (b) all atmosphere [A] model variables, and (c) ocean [O] and wave [W] model variables. The relevant forced configuration with high-resolution forcing UKW2h, UKO2h and UKA2h are treated as the reference control model. Results showing a statistically significant difference from the reference according to a t-test at the 95% confidence level are underlined.**
10  **Those also showing an improvement against the reference are also in bold. Also listed for each variable in lighter text is the number of sites for which the model configuration provided the best statistic relative to the other model configurations tested for this case. The configuration with the highest number of sites listed is highlighted in bold. Numbers in square brackets show the number of observation locations assessed for each variable.**




**(a)**

| 6 – 10 Sep 2014 | UKC2aow | | | UKC2ao | | | Forced UKA2h | | | Forced UKA2g | | |
|---|---|---|---|---|---|---|---|---|---|---|---|---|
| | RMSE | \|Bias\| | r² | RMSE | \|Bias\| | r² | RMSE | \|Bias\| | r² | RMSE | \|Bias\| | r² |
| [Aₒ] Sfc temp | 0.50 | 0.36 | **__0.22__** | 0.50 | 0.36 | **__0.21__** | 0.50 | 0.37 | 0.01 | 0.44 | 0.31 | 0.00 |
| (° C) [65] | 13 | 14 | **36** | 7 | 6 | 28 | 15 | 17 | 0 | **30** | **28** | 0 |
| [Aₒ] Air temp | **__0.94__** | **__0.42__** | 0.37 | **__0.95__** | **__0.42__** | 0.36 | 0.97 | 0.44 | 0.35 | **__0.97__** | 0.44 | 0.35 |
| (° C) [78] | **29** | 26 | 28 | 9 | 13 | 8 | 20 | 15 | 19 | 20 | 27 | 26 |
| [Aₒ] Wind speed | 1.75 | 0.84 | 0.62 | 1.75 | 0.85 | 0.63 | 1.75 | 0.82 | 0.62 | 1.77 | 0.87 | 0.61 |
| (m/s) [47] | 8 | 10 | 5 | **15** | 11 | **18** | 12 | **16** | 13 | 12 | 10 | 11 |

**(b)**

| 6 – 10 Sep 2014 | UKC2aow | | | UKC2ao | | | Forced UKA2h | | | Forced UKA2g | | |
|---|---|---|---|---|---|---|---|---|---|---|---|---|
| | RMSE | \|Bias\| | r² | RMSE | \|Bias\| | r² | RMSE | \|Bias\| | r² | RMSE | \|Bias\| | r² |
| [A] Air temp | **__1.54__** | **__0.61__** | **__0.71__** | **__1.54__** | **__0.61__** | **__0.71__** | 1.57 | 0.66 | 0.71 | **__1.57__** | **__0.64__** | 0.71 |
| (° C) [552] | 177 | 164 | 154 | 143 | 155 | 127 | 105 | 68 | 118 | 127 | 165 | 153 |
| [A] Wind speed | 1.49 | 0.64 | 0.37 | 1.49 | __0.64__ | 0.38 | 1.49 | 0.63 | 0.37 | 1.49 | 0.64 | **__0.38__** |
| (m/s) [427] | 91 | 99 | 78 | 103 | 81 | 129 | 101 | **130** | 90 | **132** | 117 | 129 |

**(c)**

| 6 – 10 Sep 2014 | UKC2aow | | | UKC2ao [UKW2l] | | | Forced UK[OW]2h | | | Forced UK[OW]2g | | |
|---|---|---|---|---|---|---|---|---|---|---|---|---|
| | RMSE | \|Bias\| | r² | RMSE | \|Bias\| | r² | RMSE | \|Bias\| | r² | RMSE | \|Bias\| | r² |
| [O] SST (° C) | **__0.55__** | **__0.43__** | 0.28 | **__0.55__** | **__0.43__** | 0.28 | 0.58 | 0.46 | 0.28 | __0.69__ | __0.56__ | 0.29 |
| [82] | 35 | 33 | 20 | 24 | 22 | 10 | 6 | 11 | 19 | 17 | 16 | **33** |
| [O] SSH (m) | 1.20 | 0.70 | 0.79 | 1.20 | 0.70 | 0.79 | 1.20 | 0.70 | 0.79 | 1.20 | 0.70 | 0.79 |
| [42] | 7 | 9 | 2 | 5 | 1 | **21** | 8 | 8 | 7 | **21** | **23** | 11 |
| [W] Sig. wave ht. | 0.15 | 0.07 | 0.61 | 0.15 | 0.07 | 0.61 | 0.15 | 0.07 | 0.59 | **__0.13__** | 0.08 | **__0.72__** |
| (m) [61] | 11 | 16 | 10 | 9 | 7 | 5 | 11 | **22** | 3 | **30** | 16 | **43** |
| [W] Peak period | **__2.86__** | **__0.96__** | 0.28 | **__2.83__** | **__0.87__** | 0.27 | 3.47 | 1.28 | 0.23 | 3.49 | 1.31 | 0.26 |
| (s) [42] | 10 | 8 | 13 | 15 | **19** | 11 | 2 | 5 | 5 | 15 | 10 | 13 |

5 **Table 5: Summary RMSE, bias and r2 correlation coefficient statistics for the 6-10 September 2014 case study. Average statistics are computed from a comparison of model output against in-situ observations of (a) atmosphere [Aₒ] model variables over sea grid cells only, (b) all atmosphere [A] model variables, and (c) ocean [O] and wave [W] model variables. The relevant forced configuration with high-resolution forcing UKW2h, UKO2h and UKA2h are treated as the reference control model. Results showing a statistically significant difference from the reference according to a t-test at the 95% confidence level are underlined.**
10 **Those also showing an improvement against the reference are also in bold. Also listed for each variable in lighter text is the number of sites for which the model configuration provided the best statistic relative to the other model configurations tested for this case. The configuration with the highest number of sites listed is highlighted in bold. Numbers in square brackets show the number of observation locations assessed for each variable.**





**(a)**

| *2 – 6 Oct 2014* | UKC2aow | | | UKC2ao | | | Forced UKA2h | | | Forced UKA2g | | |
|---|---|---|---|---|---|---|---|---|---|---|---|---|
| | *RMSE* | *\|Bias\|* | *r²* | *RMSE* | *\|Bias\|* | *r²* | *RMSE* | *\|Bias\|* | *r²* | *RMSE* | *\|Bias\|* | *r²* |
| [Aₒ] Sfc temp | **0.44** | **0.37** | **0.68** | **0.44** | **0.37** | **0.67** | 0.60 | 0.49 | 0.01 | 0.53 | 0.39 | 0.01 |
| (° C) [66] | 17 | 19 | **34** | 21 | 14 | 32 | 5 | 7 | 0 | 23 | **26** | 0 |
| [Aₒ] Air temp | **1.07** | **0.43** | 0.59 | **1.08** | **0.43** | 0.58 | 1.13 | 0.50 | 0.56 | **1.10** | **0.45** | 0.56 |
| (° C) [80] | 27 | 30 | 33 | 25 | 17 | 26 | 10 | 15 | 11 | 18 | 18 | 10 |
| [Aₒ] Wind speed | **2.99** | **1.19** | 0.58 | 3.14 | 1.37 | 0.57 | 3.15 | 1.44 | 0.58 | 3.18 | 1.46 | 0.58 |
| (m/s) [50] | 34 | 38 | 11 | 6 | 3 | 7 | 6 | 4 | **19** | 4 | 5 | 13 |

**(b)**

| *2 – 6 Oct 2014* | UKC2aow | | | UKC2ao | | | Forced UKA2h | | | Forced UKA2g | | |
|---|---|---|---|---|---|---|---|---|---|---|---|---|
| | *RMSE* | *\|Bias\|* | *r²* | *RMSE* | *\|Bias\|* | *r²* | *RMSE* | *\|Bias\|* | *r²* | *RMSE* | *\|Bias\|* | *r²* |
| [A] Air temp | **1.56** | **0.57** | **0.77** | **1.56** | **0.57** | 0.77 | 1.59 | 0.62 | 0.76 | **1.58** | **0.58** | 0.76 |
| (° C) [506] | 151 | 147 | 123 | 148 | 124 | 119 | 101 | 104 | 116 | 106 | 131 | **148** |
| [A] Wind speed | **2.13** | **0.92** | 0.58 | 2.15 | 0.95 | 0.58 | 2.16 | 0.96 | 0.58 | 2.15 | 0.97 | 0.58 |
| (m/s) [368] | 118 | 119 | 79 | 82 | 75 | 84 | 84 | 79 | 97 | 84 | 95 | **108** |

**(c)**

| *2 – 6 Oct 2014* | UKC2aow | | | UKC2ao *[UKW2l]* | | | Forced UK[OW]2h | | | Forced UK[OW]2g | | |
|---|---|---|---|---|---|---|---|---|---|---|---|---|
| | *RMSE* | *\|Bias\|* | *r²* | *RMSE* | *\|Bias\|* | *r²* | *RMSE* | *\|Bias\|* | *r²* | *RMSE* | *\|Bias\|* | *r²* |
| [O] SST (° C) | **0.41** | **0.32** | 0.71 | **0.41** | 0.33 | 0.71 | 0.43 | 0.34 | 0.70 | **0.48** | **0.40** | 0.67 |
| [86] | 28 | **29** | 18 | 30 | 26 | 16 | 14 | 13 | **28** | 14 | 18 | 24 |
| [O] SSH (m) | 1.00 | 0.65 | **0.79** | 1.00 | 0.65 | **0.79** | 1.00 | 0.65 | 0.79 | 1.00 | 0.65 | 0.79 |
| [41] | 12 | 14 | 6 | 2 | 2 | 3 | 6 | 6 | 12 | **21** | **19** | **20** |
| [W] Sig. wave ht. | 0.29 | 0.12 | 0.81 | *0.29* | *0.11* | *0.81* | 0.28 | 0.10 | 0.81 | **0.24** | 0.13 | **0.86** |
| (m) [63] | 6 | 17 | 9 | *9* | *15* | *2* | 5 | 15 | 1 | **43** | 16 | **51** |
| [W] Peak period | **2.58** | **1.64** | 0.56 | *2.68* | *1.71* | *0.56* | 2.72 | 1.81 | 0.56 | 2.72 | 1.76 | 0.58 |
| (s) [14] | **8** | 9 | 3 | *2* | *1* | *3* | 1 | 1 | 5 | 3 | 3 | 3 |

5  **Table 6: Summary RMSE, bias and r2 correlation coefficient statistics for the 2-6 October 2014 case study. Average statistics are computed from a comparison of model output against in-situ observations of (a) atmosphere [Aₒ] model variables over sea grid cells only, (b) all atmosphere [A] model variables, and (c) ocean [O] and wave [W] model variables. The relevant forced configuration with high-resolution forcing UKW2h, UKO2h and UKA2h are treated as the reference control model. Results showing a statistically significant difference from the reference according to a t-test at the 95% confidence level are underlined.**
10  **Those also showing an improvement against the reference are also in bold. Also listed for each variable in lighter text is the number of sites for which the model configuration provided the best statistic relative to the other model configurations tested for this case. The configuration with the highest number of sites listed is highlighted in bold.  Numbers in square brackets show the number of observation locations assessed for each variable.**





**(a)**

| 7 – 11 Dec 2014 | UKC2aow | | | UKC2ao | | | Forced UKA2h | | | Forced UKA2g | | |
|---|---|---|---|---|---|---|---|---|---|---|---|---|
| | RMSE | \|Bias\| | r² | RMSE | \|Bias\| | r² | RMSE | \|Bias\| | r² | RMSE | \|Bias\| | r² |
| [Aₒ] Sfc temp | **0.69** | **0.56** | **0.46** | **0.70** | **0.56** | **0.46** | 0.81 | 0.66 | 0.00 | 0.93 | 0.80 | 0.00 |
| (° C) [68] | 18 | 16 | **39** | 16 | 12 | 28 | 12 | 16 | 0 | **22** | **24** | 0 |
| [Aₒ] Air temp | 1.10 | 0.50 | 0.59 | 1.10 | 0.50 | 0.59 | 1.06 | 0.45 | 0.59 | 1.07 | **0.43** | 0.59 |
| (° C) [86] | 15 | 15 | 22 | 10 | 5 | 17 | 29 | 22 | **25** | 32 | 44 | 22 |
| [Aₒ] Wind speed | 2.93 | **1.26** | 0.63 | 3.07 | 1.47 | 0.64 | 3.08 | 1.47 | 0.63 | 3.08 | 1.47 | 0.63 |
| (m/s) [46] | **25** | **30** | 12 | 4 | 3 | 13 | 9 | 6 | 11 | 8 | 7 | 10 |

**(b)**

| 7 – 11 Dec 2014 | UKC2aow | | | UKC2ao | | | Forced UKA2h | | | Forced UKA2g | | |
|---|---|---|---|---|---|---|---|---|---|---|---|---|
| | RMSE | \|Bias\| | r² | RMSE | \|Bias\| | r² | RMSE | \|Bias\| | r² | RMSE | \|Bias\| | r² |
| [A] Air temp | 1.34 | 0.84 | 0.73 | 1.35 | 0.84 | 0.73 | 1.32 | 0.80 | 0.73 | **1.31** | **0.78** | 0.74 |
| (° C) [544] | 73 | 54 | 136 | 60 | 49 | 97 | 119 | 124 | 130 | **292** | **317** | **181** |
| [A] Wind speed | 2.32 | 1.13 | 0.60 | 2.33 | 1.15 | 0.60 | 2.34 | 1.15 | 0.60 | 2.33 | 1.15 | 0.60 |
| (m/s) [409] | 111 | 108 | 93 | 100 | 101 | 108 | 91 | 90 | 98 | 107 | 110 | 110 |

**(c)**

| 7 – 11 Dec 2014 | UKC2aow | | | UKC2ao [UKW2l] | | | Forced UK[OW]2h | | | Forced UK[OW]2g | | |
|---|---|---|---|---|---|---|---|---|---|---|---|---|
| | RMSE | \|Bias\| | r² | RMSE | \|Bias\| | r² | RMSE | \|Bias\| | r² | RMSE | \|Bias\| | r² |
| [O] SST (° C) | 0.60 | 0.47 | 0.53 | 0.60 | 0.47 | **0.53** | 0.60 | 0.48 | 0.51 | 0.65 | 0.51 | 0.48 |
| [85] | 26 | 20 | 22 | 16 | 13 | 25 | 26 | 26 | 15 | 17 | 26 | 23 |
| [O] SSH (m) | 1.36 | 0.90 | 0.78 | 1.37 | 0.91 | 0.79 | 1.37 | 0.90 | 0.79 | 1.37 | 0.91 | 0.79 |
| [41] | 16 | 18 | 16 | 2 | 1 | 1 | 6 | 3 | 10 | 17 | 19 | 14 |
| [W] Sig. wave ht. | **0.34** | 0.19 | **0.84** | *0.35* | *0.17* | *0.82* | 0.36 | 0.17 | 0.82 | 0.35 | 0.17 | **0.84** |
| (m) [63] | **30** | 16 | **32** | *4* | *12* | *3* | 5 | 11 | 8 | 24 | **24** | 20 |
| [W] Peak period | 2.98 | 2.32 | 0.61 | *2.98* | *2.28* | *0.59* | 3.02 | 2.39 | 0.60 | 2.95 | 2.32 | **0.62** |
| (s) [18] | 5 | 6 | 4 | 6 | 6 | 2 | 0 | 0 | 3 | 7 | 0 | 9 |

5 **Table 7: Summary RMSE, bias and r2 correlation coefficient statistics for the 7-11 December 2014 case study. Average statistics are computed from a comparison of model output against in-situ observations of (a) atmosphere [Aₒ] model variables over sea grid cells only, (b) all atmosphere [A] model variables, and (c) ocean [O] and wave [W] model variables. The relevant forced configuration with high-resolution forcing UKW2h, UKO2h and UKA2h are treated as the reference control model. Results showing a statistically significant difference from the reference according to a t-test at the 95% confidence level are underlined.**
10 **Those also showing an improvement against the reference are also in bold. Also listed for each variable in lighter text is the number of sites for which the model configuration provided the best statistic relative to the other model configurations tested for this case. The configuration with the highest number of sites listed is highlighted in bold. Numbers in square brackets show the number of observation locations assessed for each variable.**





**(a)**

| 7 – 11 Feb 2015 | UKC2aow | | | UKC2ao *[UKW2l]* | | | Forced UKA2h | | | Forced UKA2g | | |
|---|---|---|---|---|---|---|---|---|---|---|---|---|
| | *RMSE* | *|Bias|* | *r²* | *RMSE* | *|Bias|* | *r²* | *RMSE* | *|Bias|* | *r²* | *RMSE* | *|Bias|* | *r²* |
| [A$_o$] Sfc temp | 0.73 | 0.63 | **0.22** | 0.73 | 0.63 | **0.22** | 0.77 | 0.69 | 0.01 | 0.79 | 0.72 | 0.01 |
| (° C) [70] | 12 | 10 | **38** | 11 | 12 | 30 | 21 | 21 | 1 | **26** | **27** | 0 |
| [A$_o$] Air temp | 0.96 | 0.46 | 0.54 | 0.96 | 0.46 | 0.53 | 0.96 | 0.46 | 0.53 | 0.94 | 0.43 | 0.53 |
| (° C) [83] | 22 | 19 | **32** | 11 | 8 | 16 | 14 | 15 | 16 | **36** | **41** | 19 |
| [A$_o$] Wind speed | 2.03 | 0.76 | 0.61 | 2.07 | 0.88 | 0.62 | 2.05 | 0.82 | 0.62 | 2.05 | 0.83 | 0.62 |
| (m/s) [39] | 14 | **24** | 6 | 5 | 2 | 12 | 7 | 6 | 5 | 13 | 7 | **16** |

**(b)**

| 7 – 11 Feb 2015 | UKC2aow | | | UKC2ao *[UKW2l]* | | | Forced UKA2h | | | Forced UKA2g | | |
|---|---|---|---|---|---|---|---|---|---|---|---|---|
| | *RMSE* | *|Bias|* | *r²* | *RMSE* | *|Bias|* | *r²* | *RMSE* | *|Bias|* | *r²* | *RMSE* | *|Bias|* | *r²* |
| [A] Air temp | **1.55** | **0.56** | **0.54** | **1.56** | **0.57** | 0.53 | 1.56 | 0.58 | 0.53 | 1.58 | 0.63 | 0.53 |
| (° C) [537] | **234** | **211** | **186** | 79 | 58 | 84 | 107 | 135 | 96 | 117 | 133 | 171 |
| [A] Wind speed | 1.78 | 0.79 | 0.47 | 1.79 | 0.81 | 0.46 | 1.78 | 0.80 | 0.46 | 1.78 | 0.80 | 0.46 |
| (m/s) [402] | 126 | 136 | 106 | 64 | 63 | 78 | 89 | 72 | 85 | 123 | 131 | **133** |

**(c)**

| 7 – 11 Feb 2015 | UKC2aow | | | UKC2ao *[UKW2l]* | | | Forced UK[OW]2h | | | Forced UK[OW]2g | | |
|---|---|---|---|---|---|---|---|---|---|---|---|---|
| | *RMSE* | *|Bias|* | *r²* | *RMSE* | *|Bias|* | *r²* | *RMSE* | *|Bias|* | *r²* | *RMSE* | *|Bias|* | *r²* |
| [O] SST (° C) | 0.69 | 0.60 | 0.28 | 0.69 | 0.59 | 0.28 | 0.70 | 0.60 | 0.27 | 0.72 | 0.61 | 0.27 |
| [90] | 23 | 20 | 26 | 20 | 17 | 8 | 13 | 16 | 22 | **34** | **37** | **34** |
| [O] SSH (m) | **0.61** | **0.34** | **0.87** | **0.61** | **0.34** | **0.87** | 0.61 | 0.34 | 0.87 | 0.61 | 0.34 | 0.87 |
| [41] | **21** | **26** | 2 | 1 | 1 | 1 | 3 | 0 | 14 | 16 | 14 | **24** |
| [W] Sig. wave ht. | 0.21 | 0.12 | 0.73 | *0.21* | *0.11* | *0.73* | 0.21 | 0.11 | 0.72 | **0.19** | 0.12 | **0.78** |
| (m) [69] | 7 | 15 | 14 | *12* | *13* | *11* | 7 | 14 | 3 | **43** | **27** | **41** |
| [W] Peak period | **3.63** | **2.40** | 0.17 | *3.55* | *2.30* | *0.15* | 4.00 | 2.72 | 0.13 | 4.17 | 2.94 | 0.12 |
| (s) [15] | 5 | 4 | 3 | *6* | *7* | *5* | 2 | 1 | 2 | 2 | 3 | 5 |

**Table 8: Summary RMSE, bias and r2 correlation coefficient statistics for the 7-11 February 2015 case study. Average statistics are computed from a comparison of model output against in-situ observations of (a) atmosphere [A$_o$] model variables over sea grid cells only, (b) all atmosphere [A] model variables, and (c) ocean [O] and wave [W] model variables. The relevant forced configuration with high-resolution forcing UKW2h, UKO2h and UKA2h are treated as the reference control model. Results showing a statistically significant difference from the reference according to a t-test at the 95% confidence level are underlined. Those also showing an improvement against the reference are also in bold. Also listed for each variable in lighter text is the number of sites for which the model configuration provided the best statistic relative to the other model configurations tested for this case. The configuration with the highest number of sites listed is highlighted in bold. Numbers in square brackets show the number of observation locations assessed for each variable.**




**(a)**

| 30 Jun – 4 Jul 2015 | UKC2aow | | | UKC2ao | | | Forced UKA2h | | | Forced UKA2g | | |
|---|---|---|---|---|---|---|---|---|---|---|---|---|
| | RMSE | \|Bias\| | $r^2$ | RMSE | \|Bias\| | $r^2$ | RMSE | \|Bias\| | $r^2$ | RMSE | \|Bias\| | $r^2$ |
| [$A_o$] Sfc temp (° C) [69] | **__0.79__** | **__0.56__** | **__0.35__** | **__0.79__** | **__0.52__** | **__0.34__** | 0.91 | 0.71 | 0.00 | 0.99 | 0.80 | 0.00 |
| | 19 | 8 | **37** | 15 | 25 | 29 | 14 | 16 | 1 | 21 | 20 | 0 |
| [$A_o$] Air temp (° C) [84] | **__1.51__** | **__0.94__** | 0.58 | **__1.51__** | **__0.94__** | 0.58 | 1.55 | 1.00 | 0.56 | **__1.51__** | **__0.97__** | 0.57 |
| | 9 | 5 | 11 | 18 | 27 | 23 | 29 | 28 | 24 | 28 | 24 | 26 |
| [$A_o$] Wind speed (m/s) [46] | 2.31 | __0.96__ | 0.56 | 2.32 | 0.95 | 0.56 | 2.29 | 0.92 | 0.56 | 2.29 | 0.92 | 0.57 |
| | 13 | 12 | 7 | 5 | 2 | 7 | 11 | 15 | 15 | **17** | 17 | 17 |

**(b)**

| 30 Jun – 4 Jul 2015 | UKC2aow | | | UKC2ao | | | Forced UKA2h | | | Forced UKA2g | | |
|---|---|---|---|---|---|---|---|---|---|---|---|---|
| | RMSE | \|Bias\| | $r^2$ | RMSE | \|Bias\| | $r^2$ | RMSE | \|Bias\| | $r^2$ | RMSE | \|Bias\| | $r^2$ |
| [A] Air temp (° C) [537] | **__1.96__** | **__0.74__** | 0.77 | **__1.96__** | **__0.74__** | **__0.77__** | 1.99 | 0.78 | 0.76 | 2.03 | 0.80 | 0.76 |
| | 107 | 101 | 108 | 152 | 149 | 151 | 151 | 119 | 127 | 127 | **168** | 151 |
| [A] Wind speed (m/s) [412] | 1.86 | **__0.70__** | 0.38 | 1.89 | 0.70 | __0.38__ | 1.87 | 0.70 | 0.38 | 1.87 | 0.71 | 0.37 |
| | 107 | 107 | 96 | 107 | 71 | 100 | 81 | 101 | 95 | **117** | **133** | **121** |

**(c)**

| 30 Jun – 4 Jul 2015 | UKC2aow | | | UKC2ao *[UKW2l]* | | | Forced UK[OW]2h | | | Forced UK[OW]2g | | |
|---|---|---|---|---|---|---|---|---|---|---|---|---|
| | RMSE | \|Bias\| | $r^2$ | RMSE | \|Bias\| | $r^2$ | RMSE | \|Bias\| | $r^2$ | RMSE | \|Bias\| | $r^2$ |
| [O] SST (° C) [87] | 0.75 | 0.53 | 0.39 | __0.75__ | 0.53 | 0.39 | 0.77 | 0.53 | 0.38 | 0.76 | 0.54 | __0.42__ |
| | 11 | 16 | 8 | 15 | 12 | 14 | 15 | 17 | 14 | **46** | **42** | **51** |
| [O] SSH (m) [41] | 0.91 | 0.46 | 0.83 | 0.91 | 0.46 | 0.83 | 0.91 | 0.46 | 0.83 | **__0.91__** | **__0.45__** | **__0.83__** |
| | 5 | 7 | 4 | 7 | 4 | 5 | 2 | 3 | 9 | **26** | **26** | **22** |
| [W] Sig. wave ht. (m) [66] | **__0.22__** | **__0.10__** | **__0.66__** | *0.26* | *0.13* | *0.62* | 0.26 | 0.12 | 0.60 | **__0.18__** | **__0.08__** | **__0.73__** |
| | 10 | 16 | 17 | *0* | *6* | *3* | *1* | 6 | 5 | **55** | **38** | **41** |
| [W] Peak period (s) [12] | 1.87 | 1.23 | 0.59 | *1.92* | *1.21* | *0.58* | 2.02 | 1.34 | 0.57 | 1.97 | 1.30 | 0.60 |
| | 6 | 6 | 5 | *2* | *4* | *1* | *0* | 1 | 2 | 4 | 1 | 4 |

5  **Table 9: Summary RMSE, bias and r2 correlation coefficient statistics for the 30 June - 3 July 2015 case study. Average statistics are computed from a comparison of model output against in-situ observations of (a) all atmosphere [A] model variables, (b) atmosphere [$A_o$] model variables over sea grid cells only and (c) ocean [O] and wave [W] model variables. The relevant forced configuration with high-resolution forcing UKW2h, UKO2h and UKA2h are treated as the reference control model. Results showing a statistically significant difference from the reference according to a t-test at the 95% confidence level are underlined.**
10  **Those also showing an improvement against the reference are also in bold. Also listed for each variable in lighter text is the number of sites for which the model configuration provided the best statistic relative to the other model configurations tested for this case. The configuration with the highest number of sites listed is highlighted in bold. Numbers in square brackets show the number of observation locations assessed for each variable.**



| Met Office Unified Model branch name | Code revision | Purpose |
|---|---|---|
| pkg/andymalcolm/vn10.1_Cray_optimisations | 8184 | Root UM10.1 code base, with optimisation |
| pkg/Config/vn10.1_PS37_UKV_Configuration | 13196 | Parallel Suite 37 default code configuration settings |
| huwlewis/vn10.1_ukep_hydrol_for_jules | 18816 | Support for slope-dependent PDM in JULES |
| huwlewis/vn10.1_ukep_rivrouting | 18992 | Enable relevant river routing stashcodes for diagnostics |
| juanmcastillo/vn10.1_move_call_to_trap_uv | 18967 | Require move of call to wind limiter after solver |
| juanmcastillo/vn10.1_ukep_mslp_wave_order | 19048 | Fix model field coupling order to avoid deadlocks |
| juanmcastillo/vn10.1_AtmWaveOasis | 23941 | Add wave coupling capability to Unified Model |
| huwlewis/vn10.1_non_ctile_coupling | 23959 | Ensure valid coupling where coastlines not overlap |
| juanmcastillo/vn10.1_fix_io_server | 24543 | Fix error when using UM IO server for coupled runs |

**Table C1: Summary of Met Office Unified Model code branches merged and used in the UKA2 and UKC2 system, containing relevant adaptations from the version 10.1 baseline code. Model codes are accessible via https://code.metoffice.gov.uk/trac/um/wiki. Registered users can directly access each code branch at the revision used in UKC2 by following the direct code revision links provided.**

| JULES branch name | Code revision | Purpose |
|---|---|---|
| pkg/Config/vn4.2_PS37_UKV_Configuration | 2426 | Root JULES vn4.2 code base at PS37 UKV |
| huwlewis/vn4.2_ukep_hydrol | 3254 | Implement WP1 recommended slope-based PDM |
| huwlewis/vn4.2_river_fixes | 3274 | Apply code modification to river routing not at vn4.2 |
| juanmcastillo/vn4.2_AtmWaveOasis | 4447 | Implement wave coupling within surface exchange |

**Table C2: Summary of JULES code branches merged and used in the UKA2/UKL2 and UKC2 system, containing relevant adaptations from the version 4.2 baseline code. Model codes are accessible via https://code.metoffice.gov.uk/trac/jules/wiki. Registered users can directly access each code branch at the revision used in UKC2 by following the direct code revision links provided.**

| RFM parameter | Definition | UKC2 default value | Typically used parameter range |
|---|---|---|---|
| cland | land wave speed [m s$^{-1}$] | 0.4 | 0.2 – 0.4 |
| criver | river wave speed [m s$^{-1}$] | 0.5 | 0.62 – 1.0 |
| cbland | subsurface land wave speed [m s$^{-1}$] | 0.05 | 0.1 – 0.075 |
| cbriver | subsurface river wave speed [m s$^{-1}$] | 0.05 | 0.15 – 0.1 |
| retl | land return flow fraction (<1) | 0.005 | 0.0 – 0.0005 |
| retr | river return flow fraction (<1) | 0.005 | 0.005 – 0.0005 |
| a_thresh | threshold area [number of cells] | 13 | 1 – 10 |

**Table C3: Parameters available to define the RFM river routing algorithm (see Bell et al. (2007) for details), and values used within UKC2. Typically used parameter value ranges from global to regional applications is also shown for reference.**





| NEMO branch name | Code revision | Purpose |
|---|---|---|
| Trunk | 5518 | Root NEMO vn3.6 code used for UKO2 |
| UKMO/nemo_v3_6_STABLE_copy | 5783 | Bug fixes later implemented in NEMO trunk |
| UKMO/2015_CO6_CO5_shelfdiagnostic | 5666 | Additional output diagnostics enabled |
| UKMO/restart_datestamp | 6336 | Add writing of timestamps in restart file names |
| 2015_V36_STABLE_CO6_CO5_zenv_pomsdwl | 5793 | Enable back compatibility with CO5 configuration |
| UKMO/dev_r5518_bdy_sponge_temp | 5878 | Enable enhanced diffusion near domain boundaries |
| UKMO/dev_r5107_xios_initialize_toyoce | 6242 | Update IO for key_vvl compatibility |
| UKMO/dev_r5107_hadgem3_mct | 5631 | Add OASIS-MCT compatibility |
| UKMO/dev/r5107_hadgem3_cplseq | 5646 | Set the required order of coupling fields |
| UKMO/dev_r5107_hadgem3_cplfld | 5592 | Treat non-standard aspects of atmospheric coupling |
| UKMO/dev_r5518_ww3_coupling | 6733 | Enable coupling of variables with wave model |
| UKMO/dev_r5518_rm_um_cpl | 5884 | Remove dependencies on UM, to couple O-W only |
| UKMO/dev_r5518_amm15_test | 6344 | Read AMM15 input files already interpolated to grid |
| UKMO/dev_r5518_sst_landsea_cpl | 6709 | Distinguish NEMO land SST values for coupling |

**Table C4: Summary of NEMO code branches merged and used in the UKC2 system, containing relevant adaptations from the baseline trunk code at version 3.6 (revision 5518). Model codes are accessible via http://www.nemo-ocean.eu. Registered users can**
5  **directly access each code branch at the revision used in UKC2 by following the direct code revision links provided.**

| NEMO compilation keys used in UKC2 and UK)2 | |
|---|---|
| key_zdfgls | GLS generic length scale vertical mixing |
| key_ldfslp | Lateral diffusion |
| key_dynspg_ts | Split-explicit free surface |
| key_vectopt_loop | Inner loop index order |
| key_bdy | Unstructured open boundary conditions |
| key_tide | Tidal potential forcing |
| key_vvl | Variable volume non-linear free surface |
| key_shelf | Implement Met Office shelf seas flux forcing |

**Table C5: Summary of NEMO compile keys used in the UKC2 system.**





| WAVEWATCH III branch name | Code revision | Purpose |
|---|---|---|
| WW3v4/trunk | 966 | Root WAVEWATCH III vn4.18 code used for UKW2 |
| WW3v4/branches/dev/frhl/r966_ww3v4_ukep | 1328 | New implementation of wave coupling to A and O code |

**Table C6: Summary of WAVEWATCH III code branches merged and used in the UKC2 system, containing relevant adaptations from the baseline trunk code at version 4.18 (https://svnemc.ncep.noaa.gov/projects/ww3/released/REL-4.18). Code can be shared with registered users of WAVEWATCHIII, with further details available at http://polar.ncep.noaa.gov/waves/wavewatch/.**

| WAVEWATCH III compilation switches used in UKC2 and UKW2 | | | |
|---|---|---|---|
| ST3 | WAM 4 and variants source term package | RTD | Rotated coordinate system |
| STAB3 | Stability correction (not invoked) | WNT1 | Linear wind speed interpolation in time |
| NL1 | Discrete interaction approximation | CRT1 | Linear current interpolation in time |
| BT1 | JONSWAP bottom friction formulation | WNX1 | Approx. linear wind speed interpolation in space |
| DB1 | Battjes-Janssen depth-induced breaking | CRX1 | Approx. linear current interpolation in space |
| TR0 | No triad interactions | FLX0 | Flux computation included in source terms |
| BS0 | No bottom scattering | RWND | Correct wind speeds for current velocity |
| XX0 | No supplemental source terms | REF0 | No source term for reflection |
| LN1 | Cavaleri and Malanotte-Rizzoli linear input | PR3/UNO | Second order propagation scheme |

**Table C7: Summary of WAVEWATCH III compile switches used in the UKC2 system.**



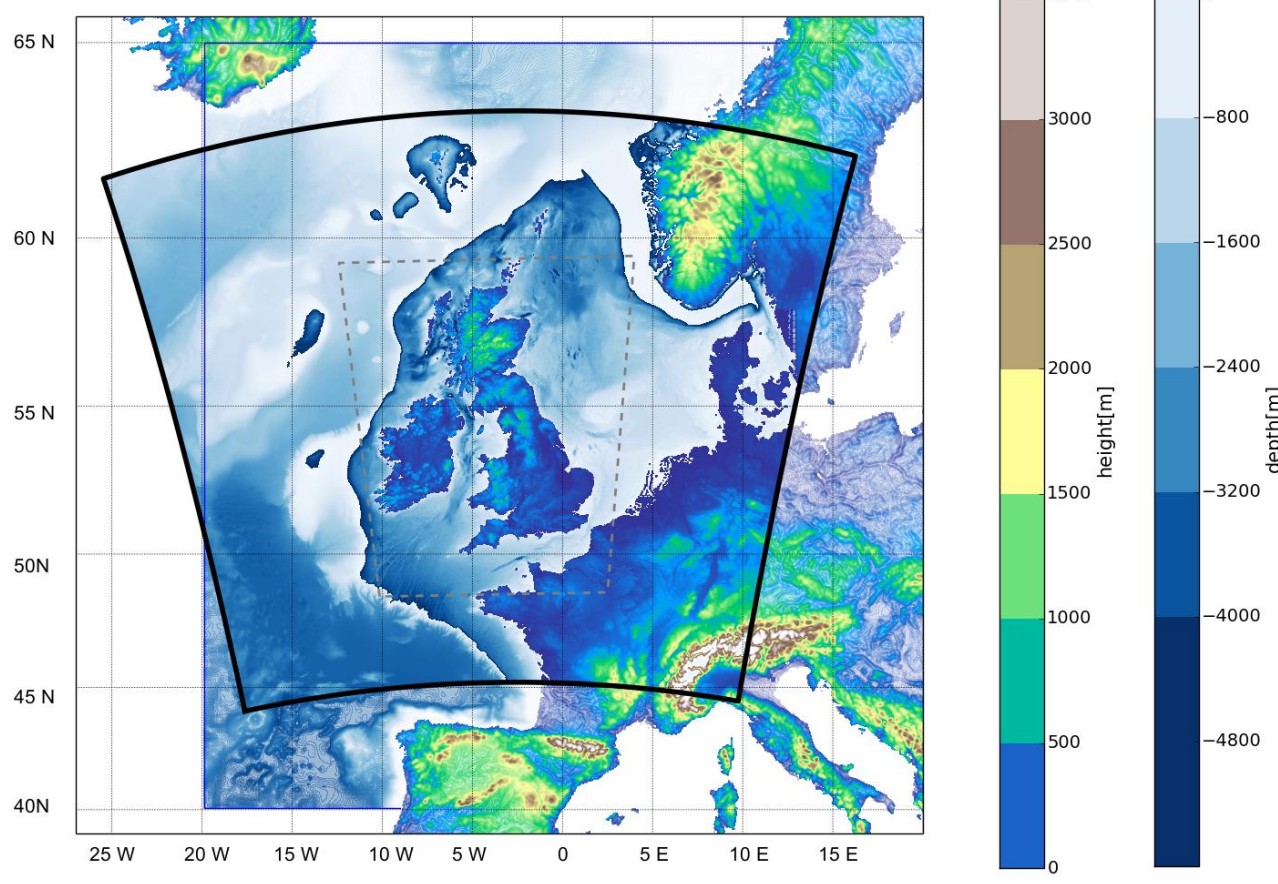

**Figure 1: The UKC2 domain (black boundary) used to define UKA2 atmosphere, UKO2 ocean and UKW2 wave components. The model orography and bathymetry are also shown. The blue outline shows the extent of the current operational AMM7 ocean domain (see Sect. 3.3). The gray dashed area shows the approximate extent of the regular 1.5 x 1.5 km inner region of the UKA2 grid.**

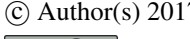


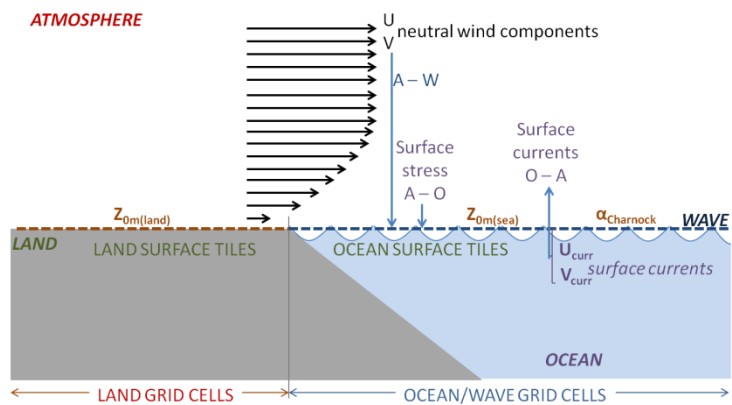

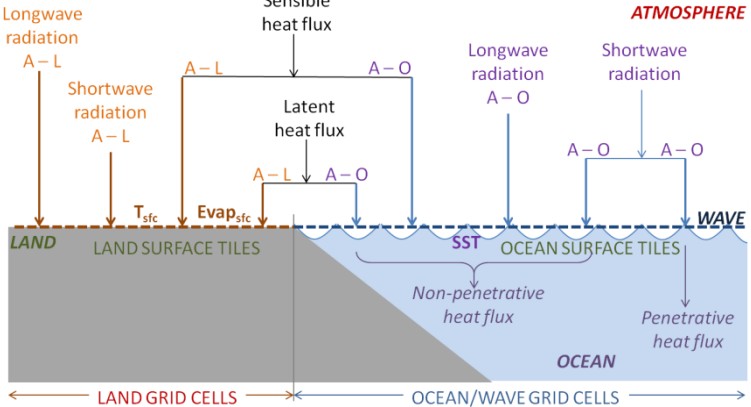

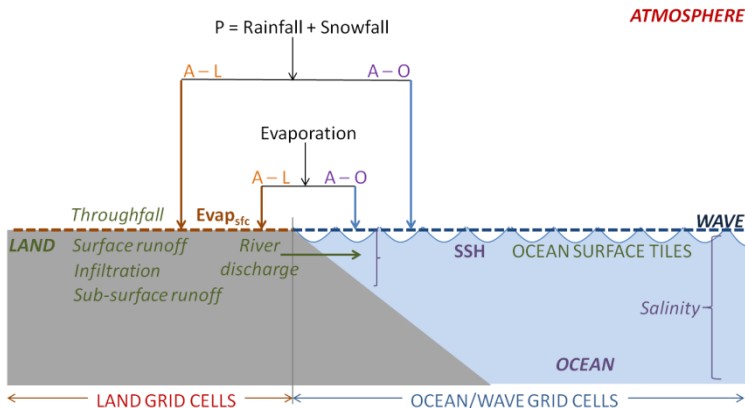

**Figure 2: Summary of instantaneous fields exchanged each hour in the UKC2 coupled simulation categorised as (a) momentum, (b) heat and (c) freshwater exchanges between atmosphere-land (UM-JULES), ocean (NEMO) and wave (WAVEWATCH III) components.**



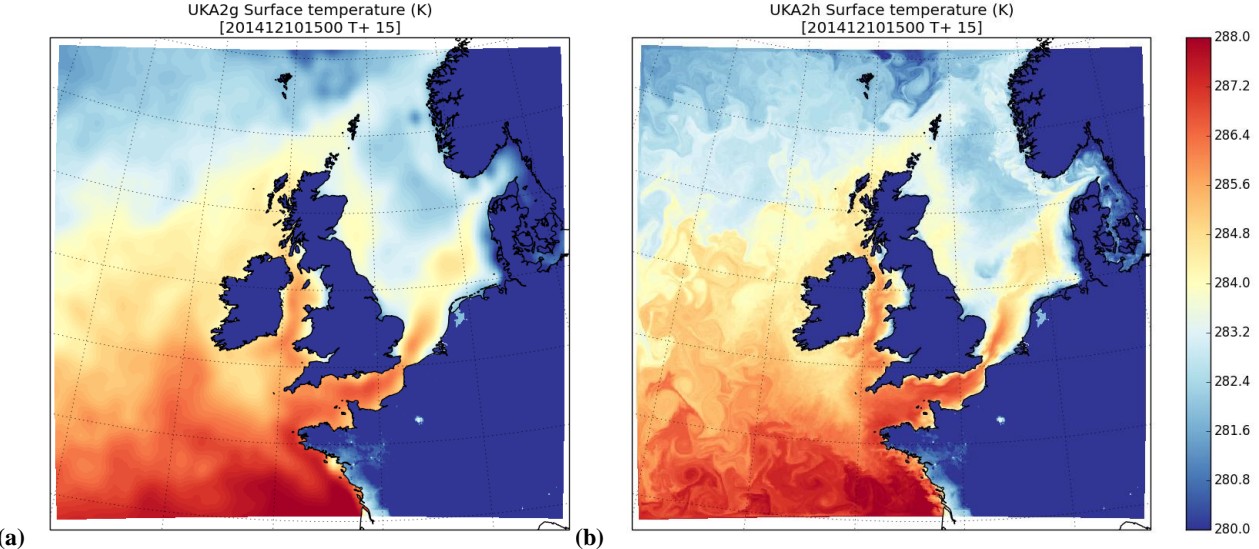

**Figure 3: Sample snapshot case study output of surface temperature showing (a) UKA2g: persisted SST interpolated from a 17 km resolution global Met Office Unified Model run and (b) UKA2h: persisted SST interpolated from 1.5 km resolution UKO2 ocean run. Note the temperature scale is truncated to highlight SST variability rather than the (evolving) land temperatures.**

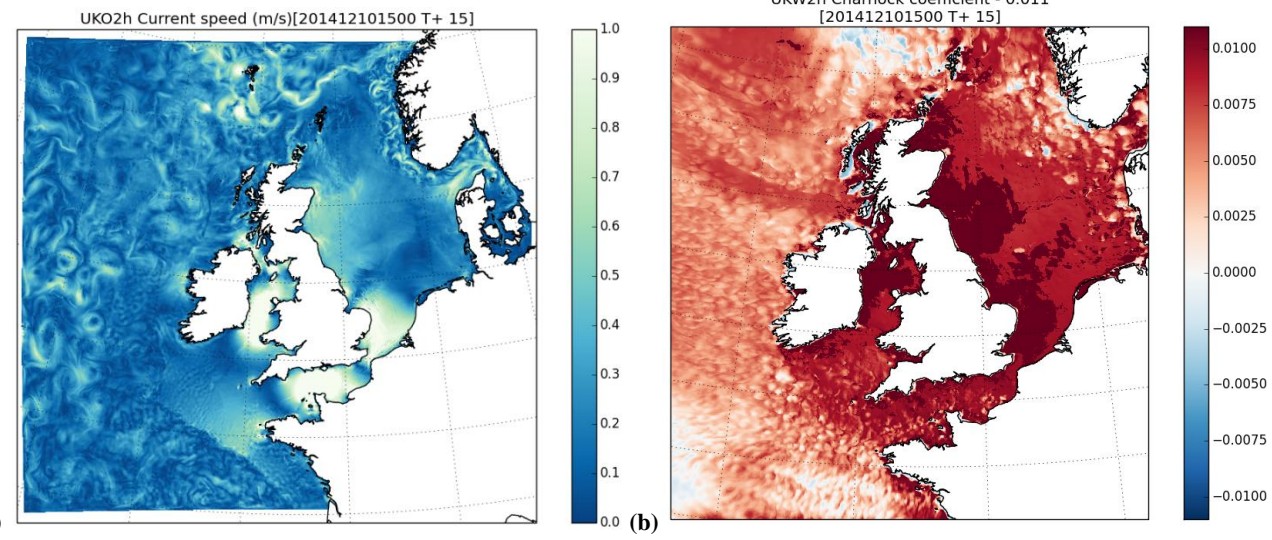

**Figure 4: Sample snapshot case study output of (a) surface currents simulated by UKO2h ocean configuration and (b) Charnock parameter simulated by UKW2h wave configuration, plotted relative to the UKV constant value of 0.011.**





**Figure 5: Summary of UKC2 system case study evaluation configurations in component-only and coupled mode. Arrows highlight the dependencies between forcing used in the control simulations.**





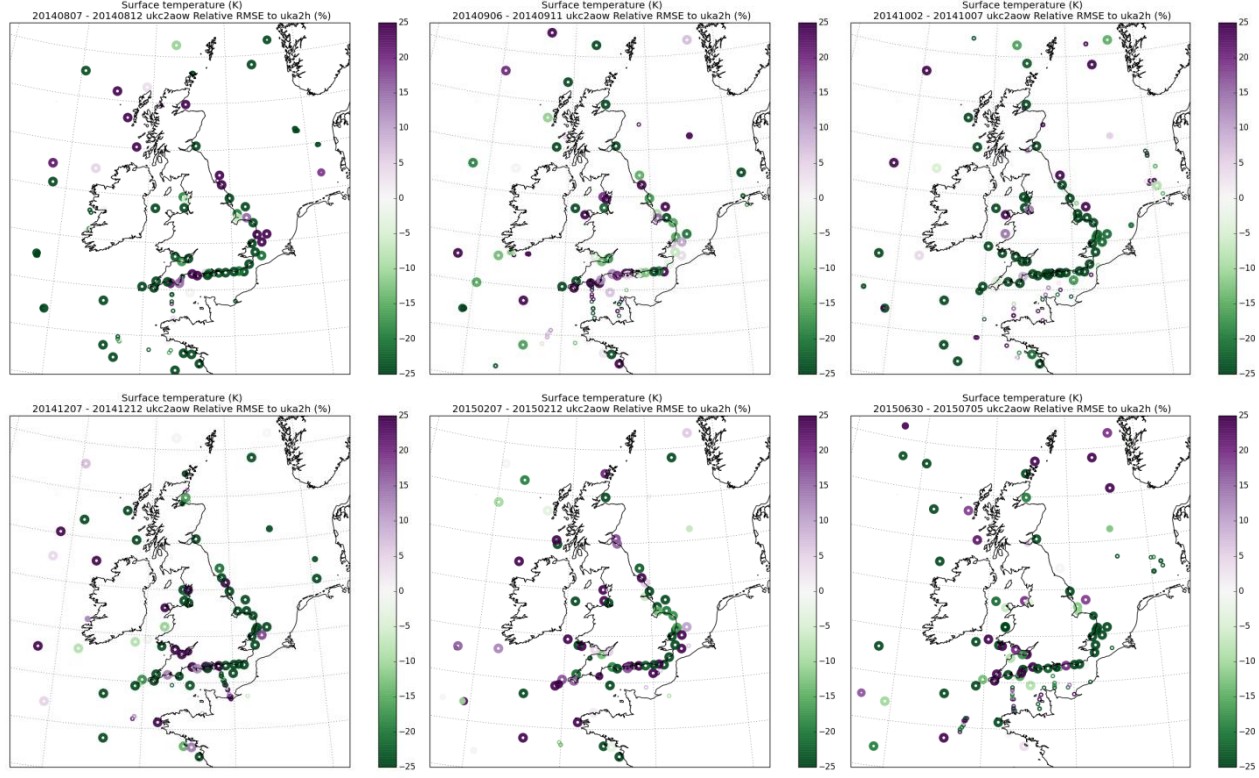

**Figure 6: Differences in case study atmosphere model RMSE statistics for sea surface temperature, comparing statistics computed for the fully coupled UKC2aow simulation as a % difference relative to the atmosphere-only UKA2h runs in which the initial SST is persisted. Green sites indicate an overall reduction in RMSE relative to the control. Results are shown for 5-day case study periods in (a) August 2014, (b) September 2014, (c) October 2014, (d) December 2014, (e) February 2015 and (f) July 2015. RMSE statistics are computed comparing UKC2aow and UKA2h model outputs with in-situ observations. Note that only an inner section of the domain is presented for clarity.**





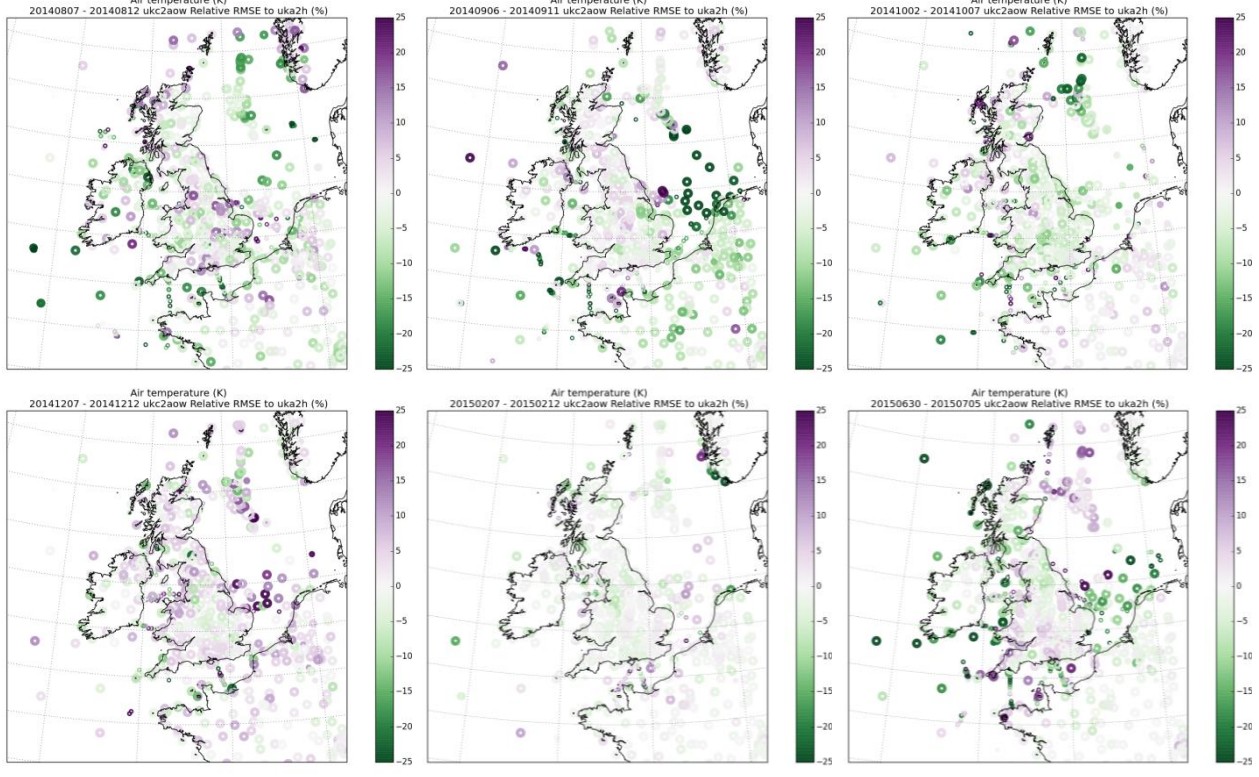

**Figure 7: Differences in case study atmosphere model RMSE statistics for 1.5 m air temperature, comparing statistics computed for the fully coupled UKC2aow simulation as a % difference relative to the atmosphere-only UKA2h runs. Green sites indicate an overall reduction in RMSE relative to the control. Results are shown for 5-day case study periods in (a) August 2014, (b) September 2014, (c) October 2014, (d) December 2014, (e) February 2015 and (f) July 2015. RMSE statistics are computed comparing UKC2aow and UKA2h model outputs with in-situ observations. Note that only an inner section of the domain is presented for clarity.**




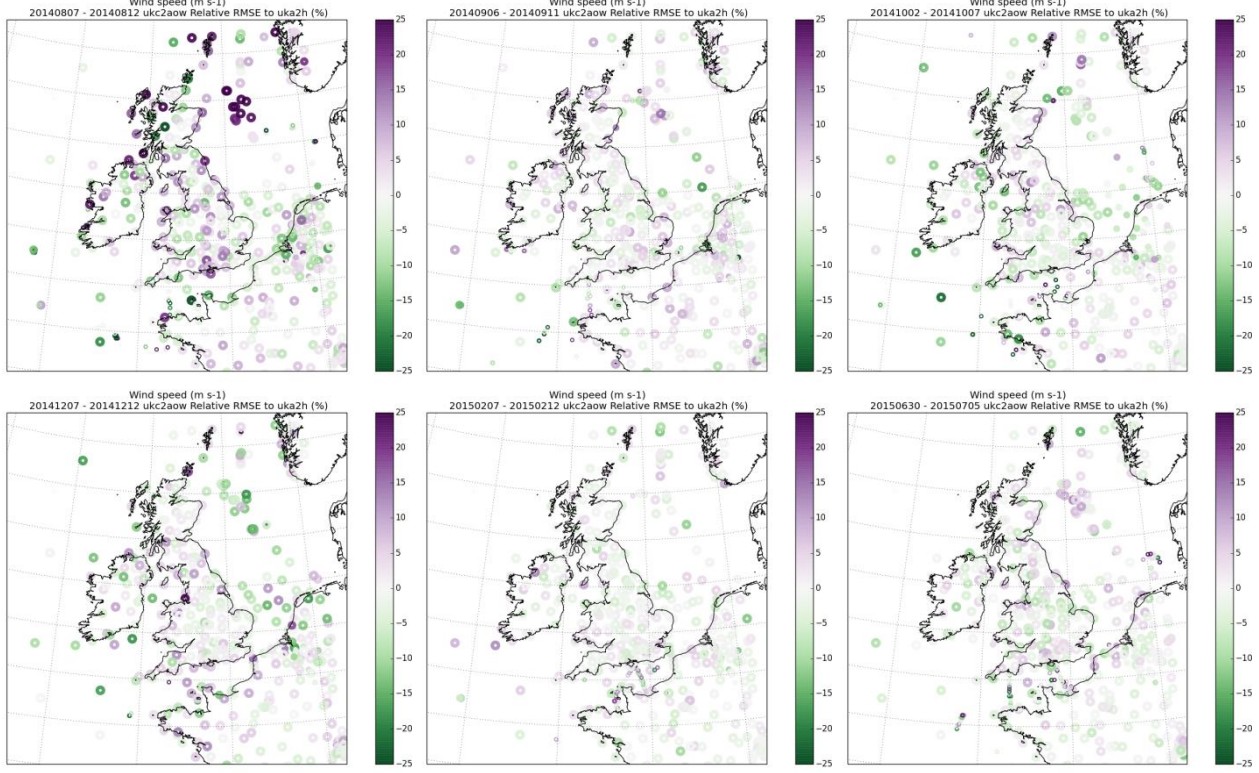

**Figure 8: Differences in case study atmosphere model RMSE statistics for near surface wind speed, comparing statistics computed for the fully coupled UKC2aow simulation as a % difference relative to the atmosphere-only UKA2h runs. Green sites indicate an overall reduction in RMSE relative to the control. Results are shown for 5-day case study periods in (a) August 2014, (b) September 2014, (c) October 2014, (d) December 2014, (e) February 2015 and (f) July 2015. RMSE statistics are computed comparing UKC2aow and UKA2h model outputs with in-situ observations. Note that only an inner section of the domain is presented for clarity.**





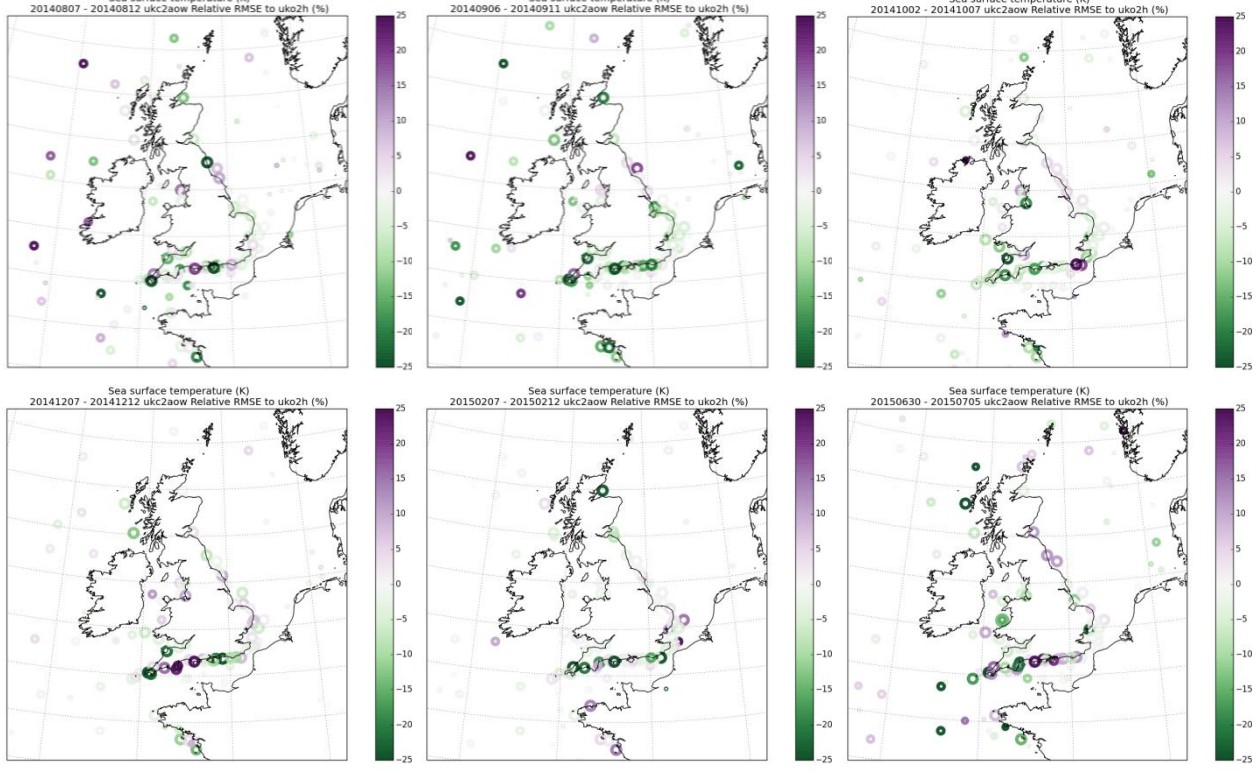

**Figure 9: Differences in case study ocean model RMSE statistics for sea surface temperature, comparing statistics computed for the fully coupled UKC2aow simulation as a % difference relative to the ocean-only UKO2h runs with high resolution meteorological forcing. Green sites indicate an overall reduction in RMSE relative to the control. Results are shown for 5-day case study periods in (a) August 2014, (b) September 2014, (c) October 2014, (d) December 2014, (e) February 2015 and (f) July 2015. RMSE statistics are computed comparing UKC2aow and UKO2h model outputs with in-situ observations. Note that only an inner section of the domain is presented for clarity.**



**Figure 10: Distribution of in-situ observation sites for which each of UKC2aow, UKC2ao, UKO2h or UKO2g ocean model SST outputs provided the lowest RMSE statistic for each 5-day case study. Note that only an inner section of the domain is presented for clarity.**





**Figure 11: Distribution of in-situ observation sites for which each of UKC2aow, UKC2ao, UKO2h or UKO2g ocean model SST outputs provided the lowest RMSE statistic for each 5-day case study. Note that only an inner section of the domain is presented for clarity.**







**Figure 12: Distribution of in-situ observation sites for which each of UKC2aow, UKW2l, UKW2h or UKW2g wave model significant wave height outputs provided the lowest RMSE statistic for each 5-day case study. Note that only an inner section of the domain is presented for clarity.**





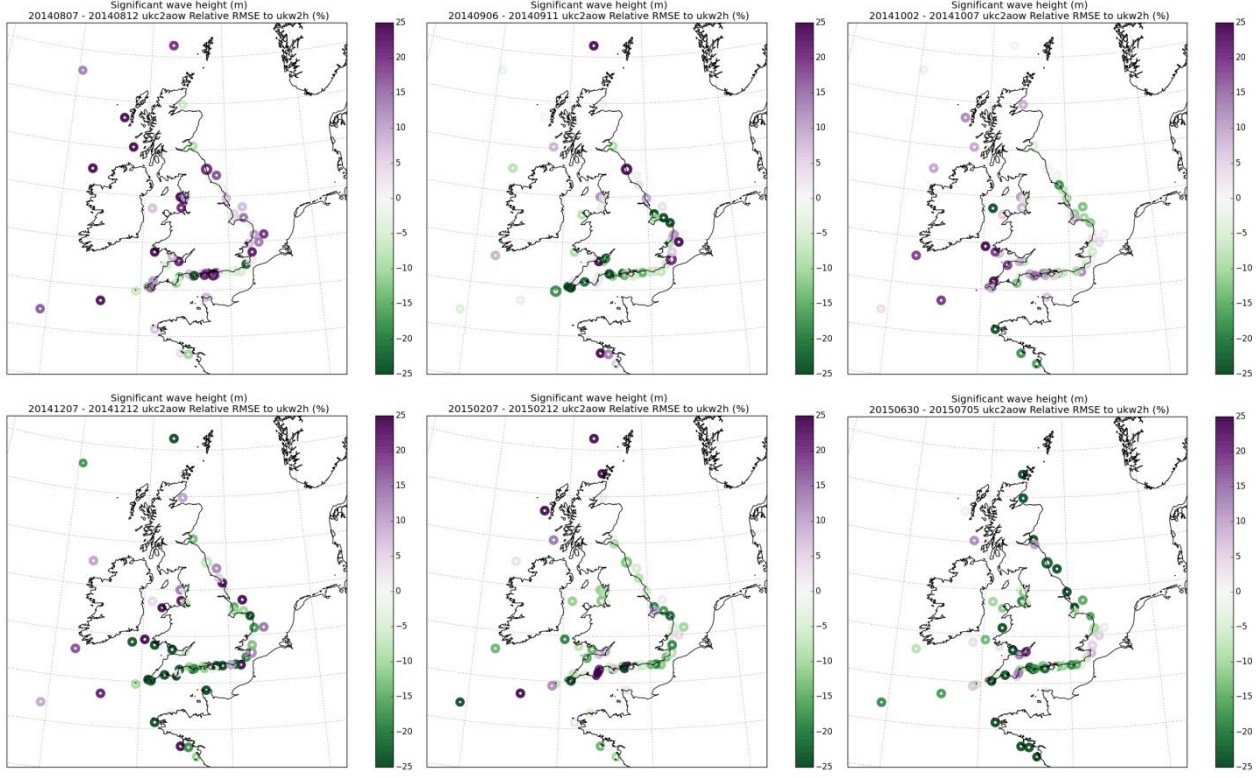

**Figure 13: Differences in case study wave model RMSE statistics for significant wave height, comparing statistics computed for the fully coupled UKC2aow simulation as a % difference relative to the wave-only UKW2h runs with high resolution wind forcing. Green sites indicate an overall reduction in RMSE relative to the control. Results are shown for 5-day case study periods in (a) August 2014, (b) September 2014, (c) October 2014, (d) December 2014, (e) February 2015 and (f) July 2015. RMSE statistics are computed comparing UKC2aow and UKW2h model outputs with in-situ observations. Note that only an inner section of the domain is presented for clarity.**



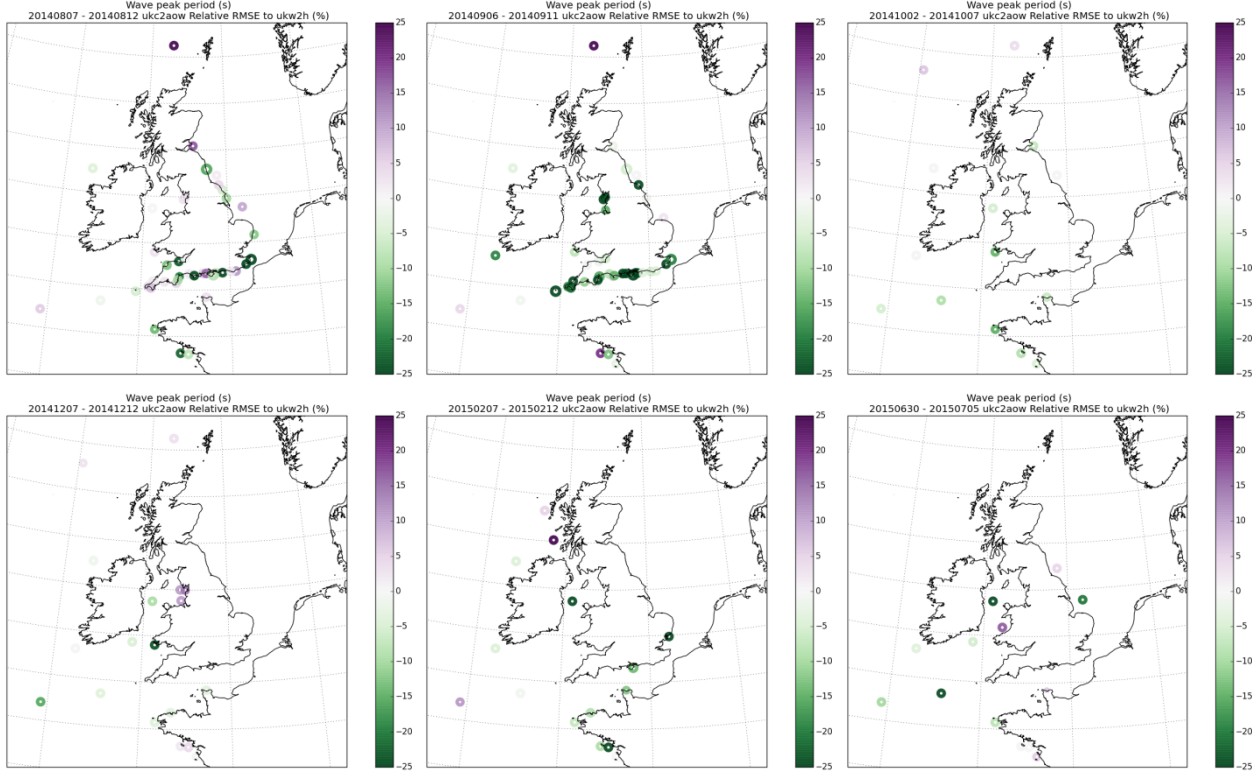

**Figure 14: Differences in case study wave model RMSE statistics for wave peak period, comparing statistics computed for the fully coupled UKC2aow simulation as a % difference relative to the wave-only UKW2h runs with high resolution wind forcing. Green sites indicate an overall reduction in RMSE relative to the control. Results are shown for 5-day case study periods in (a) August 2014, (b) September 2014, (c) October 2014, (d) December 2014, (e) February 2015 and (f) July 2015. RMSE statistics are computed comparing UKC2aow and UKW2h model outputs with in-situ observations. Note that only an inner section of the domain is presented for clarity.**