# Peer review of "The UKC2 regional coupled environmental prediction system"

_Geoscientific Model Development, 2017_

## Referee Comment (RC1) · Anonymous Referee #1 · 21 Jun 2017

This paper is a good description of the different components of the UKC2 regional coupled prediction system It is quite honest on what has implemented but also what still needs to be done (and the list is not short). The attempt at validating the system with respect to observations indicate the potential of the system. I will however deplore the very lengthy case by case statical analysis that does not get the message across, is more than likely statistical non significant because of the the short samples and the in inherent errors in the observations

Comments and notes:

page 3, line 19: add Bertin et al. 2015 to your list. This paper show a nice example when the impact of the coupling of the ocean and the wave can be expected to have large impact and another case when it is not. Xavier Bertin, Kai Li, Aron Roland, Jean-

[Figure]

Raymond Bidlot. 2015: The contribution of short-waves in storm surges: Two case studies in the Bay of Biscay. Continental Shelf Research 96, 1-15.

page 5, line 7. When referencing Breivik et al., it should be noted that this piece of work is mostly about how a wave model can alter surface fluxes and provide information on the Stokes drift. It does not however cover other effects such as the wave radiation stress, the impact of wave orbital motion on the bottom stress (see for instance Bertin et al. 2015)

There is some discussions later on addressing the issue of coupling more often than every hour, but how will handle atmospheric fluxes that are modulated by the wave model to be later passed to the ocean model. Do you have de facto a time lag of two hours between what the atmosphere produces and what the ocean sees. Is passing instantaneous fields the right thing to do then, or should there be some averaging over the coupling tome steps?

Page 6, line 25, with the surface current part of the boundary condition (5), the surface stress will be altered and in response the whole surface wind profile will adapt. Lower surface stress usually implies smoother flow and stronger 10m winds (even though the actual stress acting on the ocean surface is less) and vice versa. This has implication when forcing the waves and the oceans.

Page 7, line 20. add: Tau_hf is tabulated beforehand based on the assumption that for high frequencies the stress is in the wind direction and the spectral shape is known (fˆ-5)

Page 7, line 27: the value 0.0095 is not an universal constant and it should be noted that it can be used as a tuning parameter in order to get the mean value of the Charnock and hence the mean behaviour of the drag coefficient to fit to observation (Edson et al. 2013)

Section 4.2: what happened to all wave observations from the North Platforms (see

Andy Saulters)?

page 27, line 33: COWAST -> COAWST

---

## Referee Comment (RC2) · Anonymous Referee #2 · 17 Jul 2017

Recommendation: Paper "The UKC2 regional coupled environmental prediction system" by Huw Lewis et al. is a thorough (if not yet exhaustive) and well-written presentation of a new fully coupled regional operational setup that deserves to be published after a minor revision.

Apart from agreeing with the authors that there is a lot be done I am left with few remarks.

The work presents several case studies performed using various implementations of atmosphere-land-ocean-wave modeling setups with different levels of coupling between the modeling system components, along with estimates of influence of each specific coupling interaction, as compared to the control (i.e. not coupled) setup. Data assimilation was not implemented in any of the modeling setups. The paper shows

(and admits) that two-way coupling will not solve all the modeling issues but is nevertheless, at least in the UK case, worth pursuing on regional scales.

Such analyses of the impact of various types of coupling are surely of interest to various modeling groups around the globe. The description in the paper is sufficient to allow, at least in principle, reproducibility of simulations to modelers working with OASIS or other kinds of coupled setups (like COAWST, for example).

General comments: In the future, it would be worth simulating longer time windows to improve the robustness of the statistics and check for possible model drifts, especially in the coupled systems.

Specific comments: P8 L15-20: the authors state the formulas for computing sensible (13) and latent fluxes (14) but they do not define the turbulent exchange coefficients or explain how they were calculated (or perhaps i missed this.). How does the atmospheric model compute the fluxes? This should be explicitly included since only the fluxes (and not the bulk quantities) are exchanged during the coupling.

FIGURE 2: Table 1 is very informative and i would suggest that Figure 2 is explicitly made consistent with it. For example: it is clear from Table 1 that SST is being sent from the ocean to the atmosphere in the coupled setups. However, the ARROW indicating the SST feedback to the atmosphere is missing in Figure 2 b). The same holds for Charnock parameter and roughness length in Figure 2 a) and SSH in Figure 2 c). Perhaps this is my problem but i lost some time over this during first read.

P28 L25: COWAST → COAWST
* * *

---

## Editor Comment (EC1) · J. C. Hargreaves (Editor) · 17 Aug 2017

I have been a bit delayed with hunting down code for another paper I am editing, and then also took some holiday. Please can you send me the code for the version of the unified model used in UKC2? Meanwhile, I will attempt to access the other sub-models using the procedures outlined in the manuscript.

---

## Editor Comment (EC2) · J. C. Hargreaves (Editor) · 14 Sep 2017

In order to compile and run the system described in this paper, access must be gained to the individual models, which must then be compiled according to the information in Appendix C (particularly using the information in the tables in Appendix C). Information to this effect needs to be added to the start of the code availability section, as otherwise readers are likely to get almost as confused as I did while trying to access the code and understand how it is put together.

Table C5 does not seem to be mentioned in the text..?

There is also now a snapshot of the exact code developed for this manuscript available on the MetOffice repository. The existence of this snapshot (or whatever you like to call

it) needs to be mentioned in the paper along with some kind of reference for locating it or requesting access to it. If this paper is accepted for GMD, please include some way of linking the paper and the snapshot (eg. within the snapshot, a file with a reference to the final paper including the DOI for the paper).

It would also be appropriate to make a statement on your approach to collaboration. My impression from the email exchange is that you are rather positive and willing to assist others who wish to use this code (subject to the copyright issues of course), and that you would prefer that they correspond directly with you in order to be successful in correctly compiling and using the code. This is not the same for all model developments - often at GMD, the code is simply made available with the caveat that there will be not assistance offered to a user.

I was not able to gain access to WAVEWATCH. It is not that my access was refused, but that I have heard nothing from them almost a month after the request was made. This is a bit of a problem in the sense that the manuscript suggests that the code is available.

Version control for OASIS-MCT, Rose and FCM need adding to the paper. i.e. unless you can be certain that backward compatibility will be retained, we need to know how to access the versions used here.

There is a namelist in the supplement which I think would be better formatted such that it is ready to use (e.g. as a text file). You can add any kind of file to the supplement. Please liase with Copernicus for the best way to do this (most authors just make a compressed file containing all the files they wish to include).

The reviewers both seem to like the paper and suggest minor revisions. I like it too. In that context it may not be necessary to send the paper back to the reviewers after revision. To best enable this, please provide full and clear responses to all the points raised so that I may be able to make the assessment myself!

---

## Author Comment (AC1) · 21 Sep 2017

**Author Response to Topical Editor Initial Decision**

17 May 2017

Dear Editor,

Many thanks for your initial review of this manuscript and for your positive comments. Please find below a marked-up version of the manuscript, which is also re-submitted as the complete manuscript. We respond to your specific queries, which are listed in red, below.

Topical Editor Initial Decision: Publish subject to minor revisions (Editor review) (16 May 2017) by Julia Hargreaves
Comments to the Author:
As I understand it, the purpose of this manuscript is to describe the coupling together of a number of pre-existing models, implementing the coupled system on the regional UK scale and evaluating the coupled model with a view to future use in forecasting. This is in scope for GMD and is a good fit to a model description paper.

It appears that the "2" in UKC2 is the version number. What happened to UKC1? There is only a vague allusion to it, and no reference. Can you explain this a bit more? What was the form of UKC1 and why is there no reference?

Reference to UKC1 is updated in the slightly re-worded Line 4 of Page 3 in the modified manuscript (shown as Page 5 below). UKC1 was an interim prototype version, encompassing an earlier implementation of an atmosphere-ocean coupled system only (i.e. no wave component). No definitive reference exists for UKC1, other than some conference proceedings in which initial results were presented. We appreciate that the numbering system is somewhat unhelpful, therefore, but UKC2 really represents the first full implementation of the atmosphere-land-ocean-wave regional coupled system, with more developed documentation, testing and evaluation etc. We would also recommend the use of UKC2 over UKC1 (even in atmosphere-ocean coupled mode only) to research collaborators given its more mature system design, later code revisions and a modified domain. It is likely that UKC3 will form a more evolutionary development to UKC2 than the change between UKC1 and UKC2. In summary, UKC2 is the result of a two-year development process, supported for collaboration on shared computing platform (see Supplementary Material). Within that development process a frozen configuration termed UKC1 represented an interim stage.

The manuscript has been slightly modified to reflect this, but unfortunately it is not clear that providing very much more detail on UKC1 is of much benefit to the reader.

The Similarity report highlights a few sentences taken from other places but they mostly seem rather unimportant. The only parts I find a bit concerning are those from Lewis et al, Met App, 2015. These read as slightly polemical and, especially as the author lists of the two papers are not identical, please can you reconsider those parts. It is only a sentence or two.

The Introduction paragraphs have been reviewed and reworded, and we trust that this further revision will be acceptable.

I can see you have put quite a bit of work into the code availability section. This is much appreciated. The access must be to the precise version of the model code discussed in the paper. Thus, while the links you have provided are extremely useful for interested readers, it is not clear that readers can easily obtain the correct versions for all of the models. Please consider whether it would be possible to additionally upload a snapshot of the precise

version to a repository such as Zenodo, which also provides a DOI. This may be particularly appropriate for those models which are "continually updated" meaning that readers may be unable, in future, to find the appropriate version of the model. Would it be possible to create a version of the whole coupled system that may be accessed by anyone who qualifies for the MetOffice license? Without this, one might wonder whether the coupled system really exists!

The tables of code branches used in UKC2 (Tables C1, C2 and C4) and Table captions for Tables C1, C2, C4 and C6 have been modified in the updated manuscript to attempt to direct the reader to the exact code branches used. These are all held under code repositories and each revision is therefore exactly traceable and preserved
10 indefinitely, albeit for the lifetime of a given code repository.

Due to the various and differing licence agreements in place covering the MetUM, JULES, NEMO and WAVEWATCHIII codes it is not possible to provide a single snapshot of the code. However, it is hopefully clear to the reader that interested researchers are encouraged to contact the authors to engage in collaborative research
15 involving the sharing of codes. The authors are already engaged in such activity with other groups.

Note that a MetUM code licence is not the only constraint, but licence agreements for JULES, NEMO and WAVEWATCHIII would also be required (but are all available on request from the relevant groups as detailed in the paper). In all cases, codes are only made available to registered users, however for those registered the
20 direct links now provided in the manuscript should now give direct access and the trac systems highlight the code changes implemented withih each branch. Unfortunately there are stronger restrictions on the authors to make the WAVEWATCHIII code readily available, e.g. in providing code links within the paper, as the code is owned by NOAA.

25 If it supports the review of the manuscript, and to demonstrate that the code components of the coupled system exist at least, the authors would be happy to support the Topical Editor or any other reviewers to obtain research access to the MetUM and JULES codes.

We thank you again for your consideration of this work and constructive comments.
30
Regards,

Huw Lewis and co-authors.

[revised manuscript text omitted]

---

## Author Response (AR1)

**Author's Response to reviews on "The UKC2 regional coupled environmental prediction system" by H.W Lewis et al.**

H.W. Lewis et al.

Correspondence to: H.W. Lewis (huw.lewis@metoffice.gov.uk)

**1    Comments from referees**

**1.1 Comments**

As summarised in EC2, the Topical Editor states that "*the reviewers both seem to like the paper and suggest minor revisions*". Both reviewers suggest the paper provides a good description of the UKC2 system and gives an honest appraisal of what has been implemented, but also what still needs to be done and what the limitations are. Both reviewers have made positive comments about the discussion paper and provided a number of specific comments to be addressed. These are detailed and responded to in Section 1.3 below.

The main concern that both referees raise is on the assessment of system performance with regard to a number of 5-day case studies, with issues arising in the generation of robust statistics and their interpretation due to the short samples and inherent errors in the observations. RC1 provides an explicit criticism of this, while RC2 raises a similar theme in stating that "*in the future, it would be worth simulating longer time windows to improve the robustness of the statistics and check for possible model drifts, especially in the coupled systems.*"

**1.2 Response**

The authors thank both referees for their reviews and for their support for the publication of this paper. This UKC2 system description paper represents the first peer-reviewed paper documenting the details of a (UK-based) regional coupled prediction system aimed at short-term weather and ocean prediction applications and research. We envisage that this paper will provide a useful reference to the community and provide a citable source to underpin the publication of further research using the UKC2 system and its successors. Following the publication approach adopted for Met Office Global Atmosphere/Land configurations for example, we also envisage submitting follow-on publications in GMD that highlight changes made in subsequent releases since this configuration and the impacts these have on model performance.

The concern about the length of simulations documented in this paper is well received. Given that this work represents the first tests of the UKC2 system, a first-order comparison between different modes of simulation (i.e. coupled vs. uncoupled) has been attempted. This was conducted using a case study approach, and the results discussed in Section 5 represent 6 different case studies (30 days of simulation in total) covering a spread of seasons and environmental conditions. Section 4.1 highlights that the case study approach is considered a *"suitable starting point for evaluating the impact of coupling on short timescales, with further work planned to assess the impact of coupling on the ocean over longer timescales through longer integrations."* The discussion of results presented in this paper therefore focus on a comparison between coupled and uncoupled simulations with different forcing or initial states, rather than a validation of system skill in the traditional sense of comparison with observations over

a longer period. The discussion does address a number of different case studies however, to avoid the risk of drawing conclusions relevant only to particular conditions or time of year. Tables 4-9 present the full detail of calculated statistics in order to be transparent about the order of magnitude changes in headline statistics between different model configurations, with statistical significance highlighted where relevant using the results of a Student's t-test on the calculated statistics. The key conclusion from this work is expressed in Section 5 that *"these results highlight that all configurations provide at least representing simulations of atmosphere, ocean and wave states. UKC2aow and UKC2ao therefore represent a successful initial development of regional coupled prediction systems at high resolution…."*. This is echoed in the Abstract, which states that *"Results demonstrate that at least comparable performance can be achieved with the UKC2 system to its component control simulations"*. To reiterate our support for the reviewer comments in general for the need for longer integrations, this is now further highlighted in Section 6 in the revised manuscript.

**2.3 Specific comments and responses**

**Comment R1.1:** *page 3, line 19: add Bertin et al. 2015 to your list.*
**Response:**
This is done, assuming that the reviewer intended for this to be added on line 19 of page 2 however.

**Comment R1.2**: *page 5, line 7. When referencing Breivik et al., it should be noted that this piece of work is mostly about how a wave model can alter surface fluxes and provide information on the Stokes drift. It does not however cover other effects such as the wave radiation stress, the impact of wave orbital motion on the bottom stress (see for instance Bertin et al. 2015)*
**Response:**
It is agreed that the Breivik et al. reference does not cover the complete list of potential interactions, and is provided as the most relevant standard example reference on wave effects in NEMO. Feedbacks such as the impact on bottom stress is a current area of development in NEMO and for research using the UK coupled system.

**Comment R1.3:** *There is some discussions later on addressing the issue of coupling more often than every hour, but how will handle atmospheric fluxes that are modulated by the wave model to be later passed to the ocean model. Do you have de facto a time lag of two hours between what the atmosphere produces and what the ocean sees. Is passing instantaneous fields the right thing to do then, or should there be some averaging over the coupling time steps?*
**Response:**
Exploring the impact of coupling frequency is described as an area for ongoing research, and as the reviewer described will need some careful consideration of what information is available in each component. This comment highlights an error in the original manuscript description however, in that coupled fields are calculated as hourly mean values rather than instantaneous. Table 1 and Figure 1 caption have been updated in the revised manuscript to correct this. The order of coupling as listed in Table 1 is also important to ensure that there is no lag between what the atmosphere produces and what the ocean sees in terms of calculated the surface stress in each system for example.

**Comment R1.3:** *Page 6, line 25, with the surface current part of the boundary condition (5), the surface stress will be altered and in response the whole surface wind profile will adapt. Lower surface stress usually implies smoother flow and stronger 10m winds (even though the actual stress acting on the ocean surface is less) and vice versa. This has implication when forcing the waves and the oceans.*

**Response:**
The authors agree with this interpretation, and are in the process of quantifying this effect in ongoing work. Section 2.3.1 has been updated to include a brief comment on this point. The atmospheric forcing applied to ocean-only control simulations in this work has no surface current effects, while this is explicitly represented in
5  UKC2aow and UKC2ao. Work is ongoing to assess the consistency between stress forcing across ocean, wave and atmosphere systems, particularly in the context of comparing coupled and uncoupled runs, but is considered outside the scope of this initial description paper.

**Comment R1.4:** *Page 7, line 20. add: Tau_hf is tabulated beforehand based on the assumption that for high*
10  *frequencies the stress is in the wind direction and the spectral shape is known (f^-5)*
**Response:**
Section 2.3.1 has been updated to include this helpful comment.

**Comment R1.5:** *Page 7, line 27: the value 0.0095 is not an universal constant and it should be noted that it can*
15  *be used as a tuning parameter in order to get the mean value of the Charnock and hence the mean behaviour of the drag coefficient to fit to observation (Edson et al.2013)*
**Response:**
The original manuscript did not attempt to state that this was a universal value, but this line has been updated to attempt to clarify this.

**Comment R1.6:** *Section 4.2: what happened to all wave observations from the North Platforms?*
**Response:**
Figures 12, 13 and 14 have been reprocessed and updated to include the observations made available from offshore oil installations in the North Sea, which were unfortunately omitted from the original manuscript. A
25  change of observed variable from peak to wave mean period (Figure 14) has also been made to increase the number of observations. The corresponding statistics presented in Tables 4-9 for wave variables Sig wave ht. and Peak period.
Results including the new observations are consistent with those in the original manuscript, and further emphasise the potential of the UKC2 system to provide benefits over the wave-only simulations for the case
30  study periods presented. The new data therefore do not alter any of the conclusions discussed in Section 5.3 and therefore only minor updates are included in the manuscript text to reflect the updated figures.

**Comment R1.7:** *page 27, line 33: COWAST -> COAWST*
**Response:**
35  This has been corrected in the revised manuscript. This also covers the final comment in RC2.

**Comment R2.1:** *P8 L15-20: the authors state the formulas for computing sensible (13) and latent fluxes (14) but they do not define the turbulent exchange coefficients or explain how they were calculated (or perhaps i missed this.). How does the atmospheric model compute the fluxes? This should be explicitly included since only the*
40  *fluxes (and not the bulk quantities) are exchanged during the coupling.*
**Response:**
A line describing the calculation of surface fluxes in the MetUM is included in Section 2.3.1 (p6, line 25) – i.e. these are computed according to surface similarity theory using stability-dependent functions according to Beljaars and Holtslag (1991) and Dyer and Hicks (1970). References to Lock et al (2000) and Edwards (2007)
45  are also provided for further detail.

**Comment R2.2:** *FIGURE 2: Table 1 is very informative and i would suggest that Figure 2 is explicitly made consistent with it. For example: it is clear from Table 1 that SST is being sent from the ocean to the atmosphere in the coupled setups. However, the ARROW indicating the SST feedback to the atmosphere is missing in Figure 2 b). The same holds for Charnock parameter and roughness length in Figure 2 a) and SSH in Figure 2 c).*

**Response:**
Figure 2 has been reviewed and updated in light of this helpful comment to reflect the content of Table 1. There was some ambiguity in the original figure between representing exchanged variables and the processes/feedbacks of interest. In Figure 2a) and 2b) the arrows represent fluxes of heat and water respectively between components, and variables listed such as SST, SSH, surface stress etc represent other relevant variables exchanged between systems (but not fluxes, so not shown as arrows). The exception is in Figure 2a), which shows 'fluxes' of wind between Atm and Wave, and of Surface currents between Ocn and Atm. The revised figure attempts to capture the information contained in Table 1, while being more consistent between treatment of flux terms and bulk quantities.

**2.3 Author's changes to manuscript following reviews**

In light of the referee comments, detailed and responded to above, the following changes have been made to the manuscript:

- Additional discussion of stress terms in Section 2.3.1
- More explicit acknowledgment of the need for longer iterations as future work in Section 6.
- Correction of time averaging information in Table 1
- Review and update to way the information is presented in Figure 2
- Improvement in number of wave observations used in Figures 12-14 and Tables 4-9, and corresponding edits to Section 5.3 for consistency.

A limited number of further minor corrections (spelling and mis-labelling) have also been added by the authors to correct errors spotted during the revision of the paper (e.g. Figure 11 caption).

**2       Comments from Topical Editor**

**2.1  Comments**

Topical Editor Julia Hargreaves has attempted to follow the instructions as originally set out in the Code Availability section and the related details in Appendix C. In EC1, a request was made to access the UM code to support this activity, and in EC2 a review of the experience and specific suggestions were provided for improving the clarity of information provided on access to the UKC2 system codes for interested readers. Specific comments are highlighted and addressed in Section 2.3 below.

**2.2  Author's response**

We are indebted to the Topical Editor, Julia Hargreaves, for spending considerable time in testing the processes set out for accessing the various codes used in UKC2, and for her positive suggestions on how to improve the clarity of the Code Availability sections and Appendix C. We have also benefitted from several constructive email exchanges in recent weeks on these matters, and how best to communicate the details of the configuration, when based on using disparate codes with different licenscing agreements, registration procedures, repositories and so on. This will make submission of future updates paper describing the evolution of the UK regional

coupled prediction capability much easier for all. We entirely accept the suggestions provided, and these are addressed in the revised manuscript. Comment-by-comment responses are provided below.

We would like to re-emphasise Julia's comments on our openness to collaboration, and would reiterate the constructive support offered to interested readers in their access to, use and configuration of the codes and tools used within the UKC2 system.

Since EC1, access to the UM code repository has been provided to the Topical Editor in order to support this review. She has also registered to use JULES, and access to the JULES code repository has also been provided as a result. I understand that access to NEMO was provided, and we are still waiting on a response regarding access to WAVEWATCH III. The developers were however supportive of our approach to provide a copy of the WAVEWATCH III code used within UKC2 on a separate repository to support the review process, and provides a permanent archive of the relevant wave code, as referenced in Table C.6 in the updated manuscript.

**2.3 Specific comments and responses**

**Comment E1:** *In order to compile and run the system described in this paper, access must be gained **t**o the individual models, which must then be compiled according to the information in Appendix C (particularly using the information in the tables in Appendix C). Information to this effect needs to be added to the start of the code availability section, as otherwise readers are likely to get almost as confused as I did while trying to access the code and understand how it is put together.*
**Response E1:**
A line to this effect has been added to the start of the Code Availability section as suggested.

**Comment E2:** *Table C5 does not seem to be mentioned in the text..?*
**Response E2:**
Table C.5 is first referenced in Section 3.3 (end of third paragraph) in the main text. The contents of this Table seem more appropriate to put in an Appendix table alongside other technical details rather than being associated with Section 3 however. We agree that it would also be helpful to make reference to Table C.5 again in Appendix C, and this is reflected in the updated text. We also ensure to make explicit reference to all Tables in Appendix C, and have corrected an error in the numbering of Tables in Sections C.3 and C.4.

**Comment E3:** *There is also now a snapshot of the exact code developed for this manuscript available on the MetOffice repository. The existence of this snapshot (or whatever you like to call it) needs to be mentioned in the paper along with some kind of reference for locating it or requesting access to it. If this paper is accepted for GMD, please include some way of linking the paper and the snapshot (eg. within the snapshot, a file with a reference to the final paper including the DOI for the paper).*
**Response E3:**
The snapshot copy of the merged branches of code used for atmosphere, land, ocean and wave components was created at https://code.metoffice.gov.uk/svn/utils/ukeputils/trunk/gmd-2017-110 (repository revision 1119) to support the Topical Editor in her review. This is a closed-access repository, for which permission can be provided to users registering via https://code.metoffice.gov.uk/trac/home.
A README file exists at that location which includes a link to the GMD discussion paper, and will be updated to reference the final paper and DOI.
Even with the existence of this snapshot, it is still true to say that users wishing to access and use the codes used in UKC2 will need to register for each individual system. A line has been added at the start of the revised Code

Availability section and again at the start of Appendix C to acknowledge the existence of the snapshot, with guidance to contact the authors for further information and guidance.

**Comment E4:** *It would also be appropriate to make a statement on your approach to collaboration.My impression from the email exchange is that you are rather positive and willing to assist others who wish to use this code (subject to the copyright issues of course), and that you would prefer that they correspond directly with you in order to be successful in correctly compiling and using the code.*

**Response E4:**
The authors are very open and willing to assist potential collaborators in using this configuration, or in its subsequent development by other researchers for different applications. This comment was already made in regards to sharing the outputs from UKC2 simulations in the Data Availability section. A clearer statement of the collaborative support intended on access to the system itself has been added in Code Availability section under a new heading in order to reinforce this open approach.

**Comment E5:** *I was not able to gain access to WAVEWATCH. It is not that my access was refused,but that I have heard nothing from them almost a month after the request was made.This is a bit of a problem in the sense that the manuscript suggests that the code is available.*

**Response E5:**
The authors also sent an email to the WAVEWATCH developers on behalf of the Topical Editor to help support the request for access. However, approval was also provided by the developers at that time for the snapshot of the WAVEWATCHIII code branch used in UKC2 to be archived on the https://code.metoffice.gov.uk/svn/utils/ukeputils/trunk/gmd-2017-110 repository. This may have been considered sufficient by the WAVEWATCH developers to support the review. This provides an accessible archive of the code, and is now linked directly from Table C6.
Readers should also be reassured that it is clear from http://polar.ncep.noaa.gov/waves/wavewatch/ that WAVEWATCH III is distributed under an open source style license, through a password protected distribution site. This web page also advises that code requests may take up to a month for a response to be provided.

**Comment E6:** *Version control for OASIS-MCT, Rose and FCM need adding to the paper. i.e. unless you can be certain that backward compatibility will be retained, we need to know how to access the versions used here.*

**Response E6:**
The requested information has been added to the code availability section.

**Comment E7:** *There is a namelist in the supplement which I think would be better formatted such that it is ready to use (e.g. as a text file). You can add any kind of file to the supplement.*

**Response E7:**
The authors agree this would be a better and easier approach, and will provide a text file of namelists in the revised submission as suggested.

**2.3 Author's changes to manuscript following topical editor review**

In light of the topical editor's input, a new snapshot archive of the UKC2 system code has been established at the https://code.metoffice.gov.uk/svn/utils/ukeputils/trunk/gmd-2017-110 repository.
The Code Availability and Appendix C sections of the paper has been updated to clarify the process of accessing codes, and to reiterate the potential support available from the authors.

[revised manuscript text omitted]

---

## Author Response (AR2)

**Author's Response to Topical Editor Decision on "The UKC2 regional coupled environmental prediction system" by H.W Lewis et al.**

H.W. Lewis et al.

Correspondence to: H.W. Lewis (huw.lewis@metoffice.gov.uk)

**1 Comments from Topical Editor**

**1.1 Comments to the Author:**

As both the authors and reviewers make clear, this manuscript marks the start of something, not a final product. Such steps need to be documented so that future progress can be made. My impression is that the authors have done the best they can in their present circumstances to describe the new coupled modelling system and code developments, and so, on these grounds I intend to accept the paper, even though it could be argued that some aspects of the paper fall short of the ideals expressed in the GMD webpages.

**1.2 Author's Response**

The authors once again recognise and appreciate the time and care taken by the Topical Editor to enhance the usability of this manuscript, and highlight again that the changes suggested will greatly improve future submissions also. We also appreciate the pragmatic approach suggested, noting that the authors are attempting to work within the development practices and webpages/citeable material established separately across the disparate model codes and supporting tools and infrastructure used within UKC2. Where there are still aspects of this paper which fall short of the ideals expressed for GMD, we will continue to work with the relevant code developers to improve the situation ahead of submission of a UKC3 description paper for example.

**1.2.1 Response to specific comments and minor revisions requested**

**Comment E.1:** I was given access to Wavewatch but can only download the distributions. The good news is that model version 4.18 is accessible; but the bad news is that I have not been given access to the development tree so can't see the information pertinent to table c6. So - it seems that the information provided in the manuscript is not sufficient to attain the correct level of access.

**Response E.1:**
It is good to learn that access was finally provided by the WAVEWATCHIII developers, as a demonstration of the availability of the UKC2 codes to collaborators. This should also, we hope, streamline future reviews. As detailed further below, the issue found with Table C.6 was in fact a result of an inadvertent error on the part of the authors rather than an issue with the development tree. Following our response to the initial Referee and Topical Editor's reviews (Sept 2017), Table C.6 was updated to point to the code.metoffice.gov.uk repository 'archive' version of the WAVEWATCH code. However, the link created in the latest revision points to the svn repository URL which requires registered user login, rather than (as provided in Table C.1 for example) pointing to the corresponding trac page, which provides a more immediately accessible and human-readable interface to reading the code and viewing revision history etc. The link in Table C.6 (and also in the text referring to the

archive in Appendix C on p36) have been updated to point to the trac rather than svn link, so as to avoid this access error. In summary, it could be possible to refer to this code using either:
https://code.metoffice.gov.uk/trac/utils/browser/ukeputils/trunk/gmd-2017-110/ukw2/r966_ww3v2_ukep or
https://code.metoffice.gov.uk/svn/utils/ukeputils/trunk/gmd-2017-110/ukw2/r966_ww3v2_ukep/

The former gains access to the human-readable trac pages, while the latter points to the code location itself (e.g. as required for checking out the code). For consistency, all links provided across Tables C1-C6 point to the equivalent trac pages and should be immediately accessible to registered readers.

**Comment E.2:** Thanks for adding the version numbers for the various code on github. There remains an issue with OASIS3-MCT. All I can do is download the latest version. Is there a way of obtaining a particular versions?

**Response E.2:** The authors are unaware of direct methods to do this, but suggest that if required access to OASIS3-MCT vn2.0 would be made available on request from the CERFACS developers. It is our expectation that the functionality required in the UKC2 system is also compatible with the latest available OASIS3-MCT vn3.0.

**Comment E.3:** Weblinks are ephemeral, and at GMD we prefer information to be made more permanently accessible with a DOI. However, given the "step along the way" nature of this manuscript, and the hope and expectation that it will be built upon in a relatively short period of time, I will permit some weblinks. However, please check them all. I found that several weblinks in the current manuscript already resolve to versions of 404!!! What clearer indication of this ephemeral nature of weblinks is needed!? I suggest you consider whether any of the information can be made more permanently accessible. Failing that, please provide sufficient information related to the link such that authors can use search engines to find the relevant data in the case where the weblink fails in future. Please provide a revised manuscript in which all links work! (Note that one of the links in the caption to table c6 resolves to a "no permission" warning)

**Response E.3:** This comment is well received, and we must apologise for some of the weblinks failing – in some cases, this was due to inadvertent leading 'www' in the hyperlinks created, and in another was due to a typo, rather than webpages no longer existing. The webpages referenced also refer to relatively well known systems or projects, such that searching for the latest pages should also provide relevant links. All weblinks used have been checked for access issues, and corrections made where necessary. Further details are provided in Section 1.3 below.

With regard to the link provided in Table C.6 (and Appendix C, p36), the "no permission" warning results from linking to the svn repository URL for the code, rather than the corresponding human-readable and accessible trac URL. This oversight has been corrected in the revised manuscript.

As indicated above, we will continue to work with the system developers and observation data providers ahead of submitting any further update description papers (e.g. UKC3 configuration) to reduce the dependence on weblinks.

**1.3 Changes to the manuscript in light of Minor Revisions requested**

In line with Comment E.2 and our response above, the following changes have been made in the revised manuscript. As these relate to weblink URLs, the changes may not be apparent in the attached revised manuscript with track changes.

p15, line 31 – http://www.marine.copernicus.eu/ replaced by http://marine.copernicus.eu/. Note this is a major EU programme, and we do not expect weblinks to change, or consider that we have a better doi or citation to reference at this time which will provide readers with the direct information required.

p28, line 29 - http://www.metoffice.gov.uk/research/collaboration/um-collaboration changed to http://www.metoffice.gov.uk/research/collaboration/um-partnership. We will also investigate whether there are alternative references to the MetUM licensing for future publications.

p29, line 6 - http://subervsion.apache.org changed to http://subversion.apache.org – with apologies for this oversight.

p36, line 6 - https://code.metoffice.gov.uk/svn/utils/ukeputils/trunk/gmd-2017-110 (svn URL) changed to https://code.metoffice.gov.uk/trac/utils/browser/ukeputils/trunk/gmd-2017-110 (trac URL, human-readible and accessible)

p56, Table C.6 https://code.metoffice.gov.uk/svn/utils/ukeputils/trunk/gmd-2017-110 (svn URL) changed to https://code.metoffice.gov.uk/trac/utils/browser/ukeputils/trunk/gmd-2017-110/ukw2/r966_ww3v2_ukep (trac URL, human-readible and accessible – also updated to point directly to the archived UKW2 code branch).

[revised manuscript text omitted]

---

## Author Response (AR3)

**Author's Response to Topical Editor Decision on "The UKC2 regional coupled environmental prediction system" by H.W Lewis et al.**

H.W. Lewis et al.

Correspondence to: H.W. Lewis (huw.lewis@metoffice.gov.uk)

**1 Comments from Topical Editor**

**1.1  Comments to the Author:**

There has been some to-and-fro with the lead author over email, in regard to the availability, or not, of the WAVEWATCH code. The situation was quite confused but I think we do now understand the situation. It is true that the official WAVEWATCH releases are accessible after registration (and a month's delay!), but this does not give the new user, "developer" status, so they cannot access the svn repository and obtain the precise version of the code used in this paper. However, the appropriate version of the code is on the MetOffice's own repository, so is accessible to collaborators of the UKC2 project. The manuscript needs to be updated to reflect this situation.

It is a bit disappointing that the code is not more generally accessible. In future it would be good to see the MetOffice fall more in to line with institutions world-wide and work towards making some versions of their code publicly accessible, such that the whole UKC code snapshot that, for this paper, is only available on the MetOffice repository, could be uploaded to a public repository with a DOI.

**1.2 Author's Response and Changes to the Manuscript**

The authors once again recognise and appreciate the time and care taken by the Topical Editor to enhance the usability of this manuscript, and highlight again an important change that is required to support use and understanding of the manuscript by interested readers.

We have been in contact with the NOAA WAVEWATCHIII team, and now understand that the original reference provided to the central WAVEWATCHIII repository was aimed at "developer" contributors to the community code. As a generally registered user, it is more appropriate, and possible, to obtain a copy of the WAVEWATCH III released distribution via

http://polar.ncep.noaa.gov/waves/wavewatch/distribution

It is then straightforward to navigate to the vn4.18 version once signed in, by selecting the version of the code desired which will then give a user the link to the tar file of the code.

The UKC2 configuration described in this manuscript is dependent on a branch of WAVEWATCHIII, based on that vn4.18 release. This branch has been made available to readers, once registered as a WAVEWATCHIII user, via the code.metoffice.gov.uk repository archive linked from Table C.6.

The Code Availability description for WAVEWATCH and C.4 have been updated to include reference to the http://polar.ncep.noaa.gov/waves/wavewatch/distribution link, and to clarify the relationship between the code.metoffice.gov.uk archive WW3v4/branches/dev/frhl/r966_ww3v4_ukep branch and the NOAA release.

[revised manuscript text omitted]